# From Optimality to Robustness:
# Dirichlet Sampling Strategies in Stochastic Bandits

**Dorian Baudry**
dorian.baudry@inria.fr

**Patrick Saux**
patrick.saux@inria.fr

**Odalric-Ambrym Maillard**
odalric.maillard@inria.fr

Univ. Lille, CNRS, Inria, Centrale Lille, UMR 9198-CRIStAL, F-59000 Lille, France

## Abstract

The stochastic multi-arm bandit problem has been extensively studied under standard assumptions on the arm's distribution (e.g bounded with known support, exponential family, etc). These assumptions are suitable for many real-world problems but sometimes they require knowledge (on tails for instance) that may not be precisely accessible to the practitioner, raising the question of the robustness of bandit algorithms to model misspecification. In this paper we study a generic *Dirichlet Sampling* (DS) algorithm, based on pairwise comparisons of empirical indices computed with *re-sampling* of the arms' observations and a data-dependent *exploration bonus*. We show that different variants of this strategy achieve provably optimal regret guarantees when the distributions are bounded and logarithmic regret for semi-bounded distributions with a mild quantile condition. We also show that a simple tuning achieve robustness with respect to a large class of unbounded distributions, at the cost of slightly worse than logarithmic asymptotic regret. We finally provide numerical experiments showing the merits of DS in a decision-making problem on synthetic agriculture data.

## 1 Introduction

The $K$-armed stochastic bandit model is a decision-making problem in which a learner sequentially picks an action among $K$ alternatives, called arms, and collects a random reward. In this setting, all rewards drawn from an arm are independent and identically distributed. Hence, we can formally associate each arm $k \in \{1, \ldots, K\}$ with its reward distribution $\nu_k$, with mean $\mu_k$. The objective of the learner is to adapt her strategy $(A_t)_{t \in [T]}$ in order to maximize the expected sum of rewards obtained after $T$ selections (where $T$ is the horizon, unknown to the learner). This is equivalent to minimizing the *regret*, defined as the difference between the expected total reward of an oracle strategy always selecting an arm with largest mean and that of the algorithm, which is equal to

$$\mathcal{R}_T = \mathbb{E}\left[\sum_{t=1}^{T} \mu^\star - \mu_{A_t}\right] = \sum_{k=1}^{K} \Delta_k \mathbb{E}\left[N_k(T)\right] . \tag{1}$$

Here, $N_k(T) = \sum_{t=1}^{T} \mathbb{1}(A_t = k)$ denotes the number of selections of arm $k$ after $T$ time steps, $\mu^\star = \max_{j \in \{1,\ldots,K\}} \mu_j$ and $\Delta_k = \mu^\star - \mu_k$ is called the *gap* between arm $k$ and the largest mean. To assess the performance of a bandit algorithm, one naturally studies the best guarantees achievable by a uniformly efficient algorithm, i.e with sub-linear regret on any instance of a given class of problems. This guarantee was first provided by Lai and Robbins (1985) for 1-dimensional parametric families of distributions, and then extended by Burnetas and Katehakis (1996) for more general families. It

35th Conference on Neural Information Processing Systems (NeurIPS 2021).

states that any algorithm that is uniformly efficient[1] on a family of distributions $\mathcal{F}$ must satisfy

$$\liminf_{T \to \infty} \frac{\mathcal{R}_T}{\log(T)} \geq \sum_{k:\Delta_k > 0} \frac{\Delta_k}{\mathcal{K}_{\inf}^{\mathcal{F}}(\nu_k, \mu^\star)} \ , \quad \mathcal{K}_{\inf}^{\mathcal{F}}(\nu_k, \mu^\star) = \inf_{G \in \mathcal{F}} \{\mathrm{KL}(\nu_k, G) : \mathbb{E}_G(X) > \mu^\star\} \ . \quad (2)$$

A bandit algorithm is then called *asymptotically optimal* for a family of distributions $\mathcal{F}$ when its regret matches this lower bound. When $\mathcal{F}$ is a *Single-Parameter Exponential Family* (SPEF), $\mathcal{K}_{\inf}^{\mathcal{F}}$ is simply the Kullback-Leibler divergence between the distribution of mean $\mu_k$ and that of mean $\mu^\star$ in $\mathcal{F}$, making for a theoretically appealing setting. The quantity $\mathcal{K}_{\inf}^{B}$, corresponding to the family $\mathcal{F}_{[-\infty, B]}$ of distributions supported in $(-\infty, B]$ is also often considered in the literature, see e.g (Honda and Takemura, 2010, 2015; Cappé et al., 2013).

**Overview of existing strategies**   An efficient strategy faces the classical exploration/exploitation dilemma: it needs to obtain enough information from arms that have not been sampled a lot (exploration), but also to sample arms that are well-performing sufficiently often (exploitation). Many algorithms have been proposed for the multi-armed bandits problem (see Lattimore and Szepesvári (2020) for a survey), and we propose in the following a non-exhaustive list of such methods. A first category contains the deterministic index policies, built on the concept of *Optimism in Face of Uncertainty*, the most celebrated of which being the Upper Confidence Bound (UCB) algorithms (Agrawal, 1995; Auer et al., 2002). These algorithms can obtain a logarithmic regret under classical hypothesis on the distributions (e.g bounded, sub-gaussian, sub-exponential, . . . ), and the strongest guarantees have been achieved by kl-UCB Cappé et al. (2013), DMED (Honda and Takemura, 2010), and IMED (Honda and Takemura, 2015), which share a common pattern of solving a convex optimization problem at each round. To be asymptotically optimal, these algorithms require either 1) the knowledge of a specific SPEF for each arm, or 2) a known upper bound on the support of each arm. A second general category is that of *randomized* bandit algorithms, which has been formulated for instance in (Kveton et al., 2019b) as *General Randomized Exploration* (GRE). The common feature of these methods is that, at each time step and for each arm, the algorithm draws an index from a distribution that depends on 1) the rewards observed from the arm, and 2) some knowledge on the arms distributions and chooses the arm with the largest index. Thompson Sampling (TS) (Thompson, 1933; Agrawal and Goyal, 2012) belongs to this category, and a proper choice of Bayesian prior/posterior ensures optimality of TS in SPEF (Korda et al., 2013). Different algorithms using Bootstrapping schemes have also been proposed (Osband and Roy, 2015; Kveton et al., 2019a,b; Wang et al., 2020; Riou and Honda, 2020): they share the idea of computing a noisy mean for empirical samples, enhanced by some exploration aid appropriately tuned to the family of distributions they consider. A last category contains the methods based on *sub-sampling* Baransi et al. (2014); Chan (2020); Baudry et al. (2020, 2021b), that achieve asymptotic optimality in SPEF *without knowing which family*, when all arms share the same. However the proofs heavily rely on properties of the tails of SPEF so the results seem difficult to generalize outside these families.

**Motivations**   While many algorithms achieve optimal regret for bounded distributions with the sole knowledge of the upper bound, the assumptions needed for algorithms working with unbounded distributions (e.g SPEF, sub-Gaussian, sub-exponential) generally assume a known parametric model for the tails. While such assumption entails convenient properties on the theoretical side, the practitioner may have some difficulty to determine which setting/parameters correspond to her problem. Furthermore, this uncertainty raises the question of robustness with respect to these hypotheses. Several works have considered this question: Hadiji and Stoltz (2020) shows that adapting to an unknown bounded range requires a tradeoff between instance-dependent and worst-case regret, and recently (Agrawal et al., 2020; Ashutosh et al., 2021) proved the impossibility of an instance-dependent logarithmic regret for light-tailed distributions without explicit control on the tail parameters. The root cause for this is the lack of compactness of such families $\mathcal{F}$, which allows mass to "leak" at infinity so that maximally confusing distributions with mean $\mu^*$ exist arbitrarily close to $\nu_k$, meaning $\mathcal{K}_{\inf}^{\mathcal{F}}(\nu_k, \mu^*) = 0$. The latter work also introduces a robust variant of UCB, that trades off logarithmic regret for $\mathcal{O}\left(f(T)\log(T)\right)$, where $f$ essentially tracks the possible mass leakage at infinity. These results puts into question the usual hypotheses under which bandit algorithms are designed: considering a *parametric* control of the tails is indeed sensitive to model mis-specification, but on the other hand the examples chosen to prove infeasability results seem a bit extreme for the practitioner. In this paper, we propose simple alternative setups allowing unspecified tail shapes

---

[1]That is, for each bandit on $\mathcal{F}$, for each arm $k$ with $\Delta_k > 0$, then $\mathbb{E}[N_k(T)] = o(T^\alpha)$ for all $\alpha \in (0, 1]$.

but avoiding "mass leakage" to infinity, for instance with mild conditions linking the quantiles and the means of the distributions. We consider in this paper *light-tailed* distributions (see definition in Appendix A.1). This problem is already non-trivial, so we let possible extensions for heavy-tail distributions for future work (e.g with tools like median-of-means, see (Bubeck et al., 2013)).

**Outline** In the novel settings we consider, we want algorithms that require the smallest level of knowledge on the tails of distributions. To this extent, the *Non-Parametric Thompson Sampling* (NPTS, Riou and Honda (2020)) algorithm is a good candidate, considering how little knowledge it requires to reach asymptotic optimality for bounded distributions with known bounds. Furthermore, the flexibility of this algorithm has been recently demonstrated with its adaptation in a risk-aware setting (Baudry et al., 2021a). We provide a generalization of NPTS that we call *Dirichlet Sampling* (DS): we combine the core elements of NPTS and a duel-based framework inspired by (Chan, 2020), introducing data-dependent exploration bonuses. We present the resulting algorithm and detail the technical motivations of this approach in Section 2. We then introduce in Section 3 a first regret decomposition of DS algorithms under general assumptions, and the technical results that allow to fine-tune the algorithm for different families (see Section 3.1). We provide three instances of DS algorithms and their regret guarantees in Section 3.2: *Bounded Dirichlet Sampling* (BDS) tackles bounded distributions with possibly unknown upper bounds, *Quantile Dirichlet Sampling* proposes a first generalization to the unbounded case using truncated distributions. Last, *Robust Dirichlet Sampling* (RDS) has a slightly larger than logarithmic regret for any unspecified *light-tailed* unbounded distributions, making it a competitor to the Robust-UCB algorithm of Ashutosh et al. (2021). Finally, we study in Section 4 a use-case in agriculture using the DSSAT simulator (see Hoogenboom et al. (2019)), which naturally faces all the questions (robustness, model specification) that motivate this work and shows the merit of DS over state-of-the-art methods for this problem.

## 2 Dirichlet Sampling Algorithms

In this section we introduce Dirichlet Sampling, a strategy that aims at generalizing the Non-Parametric Thompson Sampling algorithm of Riou and Honda (2020) outside the scope of bounded distributions with a known support upper bound. For this purpose, we build an adaptive strategy in a duel-based framework, already used in sub-sampling based algorithms like SSMC (Chan, 2020).

**Background** Non-Parametric Thompson Sampling is an index strategy where the index of each arm is a *random re-weighting* of their observations, augmented by an *exploration bonus*. The weights are drawn from the Dirichlet distribution $\mathcal{D}_n = \text{Dir}((1, \ldots, 1))$ for $n$ data, which is the uniform distribution on the simplex $\mathcal{P}^n = \{w \in [0, 1]^n : w^t 1 = 1\}$ and matches the Bayesian posterior (i.e Thompson Sampling) for multinomial arms. The exploration bonus is simply the known upper bound of the support, and avoids under-exploration of potentially "unlucky" good arm. We provide further explanations on the Dirichlet distribution and NPTS respectively in Appendix C.1 and A.3.

The simplicity of NPTS and its strong theoretical guarantees are appealing for further generalization. As we fully depart from the Bayesian approach, considering other exploration bonuses, we derive a new family of algorithms under the name of *Dirichlet Sampling*. We keep the two principles of re-weighting the observations using a Dirichlet distribution and the exploration aid, and explore how to apply them to more general (e.g unbounded) distributions. In particular, we allow in DS some pre-processing of the observations before re-weighting (see section 3.1 and 3.2) and motivate in Section 3.1 the use of a *data-dependent* bonus, that use information from several arms. The complexity introduced by such bonus in the analysis requires a change of algorithm structure, dropping the index policy for a *leader vs challenger* approach (Chan, 2020).

**Round-based algorithm** We define a round as a step of the algorithm at the end of which a set of (possibly several) arms are selected to be pulled. Let $\mathcal{A}_r \subset \{1, \ldots, K\}$ be the subset of the arms pulled at the beginning of a round $r$, we call $T$-round regret the quantity

$$\mathcal{R}_T = \mathbb{E}\left[\sum_{r=1}^{T}\sum_{k=1}^{K} \Delta_k \mathbb{1}(k \in \mathcal{A}_r)\right] = \sum_{k=1}^{K} \Delta_k \mathbb{E}[N_k(T)], \quad (3)$$

where we slightly change the definition of $N_k$ (compared with 1) to $N_k(T) = \sum_{r=1}^{T} \mathbb{1}(k \in \mathcal{A}_r)$. We consider the $T$-round regret for simplicity, as it is a simple upper bound of the regret after $T$

pulls. At the beginning of each round we define a reference arm (leader), and then organize pairwise comparisons called *duels* between this arm and the other arms (challengers). The leader is chosen as the arm with largest sample size,

$$\ell^r \in \underset{k \in \{1,\dots,K\}}{\operatorname{argmax}} N_k(r) \ ,$$

where ties are broken first in favor of the best empirical arm, then with a random choice. A major motivation for this choice is that the leader will have a sample size that is *linear* in the number of rounds, as at least one arm is chosen at each round. This ensures strong statistical properties that we will exploit to design the exploration bonus of DS strategies. Randomizing the index of the leader is also unnecessary: it competes against each challenger with its *empirical mean*. We also dismiss all the arms $k$ that satisfy $N_k(r) = N_\ell(r)$ with the same argument. These choices have a practical interest as they avoid the computation time of drawing the largest weight vectors. We believe this can be an alternative of independent interest to computationally intensive index policies.

**Challenger's index** We fix an index that is not dependent on the round, but only on the history of the challenger and the leader available at this round, that we denote respectively by $\mathcal{X} = (X_1, \dots, X_n)$, $\mathcal{Y} = (Y_1, \dots, Y_N)$ for simplicity of notations. We denote by $\mu : \mathbb{R}^{\mathbb{N}} \to \mathbb{R}$ the function that computes the average of a set of observations. The duel can includes two steps, and the challenger wins if

1. $\mu(\mathcal{X}) \geq \mu(\mathcal{Y})$ (first compare the empirical means), or
2. $\widetilde{\mu}(\mathcal{X}, \mathcal{Y}) \geq \mu(\mathcal{Y})$, where $\widetilde{\mu} : \mathbb{R}^{\mathbb{N}} \times \mathbb{R}^{\mathbb{N}} \to \mathbb{R}$ denotes the chosen DS index.

We summarize in Algorithm 1 the steps of Dirichlet Sampling, that we completely detail in Appendix A.2. We write it for a generic "Dirichlet Sampling index" $\widetilde{\mu}$ that must be computed by a re-weighting of the observations augmented by an exploration bonus. As in NPTS, the weights are drawn with a Dirichlet distribution. For instance, a canonical example of Dirichlet Sampling index with a data-depend (instead of fixed) bonus $\mathcal{B}(\mathcal{X}, \mathcal{Y})$ is

$$\widetilde{\mu}(\mathcal{X}, \mathcal{Y}) = \sum_{i=1}^{n} w_i X_i + w_{n+1} \mathcal{B}(\mathcal{X}, \mathcal{Y}) \ , \ w = (w_1, \dots, w_{n+1}) \sim \mathcal{D}_{n+1} \ .$$

However, the algorithm structure in Algorithm 1 could be combined with any randomized index, which is of independent interest as we will see in Section 3. In the next section we study the theoretical properties of Dirichlet Sampling, and discuss the choice of the index $\widetilde{\mu}$ for different families of distributions.

---

**Algorithm 1** Generic Dirichlet Sampling

---

**Input:** $K$ arms, horizon $T$, Dirichlet Sampling index $\widetilde{\mu}$
**Init.:** $t = 1$, $r = 1$, $\forall k \in \{1, \dots, K\}$: $\mathcal{X}_k = \{X_1^k\}$, $N_k = 1$;        ▷ Draw each arm once
**while** $t < T$ **do**
    $\mathcal{A} = \{\}$ ;                        ▷ Arm(s) to pull at the end of the round
    $\ell = \text{Leader}((\mathcal{X}_1, N_1), \dots, (\mathcal{X}_k, N_k))$ ;                ▷ Choose a Leader
    **for** $k \in \{1, \dots, K\} : N_k < N_\ell$ **do**
        **if** $\max(\mu(\mathcal{X}_k), \widetilde{\mu}(\mathcal{X}_k, \mathcal{X}_\ell)) \geq \mu(\mathcal{X}_\ell)$ **then**
            $\mathcal{A} = \mathcal{A} \cup \{k\}$ ;                        ▷ Play the duels

Draw arms from $|\mathcal{A}|$ if $\mathcal{A}$ is non-empty, else draw arm $\ell$.
  Update $t, r, (N_k)_{k \in \{1,\dots,K\}}, (\mathcal{X}_k)_{k \in \{1,\dots,K\}}$. ;        ▷ Collect Reward(s) and update data

---

## 3 Regret Analysis and Technical Results

In this section, we analyze the regret of DS algorithms. We first derive a general regret decomposition for any index $\widetilde{\mu}$ that holds thanks to the duel-based structure. We then introduce several properties of Dirichlet sampling, that theoretically guide proper tuning of a DS index. We finally instantiate DS for three different problems and provide regret bounds in these settings. Starting with the regret decomposition, we exhibit general conditions to ensure guarantees that are independent on the index and the run of the bandit algorithm. Allowing a different family of distribution $\mathcal{F}_k$ for each arm $k$, the first one concerns the concentration of the mean of each distribution.

**Condition 1 (C1) [Concentration]** For all $\nu_k \in \mathcal{F}_k$, there exists a good rate function $I_k$ satisfying $I_k(x) > 0$ for $x \neq \mu_k$ and for all $x > \mu_k, y < \mu_k$, and any i.i.d sequence $X_1, \dots, X_n$ drawn from $\nu_k$

$$\mathbb{P}\left(\frac{1}{n}\sum_{i=1}^{n} X_i \geq x\right) \leq e^{-nI_k(x)} \text{ , and } \mathbb{P}\left(\frac{1}{n}\sum_{i=1}^{n} X_i \leq y\right) \leq e^{-nI_k(y)} \text{ .} \qquad (4)$$

This hypothesis is standard in the bandit literature, and is for instance satisfied by any *light-tailed* distributions. We refer to (Dembo and Zeitouni, 2010) for techniques to derive such functions.

We now provide an upper bound on the round-regret presented in Section 2 for Algorithm 1. To simplify the notations we consider that there is only one optimal arm and, without loss of generality, that $\forall k > 1, \mu_k < \mu_1$. Furthermore, for simplicity we write the following theorem for an index $\widetilde{\mu}(\mathcal{X}, \mu)$, that only uses the mean of the leader. The same result holds for any index using statistics on the leader's history that have concentration properties similar to (C1) (e.g possibly quantiles, variance, etc) with slight adaptations of the proof.

**Theorem 3.1** (Generic regret decomposition of DS). *Consider a bandit model $\nu = (\nu_1, \dots, \nu_K)$, where all distributions in $\nu$ satisfy (C1). Then for any DS index the expected number of pulls of each arm $k \in \{2, \dots, K\}$ is upper bounded for each $\varepsilon \in [0, \Delta_k)$ by*

$$\mathbb{E}\left[N_k(T)\right] \leq n_k(T) + B_{T,\varepsilon}^k + C_{\nu,\varepsilon}^k \text{ ,}$$

*where $n_k(T) = \mathbb{E}\left[\sum_{r=1}^{T-1} \mathbb{1}(k \in \mathcal{A}_{r+1}, \ell^r = 1)\right]$, $C_{\nu,\varepsilon}^k$ is independent on $T$ and, denoting $\mathcal{X}_n$ the set of $n$ first observations of arm 1,*

$$B_{T,\varepsilon}^k = \sum_{j=2}^{K} \sum_{n=1}^{\lceil 2\log(T)/I_1(\mu_k+\varepsilon)\rceil} \sup_{\mu\in[\mu_j-\varepsilon,\mu_j+\varepsilon]} \mathbb{E}_{\mathcal{X}_n}\left[\frac{\mathbb{1}(\mu(\mathcal{X}_n)\leq\mu)}{\mathbb{P}(\widetilde{\mu}(\mathcal{X}_n,\mu)\geq\mu)}\right] \text{ .}$$

The details of the proof of this result are to be found in Appendix B. The proof follows the general outline of Chan (2020), and makes all the components of $C_{\nu,\varepsilon}^k$ explicit. This term is related to deviations of sample means for arms $k$ and 1 and is typically bounded by a (problem-dependent) constant under light-tail concentration (C1), so it does not depend on $\widetilde{\mu}$ but only on the rate functions and the means of each arm. The other two terms of the RHS reflect the exploration strategy. $n_k(T)$ is the expected number of pulls of arm $k$ when the best arm is the leader; we interpret it as the sample size required to statistically separate both arms at horizon $T$. On the other hand, $B_{T,\varepsilon}^k$ measures the capacity of the best arm to recover from a bad (small-sized) sample.

Theorem 3.1 is formulated to be as general as possible and can be regarded as a counterpart of Theorem 1 of Kveton et al. (2019b). We will later analyze instances of Dirichlet Sampling where the first-order term of the regret is driven entirely by $n_k(T)$. We therefore introduce the following condition to control the contribution of $B_{T,\varepsilon}^k$ to the regret.

**Condition 2 (C2)** For any $\mu < \mu_1$, and any $n_1(T) = o(\log T)$ it holds that

$$\sum_{n=1}^{n_1(T)} \mathbb{E}_{\mathcal{X}_n\sim\nu_1^n}\left[\frac{\mathbb{1}(\mu(\mathcal{X}_n)\leq\mu)}{\mathbb{P}_{w\sim\mathcal{D}_{n+1}}(\widetilde{\mu}(\mathcal{X},\mu)\geq\mu)}\right] = o(\log T) \text{ .}$$

The LHS represents the expected cost in terms of regret of underestimating the optimal arm; intuitively, it measures the expected number of losing rounds before finally winning one when starting with low rewards. This is a classic decomposition in bandit analysis, and a counterpart of (C2) holds for most index policies with provable regret guarantees, e.g Theorem 1 in Kveton et al. (2019b) (GIRO) or Lemma 4 in Agrawal and Goyal (2012)) (Bernoulli Thompson Sampling). We find it noteworthy that this regret decomposition depends only on the distribution of the best arm and its randomized Dirichlet Sampling index when it is a challenger.

**Corollary 3.1.1** (Conditions for controlled regret). *If condition (C1) and (C2) holds for the DS index on the families of distribution $(\mathcal{F}_k)_{k\in\{1,\dots,K\}}$, the regret of the DS algorithm satisfies*

$$\mathcal{R}_T \leq \sum_{k=2}^{K} \Delta_k n_k(T) + o(\log T) \text{ .}$$

Up to this point this result is quite abstract, but this standardized analysis allows us to instantiate the Dirichlet Sampling algorithm on different class of problems and calibrate it in order to ensure condition (C2) holds and to make $n_k(T)$ explicit. In particular if $n_k(T) = \mathcal{O}(\log T)$, we recover the logarithmic regret. In the next section, we present technical results to justify calibrations of the DS index for several kind of families.

### 3.1 Technical tools: boundary crossing probability of a DS index

In this section, we highlight some key properties of a sum of random variables re-weighted by a Dirichlet weight vector that help us suggest a sound tuning of the bonus $\mathcal{B}(\mathcal{X}, \mathcal{Y})$ for different kind of families. We then detail such tuning.

**Boundary crossing probability (BCP)** We consider a set of $n+1$ observation points $\mathcal{X} = (X_1, \ldots, X_{n+1}) \subset \mathbb{R}^{n+1}$. (Intuitively, $n$ points are samples from a challenger arm, and one point corresponds to the added bonus). Then, for any $\mu \in \mathbb{R}$, we introduce the following "Boundary Crossing Probability" (BCP) term, conditional on $\mathcal{X}$

$$[\text{BCP}] := \mathbb{P}_{w \sim \mathcal{D}_{n+1}} \left( \sum_{i=1}^{n+1} w_i X_i \geq \mu \right) ,$$

where we recall that $\mathcal{D}_{n+1}$ is the Dirichlet distribution with parameter $(1, \ldots, 1)$ of size $n+1$, i.e the uniform distribution on the $(n+1)$-simplex. We emphasize that here $\mathcal{X}$ is considered fixed, and the only source of randomness comes from the weights $w$. When all observations are distinct this expression has a closed form, which is unfortunately untractable in the proof, as discussed in Appendix C.2. This quantity is of much interest as both the growth of $n_k(T)$ and (C2) can be derived from respectively upper and lower bounds for the BCP. Lemma 14 and Lemma 15 in (Riou and Honda, 2020) provide such bounds, resorting to classical concentration results and properties of the Dirichlet distribution that we recall in Appendix C.1 and C.2, and complete with additional technical results. The lower bounds suggest non-trivial tuning of the bonus. We first exhibit a necessary condition when the bonus is not allowed to depend on the set of observations $\mathcal{X}$.

**Lemma 3.2** (Necessary condition with a data-independent bonus). *Consider a fixed bonus* $B(\mathcal{X}, \mu) = B(\mu)$, *and a distribution* $F$ *(with CDF also denoted* $F$*). If Condition (C2) holds then*

$$B(\mu) > \mu + \frac{1}{1 - F(\mu)} \mathbb{E}_F \left[ (\mu - X)_+ \right] .$$

This result is obtained using a "worst-case" scenario when all observations are below the threshold $\mu$. Hence, it does not cover all possible trajectories, yet it suggests to investigate the properties of bonuses with a similar form. Since the right-hand side of the inequality requires a knowledge on the arms distributions that we would like to avoid, we use an empirical estimator for the expectation. This suggests to introduce some parameter $\rho$ and data-dependent bonuses of the form

$$B(\mathcal{X}, \mu, \rho) = \mu + \rho \times \frac{1}{n} \sum_{i=1}^{n} (\mu - X_i)^+ . \tag{5}$$

We interpret $\rho$ as the **leverage** of the empirical excess gap $\frac{1}{n} \sum_{i=1}^{n} (\mu - X_i)^+$ w.r.t the threshold $\mu$. We then tune $\rho$ assuming an hypothesis on some upper quantile of the arm distribution, which is much less constraining than assuming knowledge of the shape of the entire tail. In all DS algorithms we propose (see next section), we use Equation 5 as the basis for defining the appropriate bonus. Finally, we provide in Lemma 3.3 a novel lower bound on the BCP that reveals that in the general case of unbounded distributions, without further processing of the data, DS cannot achieve a logarithmic regret when the maximum of the data tends to $+\infty$ at some rate $g(n)$.

**Lemma 3.3** (Lower bound for the BCP). *Consider a set* $\mathcal{X} = (X_1, \ldots, X_{n+1}) \in \mathbb{R}^{n+1}$, *and assume that* $\overline{\mathcal{X}} = \max_{i \in \{1, \ldots, n+1\}} X_i \geq g(n)$ *for some function* $g$. *Denoting* $\bar{\Delta}_n^+ = \frac{1}{n} \sum_{i=1, X_i < \overline{\mathcal{X}}}^{n+1} (\mu - X_i)^+$ *the empirical positive gap, it holds that*

$$\mathbb{P}_{w \sim \mathcal{D}_{n+1}} \left( \sum_{i=1}^{n+1} w_i X_i \geq \mu \right) \geq \exp \left( -n \frac{\bar{\Delta}_n^+}{g(n) - \mu} \right) .$$

In particular, we see in this expression that $g(n)$ may hinder the exponential rate in $n$. In the next section we discuss three examples of DS algorithms and their theoretical guarantees.

## 3.2 Theoretical guarantees for Dirichlet Sampling algorithms

Building on the results from previous the section, we now instantiate the DS algorithms for three bandit problems. We first prove that optimal guarantees can be derived for DS with bounded distributions under a non-standard definition of the problem (i.e unknown upper bound but alternative assumptions), motivated by practical considerations. Then, we consider a natural extension to unbounded distributions using a simple truncation mechanism, ensuring logarithmic regret under assumptions on some quantile of the distributions. Finally we consider a simple DS algorithm, securing slightly larger-than-logarithmic regret for the entire family of *light-tailed distributions*. In the following we denote by $B(\mathcal{X}, \mu, \rho)$ the bonus defined in Equation 5 for a set $\mathcal{X}$, a mean $\mu$ and some parameter $\rho$. For simplicity we will keep a generic $\mu$ in our exposition, while its value is in practice the empirical mean of the leading arm. We detail each algorithm and their components in Appendix A.2 and the proofs of the three theorems in Appendix D. In all cases, the proof consists in showing that (C1) and (C2) hold for each proposed algorithms in the settings they tackle and deriving an expression for $n_k(T)$.

**Optimality for bounded distributions**   Let $\mathcal{F}_{[b,B]}$ be the set of distributions supported in $[b, B]$, and consider a bandit $\nu = (\nu_1, \ldots, \nu_K)$ with $\nu_k \sim \mathcal{F}_{[b_k, B_k]}$ for some $B_k \in \mathbb{R}$. If we assume that $B_k$ is known (case 1), then simply defining $B_k$ as the exploration bonus ensures an asymptotically optimal regret, with a direct adaptation of the proof of NPTS (Riou and Honda, 2020). However, the precise knowledge of the upper bound for each arm is sometimes inaccessible to the practitioner (e.g if the environment is new, or if no expert is available to provide an estimate of the bound). We propose an alternative setting, with the family $\mathcal{F}_{\mathcal{B}}^{\gamma,p} = \{\exists B : \nu \in \mathcal{F}_{[b,B]}, \mathbb{P}_\nu([B - \gamma, B]) \geq p\} \subset \mathcal{F}_{[b,B]}$. $B_k$ is *unknown* but we assume it is *detectable* in the sense that we will observe a sample from its neighborhood $[B_k - \gamma, B_k]$ with a reasonable probability of at least $p$, with known $\gamma, p$ (case 2). In this case we propose the following bonus, allowing to obtain theoretical results in this setting,

$$B(\mathcal{X}, \mu) := \max\{\bar{\mathcal{X}} + \gamma, B(\mathcal{X}, \mu, \rho)\}, \quad \text{where } \bar{\mathcal{X}} = \max\{x : x \in \mathcal{X}\}. \tag{6}$$

**Theorem 3.4** (Optimality of BDS). *If* $\forall k \in \{2, \ldots, K\}$, $\nu_k \sim \mathcal{F}_{\mathcal{B}}^{\gamma,\rho}$, *choosing the exploration bonus of Equation 6 with* $\rho \geq -1/\log(1-p)$ *ensures that*

$$\mathbb{E}[N_k(T)] \leq \frac{\log(T)}{\mathcal{K}_{\inf}^{B_{\rho,\gamma}}(\nu_k, \mu_1)} + O(1),$$

*where* $B_{\rho,\gamma} = \max(B + \gamma, \mu_1 + \rho\mathbb{E}_{\nu_k}[(\mu_1 - \mu_k)_+]))$.

This setting is a first example of the interest of data-dependent bonuses. It makes sense in practice by avoiding for instance distributions with a small mass arbitrarily far from the rest of their support, which may not be likely in a real-world application. We now consider the unbounded case.

**Unbounded distributions: truncating the upper tail**   Let consider the family $\mathcal{F}_{[b,+\infty]}$ for some unknown $b \in \mathbb{R}$. A natural way to extend algorithms designed for $\mathcal{F}_{[b,B]}$ (where $B < +\infty$) is to truncate the upper tail of the distributions. We propose a simple way to do this, by considering (as a parameter of the algorithm) a quantile $1 - \alpha$, denoted by $q_{1-\alpha}(\nu)$ for a distribution $\nu$, and a truncation operator $\mathcal{T}_\alpha$ that (1) do not change a distribution below its $1 - \alpha$ quantile, and (2) "summarizes" its upper tail by its expectation, known as *Conditional Value at Risk* (CVaR). Formally, we obtain $\mathcal{T}_\alpha(\nu)(A) = \nu(A)$ for any $A \subset [b, q_{1-\alpha}(\nu)]$ and $\mathcal{T}_\alpha(\nu)(\{x\}) = \alpha\mathbb{1}(x = C_\alpha(\nu))$ for any $x > q_{1-\alpha}(\nu)$, with $C_\alpha(\nu) = \mathbb{E}[X|X > q_{1-\alpha}(\nu)]$. We then propose *Quantile Dirichlet Sampling* (QDS), that computes the index of a challenger (say arm $k$, with observations $\mathcal{X}_k$) during a duel as follow: (1) apply $\mathcal{T}_\alpha$ to the empirical distribution, (2) compute the bonus $B(\mathcal{X}_k, \mu, \rho)$, and (3) *re-sample* the truncated empirical distribution with weights drawn according to $\text{Dir}(1, \ldots, 1, n_\alpha)$ where parameter $n_\alpha$ is for the weight used with the empirical CVaR, and is simply the number of observations used to compute it (to avoid a bias in the re-sampled mean). We can obtain theoretical guarantees with this method by considering the subset of distributions

$$\mathcal{F}_{[b,+\infty)}^{\alpha} = \{\nu \in \mathcal{F}_{[b,+\infty)} : \forall \mu > \mathbb{E}_\nu(X), \mathcal{K}_{\inf}^{\mathcal{F}_{[b,+\infty)}}(\nu, \mu) \geq \mathcal{K}_{\inf}^{\mathfrak{M}_k}(\mathcal{T}_\alpha(\nu), \mu)\},$$

where $\mathfrak{M}_k^q = \max\{q_{1-\alpha}(\nu_k), \mu_1 + \rho\mathbb{E}_{\nu_k}[(\mu_1 - X)^+]\}$, and the second $\mathcal{K}_{\inf}$ is taken on the family $\mathcal{F}_{[b,\mathfrak{M}_k^q]}$ (using previously introduced notations). Although technical, this condition essentially states

that the bandit problem taken on the complete family $\mathcal{F}_{[b,+\infty)}$ is no harder than an alternative bandit problem considering the truncated distributions and a bounded family, with an upper bound depending on the $1 - \alpha$ quantile and the leverage $\rho$ of the exploration bonus.

**Theorem 3.5** (Logarithmic Regret of QDS). *Consider a bandit model $\nu = (\nu_1, \ldots, \nu_K)$ satisfying $\forall k, \nu_k \in \mathcal{F}_{[b,+\infty)}^\alpha$ for some $b > -\infty$ (lower-bounded support) and a known $\alpha > 0$. Then, for any $\varepsilon_0 > 0$ small enough QDS with any parameters $\alpha' < \alpha$ and $\rho \geq (1 + \alpha')/\alpha'^2$ satisfies*

$$\mathbb{E}[N_k(T)] \leq \frac{\log T}{\mathcal{K}_{\inf}^{\mathfrak{M}_k^C}(\mathcal{T}_\alpha(\nu_k), \mu_1) - \varepsilon_0} + \mathcal{O}(1) \,,$$

*with $\mathfrak{M}_k^C = \max\{C_\alpha(\nu_k), \mu_1 + \rho \mathbb{E}_\nu[(\mu_1 - X)^+]\}$, and $\mathcal{T}_\alpha$ is the truncation operator we defined.*

This result is of particular interest as it captures the continuum between bounded and light-tailed distributions. In our opinion, it sheds new light on the interpretation of infeasability results of e.g Ashutosh et al. (2021): logarithmic regret can be achieved *without specifying the tail with precise parameters*, but a simple quantile condition is required to avoid pathological distributions that makes little sense in practice (e.g very small mass at a very large value). We further discuss this condition in Appendix E and provide examples of families for which it holds (exponential, Gaussian).

**Remark 3.6.** *The restriction to the semi-bounded case $b > -\infty$ is due to our proof technique, based on a discretization of the support of the truncated distribution (see Appendix D). Note that the actual value of $b$ is not known by the algorithm. This is intuitive since $\mathcal{K}_{\inf}^{\mathcal{F}_{-\infty,B}} = \mathcal{K}_{\inf}^{\mathcal{F}_{b,B}}$ for all $b, B \in \mathbb{R}$, as proved in Theorem 2 of (Honda and Takemura, 2015). Different theoretical tools could allow to prove a logarithmic regret for QDS in the doubly unbounded case, possibly with a symmetric treatment of the two tails. We leave this extension for future work.*

One may wonder whether the couple quantile condition/truncation is necessary to achieve theoretical results as well as good practical performance. Our last algorithm investigates this issue.

**Robust regret for light-tailed distributions** We call *Robust Dirichlet Sampling* (RDS) the algorithm with bonus $B(\mathcal{X}, \mu, \rho_n)$, where the leverage $\rho_n$ is a function of the sample size $n = |\mathcal{X}|$. We prove that while being very simple, RDS achieves a robust sub-linear regret bound when each arm comes from **any** *unknown* light-tailed distribution, that we define as the family

$$\mathcal{F}_\ell = \{\nu \in \mathcal{F}_{(-\infty,+\infty)} : \exists \lambda_\nu > 0, \forall \lambda \in [-\lambda_\nu, \lambda_\nu], \mathbb{E}_\nu[\exp(\lambda X)] < +\infty\} \,.$$

**Theorem 3.7** (Robust regret bound for RDS). *Let $\nu = (\nu_1, \ldots, \nu_K)$ a bandit model satisfying $\nu_k \in \mathcal{F}_\ell$ for all $k$. Consider **any** increasing sequence $(\rho_n)_{n \in \mathbb{N}}$ with $\rho_n \to +\infty$, $\rho_n = o(n)$. Then, for $T$ large enough the expected number of pull of any sub-optimal arm $k$ in RDS is upper bounded by*

$$\mathbb{E}[N_k(T)] \leq n_k^{\eta,\varepsilon_0}(T) + \mathcal{O}(1) \,,$$

*where for any $\eta \in (0, 1], \varepsilon_0 > 0, n_k^{\eta,\varepsilon_0}(T)$ is the sequence satisfying*

$$n_k^{\eta,\varepsilon_0}(T) = \frac{\log T}{\eta(\Delta_k - \varepsilon_0)}(M_{k,n_k^{\eta,\varepsilon_0}(T)} - \mu) \,, \text{ with } M_{k,n} = \max\left\{F_k^{-1}\left(\exp\left(-\frac{1}{n^2(\log n)^2}\right)\right), \rho_n\right\}.$$

*In particular, if $\rho_n = \mathcal{O}(\log n)$ then $\mathbb{E}[N_k(T)] = \mathcal{O}(\log(T) \log\log(T))$ for any light-tailed distribution $\nu_k \in \mathcal{F}_\ell$.*

The sequence $M_{k,n}$ is a large probability upper bound of the maximum of $n$ observations from $F_k$, that we discuss in Appendix D. For light-tailed distributions, it holds that $M_{k,n} = \mathcal{O}(\log n)$ (using Jensen inequality as in the proof of Theorem 2.5 in Boucheron et al. (2013)). Hence, choosing $\rho_n = \mathcal{O}(\log n)$ we can further obtain the simpler upper bound in $\mathcal{O}(\log(T) \log\log(T))$. This slightly larger-than-logarithmic rate is a consequence of Lemma 3.3. In our opinion this is a small cost compared to the adaptive power of RDS. We call the algorithm *robust* because these theoretical guarantees are obtained on the broad class of light-tailed distributions, without any additional assumption. We recommend the leverage function $\rho_n = \mathcal{O}(\sqrt{\log(1+n)})$, which corresponds to the growth rate of the maximum of sub-Gaussian samples and is empirically validated (see Appendix F). We emphasize that RDS thus avoids all hyperparameter tuning, a desirable feature for the practitioner with little information on the problem she faces. Furthermore, in the next section we show that this algorithm performs very well in practice despite its non-logarithmic asymptotic guarantees.

# 4 Application in a crop-farming environment

We consider a practical decision-making problem using the DSSAT[2] simulator (Hoogenboom et al., 2019). Harnessing more than 30 years of expert knowledge, this simulator is calibrated on historical field data (soil measurements, genetics, planting date...) and generates realistic crop yields. Such simulations are used to explore crop management policies *in silico* before implementing them in the real world, where their actual effect may take months or years to manifest themselves. More specifically, we model the problem of selecting a planting date for maize grains among 7 possible options, all else being equal, as a 7-armed bandit. The resulting distributions incorporate historical variability as well as exogenous randomness coming from a stochastic meteorologic model. We illustrate this in Figure 1 with the histogram of four of these distributions, computed on $10^6$ samples. They are typically right-skewed, multimodal and exhibit a peak at zero corresponding to years of poor harvest, hence they hardly fit to a convenient parametric model (e.g SPEF/sub-Gaussian...).

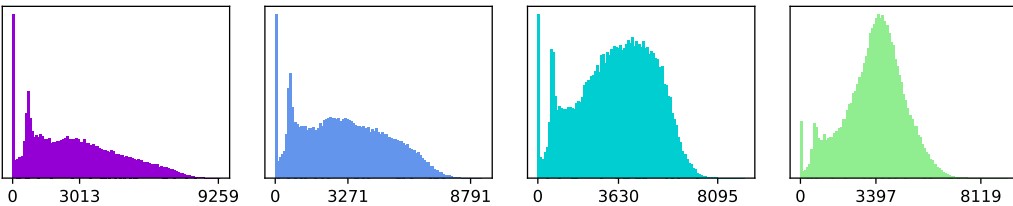

Figure 1: Distribution of simulated dry grain yield (kg/ha) for four out of seven different planting dates. Reported on the x-axis are the distribution minimum, mean and maximum values. The optimal arm is the third one (mean 3630 kg/ha).

**Benchmarks**  A natural choice for the learner would be to use algorithms adapted for bounded distributions with known support. Indeed, one could argue that crop yields are fundamentally bounded by a very large value, that can be provided with some expert knowledge. However this method may have limits when the upper bound cannot be estimated accurately (few data, new environment, ...), as a conservative bound can have a cost on the regret. For this reason, we believe that the novel Dirichlet Sampling algorithms we introduce in Section 3.2 are a good alternative choice for this problem. In particular, the three algorithms we propose in this paper are relevant in this setting: BDS keeps the bounded-support hypothesis but introduces the possible uncertainty on the bound, while the light-tailed hypothesis of RDS and the quantile condition of QDS look reasonable. In Figure 2 we compare DS algorithms to empirical IMED (Honda and Takemura, 2015) and NPTS (Riou and Honda, 2020), with two upper bounds: 1) the "exact" upper bound is provided looking at the maximum of all historical data collected (left figure), and 2) the algorithms use a conservative estimate with a value 1.5 times larger than the previous one (right figure). To avoid cluttering, we only report the performance of IMED and NPTS as they were the most competitive baselines on this problem, but report figures with other competitors (e.g UCB1, Bernoulli TS, SDA) in Appendix F.

**Tuning**  For BDS we choose the parameters $\rho = 4, \gamma = 3500$, corresponding to $p \approx 20\%$ in the hypothesis of Theorem 3.4, which is conservative in our example. For QDS, we set $\rho = 4$ to be able to compare with BDS and a quantile $95\%$. Finally for RDS, we choose $\rho_n = \sqrt{\log(1+n)}$, which enters into the theoretical framework of Theorem 3.7.

**Results**  Our results show that Dirichlet Sampling algorithms achieve similar or slightly lower regret to their competitors when the latter are allowed to use the "exact" upper bound, and compare favorably when they use a conservative estimate (1.5 times larger, right), see Figure 2. In particular, RDS is the overall winner in both experiments. We think this demonstrates the merits of trading-off logarithmic regret (albeit only by a factor $\mathcal{O}(\log\log T)$) for finite-time adaptation to the tail behaviour via the leverage $\rho_n$. As a side remark, note that our round-based implementation is more efficient than NPTS as it does not draw random weights for the leader, which is the most costly operation at each round. The code to reproduce the experiments is available in this github repository.

---

[2]*Decision Support System for Agrotechnology Transfer* is an open-source project maintained by the DSSAT Foundation, see https://dssat.net/

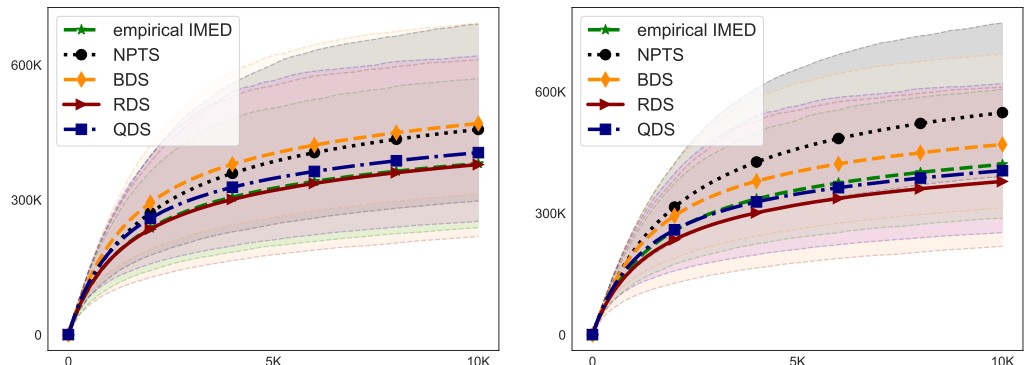

Figure 2: Average regret on 5000 simulations and horizon $T = 10^4$. Dashed lines correspond to 5%-95% regret quantiles. Empirical IMED and NPTS are run with exact upper bounds around $1.5 \times 10^4$ kg/ha (left) and the conservative upper bound $1.5 \times 10^4$ kg/ha (right).

**Other experiments**   To further illustrate the properties of DS algorithms, we perform additional experiments on synthetic examples. Due to space limits, we present our results in Appendix F. First, we test the *sensitivity* of DS w.r.t its hyperparameters, and check that their impact on the performance of the algorithms is moderate. Then, we show the merits of RDS in case of *model misspecification*, following the robustness experiments of Ashutosh et al. (2021). Finally, we consider the case of *Gaussian mixtures*, a common tool to model nonparametric distributions via *kernel density estimation*, and show that they fit the scope of DS but not that of usual bandit algorithms.

## 5   Conclusion

In this paper, we introduced a new framework for randomized exploration in stochastic bandits based on resampling of the reward history and a data-dependent bonus, which generalizes an optimal Thompson Sampling strategy for bounded distributions to *light-tailed* families. We proposed three instances of such Dirichlet Sampling (DS) algorithms, corresponding to different modeling assumptions. In our opinion, these new algorithms are appealing for the practitioner because 1) our theoretical results show strong guarantees under different settings, 2) DS algorithms are simple to implement despite the technically challenging analysis and achieve strong practical performances, and 3) they provide alternative robust ways to tackle unbounded distributions in bandit problems. Interesting future directions include extending the DS framework to *heavy tail* distributions, and tightening the analysis of *Boundary Crossing Probabilities* of Section 3.1 to design sharper bonuses for general families of distributions motivated by real use-cases. Moreover, we believe the duel-based structure associated with the generic regret decomposition of Theorem 3.1 opens up new perspectives to design exploration strategies in bandits. In particular, they allow to analyze policies using the history of two arms in the computation of a single index.

## Acknowledgments and Disclosure of Funding

The PhD of Dorian Baudry and Patrick Saux are respectively funded by a CNRS80 grant and the Université de Lille Nord Europe's I-SITE EXPAND, as part of the Bandits For Health (B4H) project. This work has been supported by the French Ministry of Higher Education and Research, Inria, Scool, and the French Agence Nationale de la Recherche (ANR) under grant ANR-16-CE40-0002 (the BADASS project).

We thank the anonymous reviewers for their careful reading of the paper and their suggestions for improvements. We also warmly thank Emilie Kaufmann who managed to carefully read the paper and make (as always) very useful comments while taking care of little Pascal, and Romain Gautron for his precious help for the experiments involving the DSSAT simulator. Experiments presented in this paper were carried out using the Grid'5000 testbed, supported by a scientific interest group hosted by Inria and including CNRS, RENATER and several Universities as well as other organizations (see https://www.grid5000.fr).

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
