# Contents

# A   Supplementary Material

In this section we introduce different elements to help the understanding of the Dirichlet Sampling algorithms. We first detail the notations used in the main text and in the proofs. Then, we provide a detailed version of Algorithm 1 and recall the three DS indexes presented in Section 3.2. Finally, we briefly recall the Non Parametric Thompson Sampling of Riou and Honda (2020), to help readers that are not familiar with this algorithm.

## A.1   Notations

In this section, we provide to the reader an index of all the notations used in Sections 1-4, and in the detailed proofs in Appendix B-D.

**Multi-Arm Bandits and families of distributions**

- $\nu = (\nu_1, \ldots, \nu_K)$ denotes a $K$-armed bandits where an $k \in \{1, \ldots, K\}$ is associated with a reward distribution $\nu_k \in \mathcal{F}_k$, for some family of distributions $\mathcal{F}_k$.

- For any family of distributions $\mathcal{F}$, any $\nu \in \mathcal{F}$ and any $\mu \in \mathbb{R}$ we denote
$$\mathcal{K}_{\inf}^{\mathcal{F}}(\nu, \mu) = \inf_{\nu' \in \mathcal{F}, \nu' \neq \nu} \{\mathrm{KL}(\nu, \nu') : \mathbb{E}_{\nu'}(X) > \mu\} \ .$$
This quantity allows to define *asymptotic optimality* for bandit algorithms in Equation 2.

- SPEF: Single-Parameter Exponential Family. The family of distributions $\mathcal{P}_\Theta$ is a SPEF on the parameter set $\Theta \subset \mathbb{R}$ if there exists a function $\Psi : \Theta \to \mathbb{R}$ and $f : \mathbb{R} \to \mathbb{R}$ such that any distribution from $\mathcal{P}_\Theta$ admits a density
$$g_\theta(x) = g(x, \theta) = e^{\theta x - \psi(\theta)} f(x) \ ,$$
for some parameter $\theta \in \Theta$. Hence, each distribution in a specified SPEF is fully characterized by its parameter $\theta \in \Theta$.

- $\mathcal{F}_{[b,B]}$ denotes the set of all distributions supported on $[b, B]$, where $b$ and $B$ can respectively take values $-\infty$ and $+\infty$. With a slight abuse of notations, we denote for any $\nu \in \mathcal{F}_{[b,B]}$, $\mu \in \mathbb{R}$
$$\mathcal{K}_{\inf}^B(\nu, \mu) := \mathcal{K}_{\inf}^{\mathcal{F}_{[b,B]}}(\nu, \mu) \ ,$$
which holds for any $b$ as detailed for instance in (Honda and Takemura, 2015).

- $\mathcal{F}_{[b,B]}^{\gamma, p} = \{\nu \in \mathcal{F}_{[b,B]} : \mathbb{P}_\nu([B - \gamma, B]) \geq p\}$ is a set of bounded distributions with an additional condition on a neighborhood of their upper bound. Considering this set allows to build strategies with logarithmic regret for bounded distributions with an *unknown* upper bound.

- We denote the set of *light-tailed distributions* in $\mathbb{R}$
$$\mathcal{F}_\ell = \{\nu \in \mathcal{F}_{(-\infty, +\infty)} : \exists \lambda_\nu > 0, \forall \lambda \in [-\lambda_\nu, \lambda_\nu], \mathbb{E}_\nu[\exp(\lambda X)] < +\infty\} \ .$$

- We denote $q_\beta : \mathcal{F}_{(-\infty, +\infty)} \to \mathbb{R}$ the operator that returns the quantile $\beta \in [0, 1]$ of any distribution $\nu \in \mathcal{F}_{(-\infty, +\infty)}$,
$$q_\beta(\nu) = \inf\{x \in \mathbb{R} : F_\nu(x) > \beta\} \ ,$$
where $F_\nu$ denotes the cdf of a distribution $\nu$.

- We denote $C_\alpha : \mathcal{F}_{(-\infty, +\infty)} \to \mathbb{R}$ the operator that returns the Conditional Value-at-Risk (CVaR) at level $\alpha \in [0, 1]$ of any distribution $\nu \in \mathcal{F}_{(-\infty, +\infty)}$,
$$C_\alpha(\nu) = \inf_{x \in \mathbb{R}} \left\{ x + \frac{1}{\alpha} \mathbb{E}_\nu \left[ (X - x)_+ \right] \right\} \ .$$
Moreover if the cdf of $\nu$ is continuous, then $C_\alpha = \mathbb{E}_\nu [X | X \geq q_{1-\alpha}(\nu)]$.

- Considering a base family $\mathcal{F} \subset \mathcal{F}_{[b,+\infty)}$ and parameters $\alpha \in (0, 1)$, $\rho > 0$, we consider the subset of $\mathcal{F}$
$$\mathcal{F}_{[b,+\infty)}^\alpha = \{\nu \in \mathcal{F} : \forall \mu > \mathbb{E}_\nu(X), \mathcal{K}_{\inf}^{\mathcal{F}}(\nu, \mu) \geq \mathcal{K}_{\inf}^{\mathfrak{M}_k^q}(\mathcal{T}_\alpha(\nu), \mu)\} \ ,$$
where $\mathfrak{M}_k^q = \max\{q_{1-\alpha}(\nu), \mu + \rho \mathbb{E}_\nu[(\mu - X)^+]\}$, and $\mathcal{T}_\alpha : \mathcal{F} \to \mathcal{F}_{[-\infty, q_{1-\alpha}(\nu_k)]}$ is the truncation operator satisfying $\mathcal{T}_\alpha(\nu)(A) = \nu(A)$ for any $A \subset [b, q_{1-\alpha}(\nu)]$ and $\mathcal{T}_\alpha(\nu)(\{x\}) = \alpha \mathbb{1}(x = C_\alpha(\nu))$ for any $x > q_{1-\alpha}(\nu)$.

**Dirichlet Sampling Algorithms**

- DS := Dirichlet Sampling, and we call "DS index" the index $\widetilde{\mu}$ computed in DS algorithms using a *re-weighting* scheme (with a Dirichlet distributions) and an *exploration bonus*.

- *Bounded Dirichlet Sampling* (BDS): the DS algorithm proposed for families of distributions from $\mathcal{F}_{[b,B]}$ with known $B$ and $\mathcal{F}_{[b,B]^{\gamma,\rho}}$ for known $(\gamma, \rho)$.

- *Robust Dirichlet Sampling* (RDS): a DS algorithm proposed for the general family of light-tailed distributions $\mathcal{F}_\ell$.

- *Quantile Dirichlet Sampling* (QDS): a DS algorithm proposed for families of distributions $\mathcal{F}^\alpha_{[b,+\infty)}$ associated with some base family $\mathcal{F} \subset \mathcal{F}_{[b,+\infty]}$.

- *Round*: a step of the algorithm indexed by some $r \in \mathbb{N}$ and at the end of which a set of arms $\mathcal{A}_r \subset \{1, \ldots, K\}$ is selected.

- *T-Round regret*: Consider a bandit $\nu = (\nu_1, \ldots, \nu_K)$ with arms' means $(\mu_1, \ldots, \mu_K)$ and an horizon of $T$ rounds,

$$\mathcal{R}_T = \sum_{k=1, \Delta_k > 0}^{K} \Delta_k \mathbb{E}[N_k(T)] \, ,$$

  where $\Delta_k = \max_{i \in \{1, \ldots, K\}} \mu_i - \mu_k$ and

$$N_k(T) = \sum_{r=1}^{T} \mathbb{1}(k \in \mathcal{A}_r)$$

  is the number of selection of an arm $k$ after $T$ rounds.

- *Duel*: a pairwise comparisons between two arms at the end of which one arm is selected as the *winner*.

- *Leader*: A reference arm $\ell^r$ chosen at the beginning of each round $r$ as

$$\ell^r \in \operatorname*{argmax}_{k \in \{1, \ldots, K\}} N_k(r) \, .$$

  We denote $\mathcal{L}^r = \operatorname*{argmax}_{k \in \{1, \ldots, K\}} N_k(r)$ the set of possible candidates for leadership. If $|\mathcal{L}^r| > 1$ the algorithm chooses uniformly at random an arm $\ell^r \in \operatorname*{argmax}_{k \in \mathcal{L}^r} \mu_k^r$, where $\mu_k^r$ is the empirical mean of arm $k$ at round $r$.

- *Duel in DS algorithms*: we denote $\mathcal{X} = (X_1, \ldots, X_n)$ and $\mathcal{Y} = (Y_1, \ldots, Y_N)$ two sets of real observations.

  1. $\mu : \mathbb{R}^\mathbb{N} \to \mathbb{R}$ denotes the application computing the mean of a set: $\mu(\mathcal{X}) = \frac{1}{n} \sum_{i=1}^{n} X_i$, and $\mu(\mathcal{Y}) = \frac{1}{N} \sum_{i=1}^{N} Y_i$.
  2. $\widetilde{\mu} : \mathbb{R}^\mathbb{N} \times \mathbb{R}^\mathbb{N} \to \mathbb{R}$ denotes an application that returns the "DS index" $\widetilde{\mu}(\mathcal{X}, \mathcal{Y})$.

  If $\mathcal{X}$ denotes the observations from a *challenger* in a DS algorithm, and $\mathcal{Y}$ the one of the current leader, then the challengers *wins the duel* if $\max(\mu(\mathcal{X}), \widetilde{\mu}(\mathcal{X}, \mathcal{Y})) \geq \mu(\mathcal{Y})$. Otherwise, the leader wins the duel.

- We denote $\mathrm{Dir}(P)$ the Dirichlet distribution with parameter $P \in \mathbb{N}^\mathbb{N}$. When $P = 1_n = (1, \ldots, 1)$ (vector of size $n$, all values equal to 1) we write this distribution $\mathcal{D}_n$. Furthermore, $\mathcal{D}_n$ is the uniform distribution on the probabilistic simplex of size $n$,

$$\mathcal{P}^n = \{w \in [0, 1]^n : \sum_{i=1}^{n} w_i = 1\} \, .$$

- We denote $\mathcal{B} : \mathcal{X} \in \mathbb{R}^\mathbb{N} \times \mathcal{Y} \in \mathbb{R}^\mathbb{N} \to \mathcal{B}(\mathcal{X}, \mathcal{Y})$ a generic *exploration bonus* for a duel involving the set $\mathcal{X}$ (challenger) and the set $\mathcal{Y}$ leader. With a slight abuse of notation we use $\mathcal{B}(\mathcal{X}, \mu(\mathcal{Y}))$ when it only uses the mean of the leader, and consider in the paper the bonus

$$B(\mathcal{X}, \mu, \rho) = \mu + \rho \times \frac{1}{n} \sum_{i=1}^{n} (\mu - X_i)_+ \, ,$$

  where $n = |\mathcal{X}|$, $\mathcal{X} = (X_1, \ldots, X_n)$ and $\rho > 0$ is a given parameter and for any $x \in \mathbb{R}$, $x_+ = \max(x, 0)$.

**Technical Notations for the proof of Theorem 3.1**

- Optimal arm = arm 1 without loss of generality
- $\mathcal{X}_n^k = (X_1^k, X_2^k, \ldots, X_n^k)$ denotes the first $n$ observations collected from arm $k$. We assume that for each arm an unknown infinite reward stream $\mathcal{X}^k$ is available and that the $j$-th reward in $\mathcal{X}^k$ is the $j$-th reward collected from this arm (independently on when it is observed). For arm 1 we simply use the notations $\mathcal{X}$ and $\mathcal{X}_n$.
- $a_r = \lceil r/4 \rceil$, $b_r = \lceil a_r/K \rceil$: in all rounds after $a_r$ the leader has at least $b_r$ observations, which is linear in $r$.
- $\mathcal{D}^r = \{\exists u \in [a_r, r] : \ell^u = 1\}$: under this event arm 1 has been leader at least once after the round $a_r$.
- We call the event $\mathcal{B}_k^r = \{\ell^r = 1, \ell^{r+1} = k\}$ a *leadership takeover* by arm $k$.
- $\mathcal{C}^r = \{\ell^r \neq 1, 1 \notin \mathcal{A}_{r+1}\}$ represents a duel lost at round $r$ by arm 1 against a sub-optimal leader. We also denote $L^r = \sum_{u=a_r}^{r} \mathbb{1}(\mathcal{C}^u)$.
- $\mathcal{H}_j^{r,n} = \left\{ N_1(r) = n, \left| \mu\left(\mathcal{X}_{N_j(r)}^j\right) - \mu_j \right| \leq \varepsilon, 1 \notin \mathcal{A}_{r+1} \right\}$ for some $\varepsilon > 0$.

## A.2 Detailed algorithms

---
**Algorithm 2** Generic Dirichlet Sampling Bandit Algorithm

---
**Input:** $K$ arms, horizon $T$, DS index $\widetilde{\mu}$
**Init.:** $t = 1, r = 1, \forall k \in \{1, ..., K\}$: $\mathcal{X}_k = \{\}, N_k = 1, S_k = 0$
**while** $t < T$ **do**

    $\mathcal{A} = \{\}$;                   ▷ Set of Arms to pull at the end of the round
    **if** $r = 1$ **then**
        $\mathcal{A} = \{1, \ldots, K\}$;          ▷ All arms are pulled at the first round
    **else**
        // Leader Choice
        $\mathcal{L} = \underset{k \in \{1,...,K\}}{\operatorname{argmax}} N_k$;     ▷ Arm(s) with the largest number of samples
        $\bar{\mathcal{L}} = \underset{k \in \mathcal{L}}{\operatorname{argmax}} \mu_k$;             ▷ Keep best arm(s) in $\mathcal{L}$
        $\ell \sim \mathcal{U}(\bar{\mathcal{L}})$;           ▷ Random choice if several candidates
        // Duels
        **for** $k \in \{1, \ldots, K\}, k \notin \mathcal{L}$;       ▷ Challengers in $\mathcal{L}$ are eliminated
        **do**
            **if** $\mu_k \geq \mu_\ell$ **then**
                $\mathcal{A} = \mathcal{A} \cup \{k\}$;          ▷ First duel with empirical mean
            **else**
                Draw DS index $\widetilde{\mu}(\mathcal{X}_k, \mathcal{X}_\ell)$
                **if** $\widetilde{\mu}(\mathcal{X}_k, \mathcal{X}_\ell) \geq \mu_\ell$ **then**
                    $\mathcal{A} = \mathcal{A} \cup \{k\}$;   ▷ Second duel with the Dirichlet Sampling index

    // Collect reward and update quantities
    **if** $|\mathcal{A}| = 0$ **then**
        $\mathcal{A} = \{\ell\}$;               ▷ If no winning challenger $\ell$ is pulled
    Shuffle $\mathcal{A}$
    **for** $k \in \mathcal{A}$ **do**
        **if** $t < T$ **then**
            Collect $X_{k,N_k}$, update history $\mathcal{X}_k = \mathcal{X}_k \cup \{X_{k,N_k}\}$
            $N_k = N_k + 1, S_k = S_k + X_{k,N_k}, \mu_k = S_k/N_k, t = t + 1$
    $r = r + 1$

---

Table 1: Summary of settings and bonuses

| Algorithm | Kind of family | $\mathcal{B}(\mathcal{X}, \mathcal{Y})$ |
|-----------|----------------|------------------------------------------|
| BDS | $\mathcal{F}_{[b,B]}$, known $B$ | $B$ |
| BDS | $\mathcal{F}_{[b,B]}^{\gamma,p}$, known $\gamma, p$ | $\max\{\bar{\mathcal{X}} + \gamma, B(\mathcal{X}, \mu, \rho)\}$ |
| RDS | $\mathcal{F}_\ell$ | $B(\mathcal{X}, \mu, \rho_n)$ with $\rho_n \to +\infty$ |
| QDS | $\mathcal{F}_{[b,+\infty)}^{\alpha}$ | $B(\mathcal{X}, \mu, \rho)$ |

In this section we provide the detailed implementation of each algorithm presented in this paper. Before that, we first recall the families on which each algorithm achieves regret guarantees and their corresponding bonuses. We denote $\bar{\mathcal{X}}$ the maximum of a set $\mathcal{X}$.

For QDS, the family $\mathcal{F}_{[b,+\infty)}^{\alpha}$ relies on an unknown base family $\mathcal{F} \subset \mathcal{F}_{[b,+\infty)}$ for some $b > -\infty$. We detail in Theorems 3.4, 3.7 and 3.5 the conditions on the tuning of $\rho$ for each algorithm to ensure a controlled regret. These three results rely on the same general analysis of a Dirichlet Sampling Algorithm with any index $\widetilde{\mu}$. Before instantiating each algorithm we first provide a detailed version of this algorithm in Algorithm 2, to complete the short version we provided in Section 2 with Algorithm 1,

We now provide the detailed computation of the DS index for the three algorithms we introduced in Section 3.2: Bounded Dirichlet Sampling (BDS), Robust Dirichlet Sampling (RDS) and Quantile Dirichlet Sampling (QDS).

**Bounded Dirichlet Sampling index**    We first introduce in Algorithm 3 the index of BDS. The set of parameters is dependent of the choice of hypothesis (B1) or (B2) in Theorem 3.4. As hypothesis (B1) corresponds to the same index as Non Parametric Thompson Sampling (that we describe in further details in Appendix A.3) we only report the bonus under case (B2).

---

**Algorithm 3** Bounded Dirichlet Sampling Index

**Input:** Two sets of data $\mathcal{X} = (X_1, \ldots, X_n)$ and $\mathcal{Y} = (Y_1, \ldots, Y_N)$, parameters $\gamma, \rho$

Set $\mu = \frac{1}{N} \sum_{i=1}^{N} Y_1$

Set $B(\mathcal{X}, \mu) = \max\left(\max_{i=1}^{n} X_i + \gamma, \mu + \frac{\rho}{n} \sum_{i=1}^{n}(\mu - X_i)^+\right)$

Draw $w = (w_1, \ldots, w_{n+1}) \sim \mathcal{D}_{n+1}$

**return** $\sum_{i=1}^{n} w_i X_i + w_{n+1} B(\mathcal{X}, \mu)$

---

We recall that if $\gamma$ and $p$ are known in (B2) then the theoretical tuning of $\rho$ we provide in Theorem 3.4 is $\rho \geq -1/\log(1-p)$.

**Robust Dirichlet Sampling index**    We now report in Algorithm 4 the index of RDS, which only depends on the choice of an increasing sequence $\rho_n$ satisfying $\rho_n \to +\infty$ and $\rho_n = o(\sqrt{n})$.

---

**Algorithm 4** Robust Dirichlet Sampling Index

**Input:** Two sets of data $\mathcal{X} = (X_1, \ldots, X_n)$ and $\mathcal{Y} = (Y_1, \ldots, Y_N)$, $(\rho_n)_{n \in \mathbb{N}}$

Set $\mu = \frac{1}{N} \sum_{i=1}^{N} Y_1$

Set $B(\mathcal{X}, \mu) = \mu + \rho_n \times \frac{1}{n} \sum_{i=1}^{n}(\mu - X_i)^+$

Draw $w = (w_1, \ldots, w_{n+1}) \sim \mathcal{D}_{n+1}$

**return** $\sum_{i=1}^{n} w_i X_i + w_{n+1} B(\mathcal{X}, \mu)$

---

RDS is the simplest instance of Dirichlet Sampling we propose, and achieves robust regret guarantees for light-tailed distributions. We also propose QDS for unbounded distribution, that achieves logarithmic regret guarantees under a mild quantile condition.

**Quantile Dirichlet Sampling index** We finally present in Algorithm 5 the index of QDS, depending this time on the choice of a quantile $q_{\alpha'}$ and a parameter $\rho$. If the regret guarantees require quite complicated notations for a proper formalism, the index is actually quite simple to implement: the bonus is similar to BDS and RDS, and the re-weighting step only require to compute the $\alpha'$ quantile of the data, the corresponding CVaR (mean of the observations larger than this quantile) and to draw a Dirichlet weight with a slightly different parameter. Furthermore, the computation time of these steps can be optimized in practice (keeping in memory the sorted data, quantile and CVaR).

---

**Algorithm 5** Quantile Adaptive Dirichlet Sampling Index

---

**Input:** Two sets of data $\mathcal{X} = (X_1, \ldots, X_n)$ and $\mathcal{Y} = (Y_1, \ldots, Y_N)$, quantile $\alpha'$, $\rho$
  Sort $\mathcal{X}$ to have $X_1 \leq X_2 \leq \cdots \leq X_n$

Set $\mu = \frac{1}{N} \sum_{i=1}^{N} Y_1$

Set $B(\mathcal{X}, \mu) = \mu + \frac{\rho}{n} \sum_{i=1}^{n} (\mu - X_i)^+$

Set quantile index $n_{\alpha'} = \lceil n(1 - \alpha') \rceil / n$

Set $C_{\alpha'} = \frac{1}{n - n_{\alpha'} + 1} \sum_{i=1}^{n - n_{\alpha'} + 1} X_{n_{\alpha'} + i}$

Draw $w = (w_1, \ldots, w_{n_{\alpha'} + 1}) \sim \text{Dir}((1, \ldots, 1, \mathbf{n}_{\alpha'}, 1))$ ;      $\triangleright$ `Only ones except for` $w_{n_{\alpha'}}$
**return** $\sum_{i=1}^{n_{\alpha'} - 1} w_i X_i + w_{n_{\alpha'}} C_{\alpha'} + w_{n_{\alpha'} + 1} B(\mathcal{X}, \mu)$

---

### A.3   Non-Parametric Thompson Sampling

In this section we briefly introduce the Non Parametric Thompson Sampling Algorithm, introduced in Riou and Honda (2020), for readers that are not familiar with it. NPTS is an index policy, that we detail in Algorithm 6. To keep consistent notations, we write it using the notations of this paper.

---

**Algorithm 6** Non Parametric Thompson Sampling (Riou and Honda, 2020)

---

**Input:** Horizon $T$, $K$ arms, known upper bound $B$
**Init.:** $\forall k \in \{1, ..., K\}$: $\mathcal{X}_k = \{\}$, $N_k = 0$
**for** $t \in \{1, \ldots, T\}$ **do**
    **for** $k \in \{1, \ldots, K\}$ **do**
        Sample $w \sim \mathcal{D}_{N_k + 1}$
        Set $\mu_k = \sum_{i=1}^{N_k} w_i X_i + w_{N_k + 1} B$
    Pull arm $A_t = \underset{k \in \{1, ..., K\}}{\text{argmax}} \mu_k$, observe $X_{N_k + 1}$
    Update $\mathcal{X}_k = \mathcal{X}_k \cup \{X_{n_k + 1}\}$, $N_k = N_k + 1$

---

We recall that this algorithm is asymptotically optimal when arms belong to $\mathcal{F}_{[b,B]}$ for a known $B$.

## B   Proof of Theorem 3.1

We first recall the statement of result we want to prove.

**Theorem 3.1** (Generic regret decomposition of DS). *Consider a bandit model $\nu = (\nu_1, \ldots, \nu_K)$, where all distributions in $\nu$ satisfy (C1). Then for any DS index the expected number of pulls of each arm $k \in \{2, \ldots, K\}$ is upper bounded for each $\varepsilon \in [0, \Delta_k)$ by*

$$\mathbb{E}\left[N_k(T)\right] \leq n_k(T) + B_{T,\varepsilon}^k + C_{\nu,\varepsilon}^k \,,$$

*where $n_k(T) = \mathbb{E}\left[\sum_{r=1}^{T-1} \mathbb{1}(k \in \mathcal{A}_{r+1}, \ell^r = 1)\right]$, $C_{\nu,\varepsilon}^k$ is independent on $T$ and, denoting $\mathcal{X}_n$ the set of $n$ first observations of arm 1,*

$$B_{T,\varepsilon}^k = \sum_{j=2}^{K} \sum_{n=1}^{\lceil 2\log(T)/I_1(\mu_k + \varepsilon) \rceil} \sup_{\mu \in [\mu_j - \varepsilon, \mu_j + \varepsilon]} \mathbb{E}_{\mathcal{X}_n} \left[ \frac{\mathbb{1}\left(\mu(\mathcal{X}_n) \leq \mu\right)}{\mathbb{P}(\widetilde{\mu}(\mathcal{X}_n, \mu) \geq \mu)} \right] \,.$$

In this section we study the regret of a general DS algorithm, for any index function $\widetilde{\mu}$. We exploit the duel structure of the algorithm in order to exhibit the the two crucial terms that depend on the DS index and have to be controlled in Theorem 3.1. Before starting the analysis we recall the first condition that we assume on the arms distributions, which ensures the concentration of the means.

**Condition 1 (C1): concentration of the means** For all $k$, there exists a good rate function $I_k$ satisfying for all $x \geq \mu_k, y \leq \mu_k$, and any i.i.d sequence $X_1, \ldots, X_n$ drawn from $\nu_k$

$$\mathbb{P}\left(\frac{1}{n}\sum_{i=1}^{n} X_i \geq x\right) \leq e^{-nI_k(x)}, \text{ and } \mathbb{P}\left(\frac{1}{n}\sum_{i=1}^{n} X_i \leq y\right) \leq e^{-nI_k(y)}.$$

This is actually the only result we need to prove Theorem 3.1, we do not need to introduce the DS indexes for this result. We will carefully detail each term of the Theorem, in particular the constant $C_{\nu,\varepsilon}^k$ whose components are all explicit.

## B.1 Regret decomposition

Thanks to the duel structure of DS, the fact that an arm is pulled or not depends of its status as a leader or a challenger. If an arm is a challenger, it can be pulled only if it wins its duel against the leader. For this reason, a natural first regret decomposition consists in considering the cases when 1) the optimal arm is the leader and some sub-optimal arms are pulled, and 2) the optimal arm is not the leader. Thanks to the definition of the round-regret $\mathcal{R}_T$ in Equation 3, and denoting arm 1 as the unique optimal arm (without loss of generality), we upper bound $\mathcal{R}_T$ by controlling the expectation of the number of pulls of each sub-optimal arm $k \in \{2, \ldots, K\}$. Using that all arms are pulled during the first round we obtain

$$\mathbb{E}[N_k(T)] = 1 + \mathbb{E}\left[\sum_{r=1}^{T-1} \mathbb{1}(k \in \mathcal{A}_{r+1})\right]$$

$$= 1 + \sum_{r=1}^{T-1} \mathbb{E}\left[\mathbb{1}(k \in \mathcal{A}_{r+1}, \ell^r = 1) + \mathbb{1}(k \in \mathcal{A}_{r+1}, \ell^r \neq 1)\right]$$

$$\leq 1 + \underbrace{\mathbb{E}\left[\sum_{r=1}^{T-1} \mathbb{1}(k \in \mathcal{A}_{r+1}, \ell^r = 1)\right]}_{n_k(T)} + \underbrace{\mathbb{E}\left[\sum_{r=1}^{T-1} \mathbb{1}(\ell^r \neq 1)\right]}_{E_T^k}.$$

We already extracted the term $n_k(T)$ of Theorem 3.1 at this step, and introduced a term $E_T^k$ that contains both $B_{T,\varepsilon}^k$ and $C_{\nu,\varepsilon}^k$. In the rest of this proof we work on the term $E_T^k$.

## B.2 Upper bound on $E_T^k$

The following part of the proof is inspired by the proof of the SSMC algorithm in Chan (2020). Furthermore, we will see that we can further decompose $E_T^k$ into several events that we will handle using condition (C1), showing the interest of the algorithm' structure in rounds and duels. After that, we will finally exhibit the term that requires the condition (C2).

Before that, we start by analyzing two alternatives that can cause the event $\{\ell(r) \neq 1\}$, namely 1) arm 1 has already been leader and has lost the leadership at some point (leadership takeover), or 2) arm 1 has never been leader. To formalize these alternatives we define the sequence $a_r = \lceil r/4 \rceil$ and the event

$$\mathcal{D}_r = \{\exists u \in [a_r, r] : \ell^u = 1\}.$$

**$\mathcal{D}_r$ is true: arm 1 has already been leader after round $a_r$**

We first justify the choice of $a_r$: starting after a number of rounds that is linear in $r$ ensures a number of observations for the leader during the whole segment $[a_r, r]$ that is also linear in $r$, that is for all

$s \in [a_r, r]$,

$$N_{\ell^s}(s) \geq \lceil a_r/K \rceil := b_r .$$

Under $\mathcal{D}_r$ we study the probability of a leadership takeover by a sub-optimal arm between $a_r$ and $r$. Such takeover can happen only if 1) a sub-optimal arm obtain the same number of samples as arm 1, and 2) its sample average is larger than the one of arm 1 at the round when it happens. We denote $\mathcal{X}_n^k$ the history available for arm $k$ after it has been sampled $n$ times, and drop the exponent for arm 1. We formalize the leadership takeover with the following events,

$$
\begin{aligned}
\{\ell^r \neq 1\} \cap \mathcal{D}^r &\subset \cup_{u=a_r}^{r-1} \{\ell^u = 1, \ell^{u+1} \neq 1\} \\
&\subset \cup_{u=a_r}^{r-1} \cup_{k=2}^K \{\ell^u = 1, k \in \mathcal{A}_{u+1}, N_k(u+1) = N_1(u+1), \mu(\mathcal{X}_{N_k(u+1)}^k) \geq \mu(\mathcal{X}_{N_1(u+1)})\} \\
&\subset \cup_{u=a_r}^{r-1} \cup_{k=2}^K \{N_k(u+1) = N_1(u+1), \mu(\mathcal{X}_{N_k(u+1)}^k) \geq \mu(\mathcal{X}_{N_1(u+1)})\} \\
&=\subset \cup_{u=a_r+1}^{r} \cup_{k=2}^K \{N_k(u) = N_1(u), \mu(\mathcal{X}_{N_k(u)}^k) \geq \mu(\mathcal{X}_{N_1(u)})\} \\
&:= \cup_{u=a_r+1}^{r} \cup_{k=2}^K \mathcal{B}_k^u .
\end{aligned}
$$

Starting the sum on the rounds at some round $r_0$, we first develop and express the sum of these terms as

$$
\begin{aligned}
\mathbb{E}\left[\sum_{r=r_0}^{T-1} \sum_{u=a_r+1}^{t} \mathbb{1}(\mathcal{B}_k^u)\right] &= \mathbb{E}\left[\sum_{r=r_0}^{T-1} \sum_{u=a_r+1}^{t} \mathbb{1}(N_k(u) = N_1(u), \mu(\mathcal{X}_{N_k(u)}^k) \geq \mu(\mathcal{X}_{N_1(u)}))\right] \\
&\leq \mathbb{E}\left[\sum_{r=r_0}^{T-1} \sum_{u=a_r+1}^{r} \sum_{n=\lceil u/K \rceil}^{u} \mathbb{1}(N_k(u) = N_1(u) = n, \mu(\mathcal{X}_n^k) \geq \mu(\mathcal{X}_n))\right] \\
&\leq \mathbb{E}\left[\sum_{r=r_0}^{T-1} \sum_{u=a_r+1}^{r} \sum_{n=\lceil u/K \rceil}^{u} \mathbb{1}(\mu(\mathcal{X}_n^k) \geq \mu(\mathcal{X}_n))\right] .
\end{aligned}
$$

We now define $x_k = \frac{\mu_1 + \mu_k}{2}$. If $\mu(\mathcal{X}_n^k) \geq \mu(\mathcal{X}_n)$, then either the arm 1 had "under-performed" or arm $k$ has "over-performed", which means that $\mu(\mathcal{X}_n^k) \geq x_k$ or $\mu(\mathcal{X}_n) \leq x_k$, which gives

$$
\begin{aligned}
\mathbb{E}\left[\sum_{r=r_0}^{T-1} \sum_{u=a_r+1}^{t} \mathbb{1}(\mathcal{B}_k^u)\right] &\leq \mathbb{E}\left[\sum_{r=r_0}^{T-1} \sum_{u=a_t+1}^{t} \sum_{n=\lceil u/K \rceil}^{u} \mathbb{1}(\mu(\mathcal{X}_n^k) \geq x_k \cup \mu(\mathcal{X}_n) \leq x_k)\right] \\
&\leq \sum_{r=r_0}^{T-1} \sum_{u=a_t+1}^{t} \sum_{n=\lceil u/K \rceil}^{u} \left[\mathbb{P}(\mu(\mathcal{X}_n^k) \geq x_k) + \mathbb{P}(\mu(\mathcal{X}_n) \leq x_k)\right] \\
&\leq \sum_{r=r_0}^{T-1} r^2 \left[\mathbb{P}(\mu(\mathcal{X}_{b_r}^k) \geq x_k) + \mathbb{P}(\mu(\mathcal{X}_{b_r}) \leq x_k)\right] \\
&\leq \sum_{r=r_0}^{T-1} r^2 \left(e^{-b_r I_1(x_k)} + e^{-b_r I_k(x_k)}\right) \\
&= O(1) ,
\end{aligned}
$$

where the two last lines come from the condition (C1), which is the existence of a good rate function for each arm's distribution. Thanks to the structure of the algorithm this condition is enough to upper bound the cost of leadership takeover by sub-optimal arms by constants in the regret. This convergent series is the first component of the term $C_{\nu,\varepsilon}^k$ in Theorem 3.1. We now consider the case when arm 1 has never been leader.

**Upper bound when $\mathcal{D}_r$ is not true**

The idea in this part is to leverage the fact that if the optimal arm is not leader between $\lceil r/4 \rceil$ and $r$, then it has necessarily lost a lot of duels. We introduce the count of the number of duels lost by arm 1,

$$L^r = \sum_{u=a_r}^{r} \mathbb{1}(\mathcal{C}^u) \, ,$$

with $\mathcal{C}^u = \{\exists k \neq 1, \ell^u = k, 1 \notin \mathcal{A}_{u+1}\}$ representing the event that at round $u$ arm 1 is challenger and has lost its duel. We then consider the upper bound

$$\mathbb{P}(\ell^r \neq 1 \cap \bar{\mathcal{D}}^r) \leq \mathbb{P}(L^r \geq r/4) \, . \tag{7}$$

This result is a direct adaptation from (Chan, 2020) (Equation 7.12), we then use of the Markov inequality to obtain

$$\mathbb{P}(L^r \geq r/4) \leq \frac{\mathbb{E}(L^r)}{r/4} = \frac{4}{r} \sum_{u=a_r}^{r} \mathbb{P}(\mathcal{C}^u) \, . \tag{8}$$

We then remove the double sum on $u$ and $t$ by simply counting the number of occurrences of each term,

$$\sum_{r=r_0}^{T-1} \mathbb{P}(L^r \geq r/4) \leq \mathbb{E}\left[ \sum_{r=r_0}^{T-1} \frac{4}{r} \sum_{u=a_r}^{r} \mathbb{1}(\mathcal{C}^u) \right]$$

$$\leq \mathbb{E}\left[ \sum_{r=r_0}^{T-1} \sum_{u=a_{r_0}}^{T-1} \frac{4}{r} \mathbb{1}(\mathcal{C}^u) \mathbb{1}(u \in [a_r, r]) \right]$$

$$\leq \mathbb{E}\left[ \sum_{u=a_{r_0}}^{T-1} \mathbb{1}(\mathcal{C}^u) \sum_{r=r_0}^{T-1} \frac{4}{r} \mathbb{1}(u \in [a_r, r]) \right] \, .$$

From this step we can control independently the sum in $r$,

$$\sum_{r=r_0}^{T-1} \frac{4}{r} \mathbb{1}(u \in [a_r, r]) \sum_{r=r_0}^{T-1} 4 \frac{\mathbb{1}(u \leq r)}{r} \mathbb{1}(a_r \leq u)$$

$$\leq \frac{4}{u} \sum_{r=r_0}^{T-1} \mathbb{1}(a_r \leq u)$$

$$\leq \frac{4}{u} \sum_{r=r_0}^{T-1} \mathbb{1}(\lceil r/4 \rceil \leq u)$$

$$\leq \frac{4}{u} \sum_{r=r_0}^{T-1} \mathbb{1}(r/4 \leq u+1)$$

$$\leq \frac{4}{u} \times 4(u+1)$$

$$\leq 32 \, .$$

With this result we obtain that

$$\sum_{r=r_0}^{T-1} \mathbb{P}(L^r \geq r/4) \leq 32 \sum_{r=a_{r_0}}^{T-1} \mathbb{P}(\mathcal{C}^r) \, .$$

We then decompose as the union of all the terms corresponding to each possible sub-optimal leader,

$$\mathcal{C}^r = \cup_{j=2}^K \{\ell^r = j, 1 \notin \mathcal{A}_{r+1}\} := \cup_{j=2}^K \mathcal{C}_j^r .$$

We can now fix any sub-optimal leader $j$ and work on the term $\mathcal{C}_j^r$. We recall that arm 1 has two chances to win the duel: first with its empirical mean, and then with a random index. We first handle the case when the sub-optimal leader could be "over-performing", by writing for any $\varepsilon > 0$

$$\mathcal{C}_j^r \subset \left\{ \left| \mu\left(\mathcal{X}_{N_j(r)}^j\right) - \mu_j \right| \geq \varepsilon, \ell^r = j \right\}$$
$$\cup \left\{ \left| \mu\left(\mathcal{X}_{N_j(r)}^j\right) - \mu_j \right| \leq \varepsilon, \ell^r = j, \mu\left(\mathcal{X}_{N_1(r)}\right) \leq \mu\left(\mathcal{X}_{N_j(r)}^j\right), \widetilde{\mu}\left(\mathcal{X}_{N_1(r)}, \mathcal{X}_{N_j(r)}^j\right) \leq \mu\left(\mathcal{X}_{N_j(r)}^j\right) \right\} .$$

We then upper bound the left-hand term using again the concentration of the leader, and obtain

$$\sum_{r=1}^{T-1} \mathbb{P}\left( \left| \mu\left(\mathcal{X}_{N_j(r)}^j\right) - \mu_j \right| \geq \varepsilon, \ell^r = k \right)$$
$$= \sum_{r=1}^{T-1} \sum_{n_j = \lceil r/K \rceil} \mathbb{P}\left( \left| \mu\left(\mathcal{X}_{N_j(r)}^j\right) - \mu_j \right| \geq \varepsilon, N_j(r) = n_j \right)$$
$$= O(1).$$

For the simplicity of notations we keep the notation $\mathcal{C}_j^r$ to define the right-hand term. We then continue the analysis of $\mathcal{C}_j^r$ by considering the number of samples of arm 1, and in particular if $N_1(r) \geq n_1(T)$ or not, for some new function $n_1(T)$. Considering that arm 1 has a first chance with its empirical mean, and that under $\mathcal{C}_k^r$ the leader's mean is controlled we can fix $n_1(T)$ in order to ensure that for any $n \geq n_1(T)$: $\mathbb{P}(\mu\left(\mathcal{X}_n\right) \leq \mu_j + \varepsilon) \leq 1/T^2$. Thanks to condition (C1) this is the case for $n_1(T) = \lceil 2/I_1(\mu_j + \varepsilon) \log T \rceil$. We now define the event

$$\mathcal{H}_j^{r,n} = \left\{ N_1(r) = n, \left| \mu\left(\mathcal{X}_{N_j(r)}^j\right) - \mu_j \right| \leq \varepsilon, \mu\left(\mathcal{X}_n\right) \leq \mu\left(\mathcal{X}_{N_j(r)}^j\right), \right.$$
$$\left. \widetilde{\mu}\left(\mathcal{X}_n, \mathcal{X}_{N_j(r)}^j\right) \leq \mu\left(\mathcal{X}_{N_j(r)}^j\right) \right\} ,$$

and use it to define a new upper bound on $\mathcal{C}_j^r$,

$$\sum_{r=a_{r_0}}^{T-1} \mathbb{P}\left(\mathcal{C}_j^r\right) \leq \sum_{r=a_{r_0}}^{T-1} \sum_{n=1}^{T-1} \mathbb{P}\left(\mathcal{H}_j^{r,n}\right)$$
$$\leq \sum_{r=a_{r_0}}^{T-1} \sum_{n=n_1(T)}^{T-1} \mathbb{P}\left(N_1(r) = n, \mu(\mathcal{X}_n) \leq \mu_j + \varepsilon\right) + \sum_{r=a_{r_0}}^{T-1} \sum_{n=1}^{n_1(T)} \mathbb{P}\left(\mathcal{H}_j^{r,n}\right)$$
$$\leq 1 + \sum_{r=a_{r_0}}^{T-1} \sum_{n=1}^{n_1(T)} \mathbb{P}\left(\mathcal{H}_j^{r,n}\right) .$$

We then use as in [Riou and Honda (2020)](#) that

$$\sum_{r=a_{r_0}}^{T-1} \mathbb{1}(\mathcal{H}_j^{r,n}) = \sum_{m=1}^{T-1} \mathbb{1}\left( \sum_{r=a_{r_0}}^{T-1} \mathbb{1}(\mathcal{H}_j^{r,n}) \geq m \right) ,$$

and define as $\tau_1^n, \ldots, \tau_m^n$ the $m$ first rounds for which $\mathcal{H}_j^{r,n}$ hold. If $\mathbb{1}\left(\sum_{r=a_{r_0}}^{T-1} \mathbb{1}(\mathcal{H}_j^{r,n}) \geq m\right)$ is true then $\mathcal{H}_j^{\tau_j,n}$ holds for any $i \leq m$ and all these $\tau_i$ are finite, which means that

$$\mathbb{1}\left(\sum_{r=a_{r_0}}^{T-1} \mathbb{1}(\mathcal{H}_j^{r,n}) \geq m\right) \leq \prod_{i=1}^{m} \mathbb{1}\left(\mathcal{H}_j^{\tau_i^n,n}\right) .$$

So, we denote the term that is left to upper bound as

$$
\begin{aligned}
D_{T,\varepsilon}^j &= \sum_{n=1}^{n_1(T)} \sum_{m=1}^{T-1} \mathbb{E}\left[\prod_{i=1}^{m} \mathbb{1}\left(\mathcal{H}_j^{\tau_i^n,n}\right)\right] \\
&= \sum_{n=1}^{n_1(T)} \sum_{m=1}^{T-1} \mathbb{E}_{\mathcal{X}_n}\left[\prod_{i=1}^{m} \mathbb{P}\left(\widetilde{\mu}\left(\mathcal{X}_n, \mathcal{X}_{N_j(\tau_i^n)}^k\right) \leq \mu\left(\mathcal{X}_{N_j(\tau_i^n)}^k\right) \Big| \mathcal{X}_n\right) \mathbb{1}\left(\mathcal{H}_j^{\tau_i^n,n}\right)\right] .
\end{aligned}
$$

In the next steps we remove the dependency of the index in $\mathcal{X}_{N_j(\tau_i^n)}^k$, knowing that the index only depends of its mean and that the mean is located in a small range around $\mu_j$. At this step, we notice that we could have replaced this control of the mean by locating the empirical distribution of $k$ in any space "around" its true distribution, assuming a concentration similar as (C1) for the corresponding event. However, to simplify the notations we assume that the index $\widetilde{\mu}$ only depends on the history of $k$ through its mean, and we further upper bound $B_{T,\varepsilon}^k$ by simply taking the maximum value of the expectation for any possible value of $\mu\left(\mathcal{X}_{N_j(\tau_i^n)}^k\right)$,

$$
\begin{aligned}
D_{T,\varepsilon}^j &\leq \sum_{n=1}^{n_1(T)} \sum_{m=1}^{T-1} \sup_{\mu \in [\mu_j - \varepsilon, \mu_j + \varepsilon]} \mathbb{E}_{\mathcal{X}_n}\left[\mathbb{P}\left(\widetilde{\mu}\left(\mathcal{X}_n, \mu\right) \leq \mu\right)^m \mathbb{1}\left(\mu\left(\mathcal{X}_n\right) \leq \mu\right)\right] \\
&\leq \sum_{n=1}^{n_1(T)} \sup_{\mu \in [\mu_j - \varepsilon, \mu_j + \varepsilon]} \mathbb{E}_{\mathcal{X}_n}\left[\frac{\mathbb{P}\left(\widetilde{\mu}\left(\mathcal{X}_n, \mu\right) \leq \mu\right)}{\mathbb{P}\left(\widetilde{\mu}\left(\mathcal{X}_n, \mu\right) \geq \mu\right)} \mathbb{1}\left(\mu\left(\mathcal{X}_n\right) \leq \mu\right)\right] .
\end{aligned}
$$

which concludes the proof if we define $B_{T,\varepsilon}^k = \sum_{j=2}^{K} D_{T,\varepsilon}^j$ in Theorem 3.1.

## C   Technical results on the Dirichlet distribution

In this section we provide the proofs of the technical results in Section 3.1, along with other results that we use in the proofs of Appendix D but did not introduced in the main paper due to space limitations. Before proving the upper and lower bounds on the Boundary Crossing Probability, we present some basic properties of the Dirichlet distribution for readers that are not familiar with this distribution.

### C.1   Basic properties of the Dirichlet distribution

We consider the Dirichlet distribution $\text{Dir}(\alpha)$ for some parameter $\alpha = (\alpha_1, \ldots, \alpha_n)$. Let $w = (w_1, \ldots, w_n)$ be a random variable drawn from the distribution $\text{Dir}(\alpha)$. We first recall that $w$ takes its values in the probability simplex $\mathcal{P}^n = \{p \in [0,1]^n : \sum_{i=1}^{n} p_i = 1\}$. The distribution admits the following density,

$$f(w_1, \ldots, w_n) = \frac{\Gamma(\sum_{i=1}^{n} \alpha_i)}{\prod_{i=1}^{n} \Gamma(\alpha_i)} \prod_{i=1}^{n} w_i^{\alpha_i - 1} ,$$

where $\Gamma$ denotes the Gamma function. In this paper we only consider integer values for the coefficient $(\alpha_i)_{i \in \mathbb{N}}$, and for any $m \in \mathbb{N}$ $\Gamma(m) = (m-1)!$. Denoting $N = \sum_{i=1}^{n} \alpha_i$, we obtain the more convenient form

$$f(w_1, \ldots, w_n) = \frac{(N-1)!}{\prod_{i=1}^{n} (\alpha_i - 1)!} \prod_{i=1}^{n} w_i^{\alpha_i - 1} .$$

This distribution has a lot of convenient properties. First, interpreting $\alpha_i/N$ has the frequency of an item in a set of observations drawn from a finite collection (empirical distribution), and $w$ a random re-weighting of these observation providing a "noisy" empirical distribution, the Dirichlet distribution ensures that the noisy frequency of each item is unbiased with respect to the observed frequency, with a variance that is inversely proportional to the total number of items collected. For any $i \in [1, n]$,

$$\mathbb{E}[w_i] = \frac{\alpha_i}{N} , \qquad \text{and} \qquad \mathbb{V}(w_i) = \frac{\alpha_i(N - \alpha_i)}{N^2(N + 1)} ,$$

and the marginal density of each component of $w$ is actually a distribution $\mathrm{Beta}(\alpha_i, N - \alpha_i)$. This explains the use of the Dirichlet distribution to generalize the Beta-Bernoulli Thompson Sampling.

In this paper we also use two main properties of the Dirichlet distribution, both using the relation between the Dirichlet distribution and the Exponential distribution. Let $R_1, \ldots, R_n$ be $n$ i.i.d random variables drawn from exponential distributions with respective parameters $\alpha_i$, $R_i \sim \mathcal{E}(\alpha_i)$. Then the vector $w = (w_1, \ldots, w_n)$ with $w_i = \frac{R_i}{\sum_{j=1}^{n} R_j}$ follows a Dirichlet distribution $\mathrm{Dir}(\alpha)$.

The second property is a consequence of the first one, and is that the components of a random variable drawn from a Dirichlet distribution and can be aggregated, providing another Dirichlet distribution: if $w \sim \mathrm{Dir}(\alpha)$, then $w' = (w_1, \ldots, w_i + w_j, \ldots, w_n) \sim \mathrm{Dir}((\alpha_1, \ldots, \alpha_i + \alpha_j, \ldots, \alpha_n))$ (putting the sum in the $i - th$ slot and removing the $j$-th slot without changing the other indices). In particular, we will make use of this property in the proofs of Theorem 3.4 and 3.5. Indeed, we will *discretize* the data, i.e grouping observations from continuous distributions into bins of the same size and studying the properties of the resulting distribution.

## C.2   Boundary Crossing Probability for Dirichlet re-weighting

We recall the definition of the *Boundary Crossing Probability*, given in Section 3.1. In this section we prove Lemma 3.2 and 3.3, that we restate below, along with other technical results including some from Riou and Honda (2020).

**Boundary crossing probability (BCP)**   We consider a set of $n + 1$ observation points $\mathcal{X} = (X_1, \ldots, X_{n+1}) \subset \mathbb{R}^{n+1}$. (Intuitively, $n$ points are samples from a challenger arm, and one point corresponds to the added bonus). Then, for any $\mu \in \mathbb{R}$, we introduce the following "Boundary Crossing Probability" (BCP) term, conditional on $\mathcal{X}$

$$[\text{BCP}] \coloneqq \mathbb{P}_{w \sim \mathcal{D}_{n+1}} \left( \sum_{i=1}^{n+1} w_i X_i \geq \mu \right) .$$

When all observations are *distinct* from each other this BCP has a closed formula, which has been derived for instance in Cho and Cho (2001) as

$$\mathbb{P}_{w \sim \mathcal{D}_{n+1}} \left( \sum_{i=1}^{n+1} w_i X_i \geq \mu \right) = \sum_{i=1}^{n+1} \frac{(X_i - \mu)_+^n}{\prod_{j=1, j \neq i}^{n+1} (X_i - X_j)} . \tag{9}$$

This expression is obtained by computing the volume of the half-space of the simplex defined by the hyperplane $\sum_{i=1}^{n+1} w_i X_i \geq \mu$. Unfortunately, this formula is not very informative: for sorted data the terms are alternatively positive and negative, and can take large values (compensating each other). This makes the exact formula hardly tractable even for numerical simulations. We also add that the closed formula does not exist for a Dirichlet distribution with some parameters larger than 1, as it would require a closed formula for the incomplete beta function.

Hence, both upper and lower bounds for the BCP have to be studied independently of this formula. We start with upper bounds.

**Upper bound on the BCP**   The first result we introduce is a variant of Lemma 15 of Riou and Honda (2020).

**Lemma C.1** (Upper bound on the BCP). *Consider a set $\mathcal{X} = (X_1, \ldots, X_{n+1})$ and a target value $\mu \in \mathbb{R}$, and denote $\bar{\mathcal{X}} = \max_{i=1}^{n+1} X_i$ and $\nu_{\mathcal{X}}$ the empirical distribution associated to $\mathcal{X}$. The BCP satisfies*

$$\mathbb{P}_{w \sim \mathcal{D}_{n+1}}\left(\sum_{i=1}^{n+1} w_i X_i \geq \mu\right) \leq e^{-\max_{\lambda \in [0,1)} \sum_{i=1}^{n+1}\left[\log\left(1 - \lambda \frac{X_i - \mu}{\bar{\mathcal{X}} - \mu}\right)\right]} \leq e^{-(n+1)\mathcal{K}_{\inf}^{\bar{\mathcal{X}}}(\nu_{\mathcal{X}}, \mu)},$$

*When distributions have support upper-bounded by $B$, replacing $\bar{\mathcal{X}}$ by $B$ makes the $\mathcal{K}_{\inf}$ functional data-independent.*

*Proof.* This result is a variant of Lemma 15 of Riou and Honda (2020), hence we rewrite the beginning of their proof, which consists in using the Chernoff method to upper bound the BCP and writing the Dirichlet weight with exponential variables. For any $\lambda \in [0, 1)$, denoting $\bar{\mathcal{X}} = \max_{i=1}^{n+1} X_i$ it holds that

$$[\text{BCP}] \leq \mathbb{P}_{R_1, \ldots, R_n \sim \mathcal{E}(1)}\left(\sum_{i=1}^{n+1} R_i \frac{X_i - \mu}{\bar{\mathcal{X}} - \mu} \geq 0\right)$$

$$\leq \mathbb{P}_{R_1, \ldots, R_n \sim \mathcal{E}(1)} \mathbb{P}\left(\exp\left(\lambda \sum_{i=1}^{n+1} R_i \frac{X_i - \mu}{\bar{\mathcal{X}} - \mu}\right) \geq 1\right)$$

$$\leq \prod_{i=1}^{n+1} \mathbb{E}_{R_i}\left[\exp\left(\lambda R_i \frac{X_i - \mu}{\bar{\mathcal{X}} - \mu}\right)\right],$$

which is so far exactly the Chernoff method, up to the rescaling by $\bar{\mathcal{X}} - \mu$. For $\lambda < 1$, each MGF is defined and has an explicit formula, so

$$[\text{BCP}] \leq \prod_{i=1}^{n+1} \frac{1}{1 - \lambda \frac{X_i - \mu}{\bar{\mathcal{X}} - \mu}}$$

$$= \exp\left(-\sum_{i=1}^{n+1} \log\left(1 - \lambda \frac{X_i - \mu}{\bar{\mathcal{X}} - \mu}\right)\right).$$

We obtain the first inequality of the lemma by choosing the maximum over all possible values of $\lambda$, and the second inequality is direct when writing the dual problem associated with $\mathcal{K}_{\inf}^{\bar{\mathcal{X}}}(\hat{\nu}_{n+1}, \mu)$ (see e.g Honda and Takemura (2010, 2015)). $\square$

**Lower bounds on the BCP** We now consider the anti-concentration of the BCP, that suggests a sound tuning of the bonus in DS algorithms. Under this perspective, we first provide a necessary condition of the bonus in DS to ensure sufficient exploration.

**Lemma C.2** (Necessary condition with a data-independent bonus). *Consider a fixed bonus $B(\mathcal{X}, \mu) = B(\mu)$, and a distribution $F$ (with CDF also denoted $F$). If Condition (C2) holds then*

$$B(\mu) > \mu + \frac{1}{1 - F(\mu)} \mathbb{E}_F\left[(\mu - X)_+\right].$$

*Proof.* When all the observations are below the threshold equation 9 provides

$$\mathbb{P}_{w \sim \mathcal{D}_n}\left(\sum_{i=1}^{n} w_i X_i + w_{n+1} B(\mu) \geq \mu\right) = \prod_{i=1}^{n} \frac{B - \mu}{B - X_i},$$

so plugging this term in (C2) gives the expression

$$\mathbb{E}\left[\prod_{i=1}^{n}\left(\frac{B - X_i}{B - \mu}\right) \mathbb{1}(X_i \leq \mu)\right] = \mathbb{E}_{X_1 \sim F}\left[\left(\frac{B - X_1}{B - \mu}\right) \mathbb{1}(X_1 \leq \mu)\right]^n.$$

Condition (C2) can hold only if the expectation is smaller than 1, which is equivalent to

$$(B - \mu)(1 - F(\mu)) \geq \mathbb{E}\left[(\mu - X)_+\right],$$

which gives the result. $\square$

**Remark C.3.** *The proof also work if we do not consider the events $\{X_i \leq \mu\}$ but instead $\{X_i \leq y\}$ for any $y \in \mathbb{R}$. We use this property in the proof of Theorem 3.4 for instance. If we directly consider the quantile $q_{1-\alpha}$ of $F$ we can define*

$$B(\mu, \alpha) = \mu + \frac{1}{\alpha} \mathbb{E}_F \left[ (\mu - X) \mathbb{1}(X \leq q_{1-\alpha}) \right] .$$

*Another alternative hypothesis could be that there exist some $\alpha > 0, B_\alpha$ satisfying*

$$\mathbb{E}_F \left[ (\mu_1 - X) \mathbb{1}(X \leq q_{1-\alpha}) \right] \leq B_\alpha ,$$

*that would then provide a condition*

$$B(\mu, \alpha) \geq \mu + \frac{B_\alpha}{\alpha} .$$

**Lemma C.4** (Lower bound for the BCP). *Consider a set $\mathcal{X} = (X_1, \ldots, X_{n+1}) \in \mathbb{R}^{n+1}$, and assume that $\overline{\mathcal{X}} = \max_{i \in \{1, \ldots, n+1\}} X_i \geq g(n)$ for some function g. Denoting $\bar{\Delta}_n^+ = \frac{1}{n} \sum_{i=1, X_i < \overline{\mathcal{X}}}^{n+1} (\mu - X_i)^+$ the empirical positive gap, it holds that*

$$\mathbb{P}_{w \sim \mathcal{D}_{n+1}} \left( \sum_{i=1}^{n+1} w_i X_i \geq \mu \right) \geq \exp \left( -n \frac{\bar{\Delta}_n^+}{g(n) - \mu} \right) .$$

*Proof.* We obtain this lower bound by truncating all the observations that are larger than the threshold except the maximum of $\bar{\mathcal{X}}$, allowing to use Equation 9. Combining this property with $\log(1 + x) \leq x$ we obtain

$$\mathbb{P}_{w \sim \mathcal{D}_{n+1}} \left( \sum_{i=1}^{n+1} w_i X_i \geq \mu \right) \geq \mathbb{P}_{w \sim \mathcal{D}_{n+1}} \left( \sum_{i=1}^{n} w_i \min(X_i, \mu) + w_{n+1} \bar{\mathcal{X}} \geq \mu \right)$$

$$= \frac{(\bar{\mathcal{X}} - \mu)^n}{\prod_{i=1}^{n} (\bar{\mathcal{X}} - \min(X_i, \mu))}$$

$$= \exp \left( -\sum_{i=1}^{n} \log \left( \frac{\bar{\mathcal{X}} - \min(X_i, \mu)}{\bar{\mathcal{X}} - \mu} \right) \right)$$

$$= \exp \left( -\sum_{i=1}^{n} \log \left( 1 + \frac{\mu - \min(X_i, \mu)}{\bar{\mathcal{X}} - \mu} \right) \right)$$

$$\geq \exp \left( -\sum_{i=1}^{n} \frac{\mu - \min(X_i, \mu)}{\bar{\mathcal{X}} - \mu} \right)$$

$$= \exp \left( -\sum_{i=1}^{n} \frac{(\mu - X_i)_+}{\bar{\mathcal{X}} - \mu} \right) ,$$

which yields the result. $\qquad\square$

We finally provide another lower bound on the BCP that is used to derive Lemma 14 in (Riou and Honda, 2020).

**Lemma C.5** (Second Lower Bound for the BCP). *Consider observations $\mathcal{X} = (X_1, \ldots, X_n)$, a parameter $\alpha = (\alpha_1, \ldots, \alpha_n) \subset \mathbb{N}^n$, and $\mu \in \mathbb{R}$. We add a value $B$ to the dataset, and denote $\widetilde{\alpha} = (\alpha_1, \ldots, \alpha_n, 1)$. We also denote $\bar{\mathcal{X}} = \max \{\max_{i=1}^{n} X_i, B\}$, and $N = \sum_{i=1}^{n} \alpha_i$. Then, for any vector $w^\star \in \mathcal{P}^{n+1}$ satisfying $\sum_{i=1}^{n} w_i^\star X_i + w_{n+1}^\star B \geq \mu$ it holds that*

$$\mathbb{P}_{w \sim Dir(\widetilde{\alpha})} \left( \sum_{i=1}^{n} w_i X_i + w_{n+1} B \geq \mu \right) \geq \frac{N!}{\prod_{i=1}^{n} \alpha_i!} \prod_{i=1}^{n} (w_i^\star)^{\alpha_i} \left( \alpha_{i_M} \frac{w_{n+1}^*}{w_{i_M}^*} \right) ,$$

*where $i_M$ is the index satisfying $X_{i_M} = \bar{\mathcal{X}}$ (setting $i_M = n + 1$ if $\bar{\mathcal{X}} = B$).*

*Proof.* We use the density of the Dirichlet distribution defined in Appendix C.1, and consider the set $\mathcal{S} = \{w \in \mathcal{P}^{n+1} : \sum_{i=1}^{n} w_i X_i + w_{n+1} B \geq \mu\}$, which is the set defined in the BCP. Then, considering any allocation $w^\star \in \mathcal{S}$, we define the subset $\mathcal{S}_2 = \{w \in \mathcal{P}^{n+1} : \forall i \neq i_M, w_i \in [0, w_i^\star]\}$. The inclusion $\mathcal{S}_2 \subset \mathcal{S}$ is direct : transferring weights to the maximum can only increase the value of the weighted sum. Hence, $\mathbb{P}(w \in \mathcal{S}_2)$ is a lower bound of $\mathbb{P}(w \in \mathcal{S})$. We then obtain

$$
\mathbb{P}_{w \sim \text{Dir}(\alpha)} \left( \sum_{i=1}^{n+1} w_i X_i \geq \mu \right) \geq \mathbb{P}(w \in \mathcal{S}_2)
$$

$$
\geq \frac{N!}{\prod_{i=1}^{n}(\alpha_i - 1)!} \int_0^{w_1^\star} \cdots \int_0^{w_{n+1}^\star} \prod_{i=1}^{n} w_i^{\alpha_i - 1} \prod_{i=1, i \neq i_M}^{n+1} \mathrm{d}w_i
$$

$$
= \frac{N!}{\prod_{i=1, i \neq i_M}^{n} \alpha_i!} \prod_{i=1}^{n} (w_i^\star)^{\alpha_i} \left( \frac{w_{n+1}^*}{w_{i_M}^*} \right)
$$

$$
= \frac{N!}{\prod_{i=1}^{n} \alpha_i!} \prod_{i=1}^{n} (w_i^\star)^{\alpha_i} \left( \alpha_{i_M} \frac{w_{n+1}^*}{w_{i_M}^*} \right) ,
$$

which concludes the proof. $\qquad\square$

A direct corollary of this result is to choose the weights $w^\star$ that maximize the lower bound, which gives as in Lemma C.1 an expression with $\mathcal{K}_{\text{inf}}^{\bar{\mathcal{X}}}$.

**Corollary C.5.1.** *Take the notations of Lemma C.5 and Consider $\nu_{\mathcal{X},\alpha}$ the multinomial distribution with atoms $\mathcal{X} = (X_1, \ldots, X_n)$ and probabilities $p_\alpha = \left( \frac{\alpha_1}{N}, \ldots, \frac{\alpha_n}{N} \right)$. It holds that*

$$
\mathbb{P}_{w \sim Dir(\widetilde{\alpha})} \left( \sum_{i=1}^{n} w_i X_i + w_{n+1} B \geq \mu \right) \geq \frac{N!}{\prod_{i=1}^{n} \alpha_i!} \exp\left( -N \left( \mathcal{K}_{\text{inf}}^{\bar{\mathcal{X}}} (\nu_{\mathcal{X},\alpha}, \mu) + H(\nu_{n,\alpha}, \mu) \right) \right) .
$$

*Proof.* Consider two multinomial distributions $\nu_1, \nu_2$ with same support and respective probabilities $p = (p_1, \ldots, p_n)$ and $q = (q_1, \ldots, q_n)$ the Kullback-Leibler divergence is simply

$$
\text{KL}(\nu_1, \nu_2) = \sum_{i=1}^{n} p_i \log(p_i/q_i) := - \sum_{i=1}^{n} p_i \log(q_i) - H(\nu_1) .
$$

We first write the result of Lemma C.5 with $p = p_\alpha$ and $q = w^\star$, and choose $w^\star = \inf \text{KL}(p_\alpha, w^\star) = \mathcal{K}_{\text{inf}}^{\bar{\mathcal{X}}}(p_\alpha, \mu)$ (with a slight abuse of notation, denoting the distributions by their probabilities). Furthermore, we simplify the constants using that

$$
\alpha_{i_M} \frac{w_{n+1}^\star}{w_{i_M}^\star} \geq \frac{w_{n+1}^\star}{w_{i_M}^\star} \geq 1 ,
$$

with equality only if $i_M = n + 1$. Indeed, the optimal allocation will necessarily put more weights on largest values, so $w_{i_M}^\star \geq w_{n+1}^\star$. $\qquad\square$

# D   Regret bounds of Section 3.2

In this section we provide the complete proof of Theorems 3.4, 3.7 and 3.5, presented in Section 3.2. For each of these results we follow the same path: starting from Theorem 3.1, which holds in each case, we then detail the terms $n_k(T)$. This first part exhibits the first-order terms of the regret bound. Then, we prove that condition (C2) holds (anti-concentration of the BCP, avoiding under-exploration of the best arm). We justify in the proofs the hypothesis we consider and the theoretical tuning of the parameters of the algorithms. In all cases, we justify that (C1) holds under the settings we consider.

We recall the terms we have to study, from Theorem 3.1.

**First-order term**

$$
\forall k \in \{1, \ldots, K\}, : n_k(T) = \mathbb{E}\left[ \sum_{r=1}^{T-1} \mathbb{1}(k \in \mathcal{A}_{r+1}, \ell^r = 1) \right] .
$$

**Condition (C2)** For any $\mu < \mu_1$, and any $n_1(T) = o(\log T)$ it holds that

$$\sum_{n=1}^{n_1(T)} \mathbb{E}_{\mathcal{X}_n \sim \nu_1^n} \left[ \frac{\mathbb{1}(\mu(\mathcal{X}_n) \leq \mu)}{\mathbb{P}_{w \sim \mathcal{D}_{n+1}} \left( \widetilde{\mu}(\mathcal{X}_n, \mu) \geq \mu \right)} \right] = o(\log(T)) .$$

The proofs of the three theorems share common elements, so before instantiating the proof for each algorithm we further work on these two terms under general assumptions.

### D.1 General proof sketches

In this section we derive the parts of the proofs of condition (C2) and (C3) that are shared by all three instances of DS.

**Further characterization of $n_k(T)$**

In this section we consider an arm $k \in \{2, \dots, K\}$, of distribution $\nu_k$.

**Lemma D.1.** *Assume that $\nu_k$ satisfies (C1), and denote $\mathcal{X}_n^k = (X_1^k, \dots, X_n^k)$ a set of $n$ random variables drawn from $\nu_k$. Assume that for any $n \in \mathbb{N}$ there exists a subset $\mathcal{B}_{k,n} \subset \mathbb{R}^n$ satisfying*

*1. $\mathcal{X}_n^k \subset \mathcal{B}_{k,n} \Rightarrow \mathbb{P} \left( \widetilde{\mu}(\mathcal{X}_n^k, \mu) \geq \mu \right) \leq \exp \left( -f_k(n, \mathcal{B}_{k,n}, \mu) \right)$, for a DS index $\widetilde{\mu}$, a fixed threshold $\mu \in \mathbb{R}$ and a fixed strictly increasing function $f_k$.*

*2. $\sum_{n=1}^{T-1} \mathbb{P} \left( \mathcal{X}_n^k \notin \mathcal{B}_{k,n} \right) \leq C_{\mathcal{B}_k}$ for some constant $C_{\mathcal{B}_k}$.*

*If these two conditions hold, then it holds that*

$$n_k(T) = m_k(T) + \mathcal{O}(1) ,$$

*where $m_k(T)$ is the sequence satisfying $f_k(n_k(T), \mathcal{B}_{k,n}, \mu_1 + \varepsilon) = \log T$.*

*Proof.* We first split the sequence $(\mathbb{1} (k \in \mathcal{A}_{r+1}, \ell^r = 1))_{r=1,\dots,T-1}$ in a pre-convergence phase, the size of which we control, and a post-convergence phase, for which arm $k$ has been pulled enough times so we can use concentration (C1) to hold. To this end, we define a function $m_k(T)$ without specifying it for the moment and write

$$\mathbb{E}\left[ \sum_{r=1}^{T-1} \mathbb{1}(k \in \mathcal{A}_{r+1}, \ell^r = 1) \right] \leq \mathbb{E}\left[ \sum_{r=1}^{T-1} \mathbb{1}(k \in \mathcal{A}_{r+1}, \ell^r = 1, N_k(r) < m_k(T)) \right]$$

$$+ \mathbb{E}\left[ \sum_{r=1}^{T-1} \mathbb{1}(k \in \mathcal{A}_{r+1}, \ell^r = 1, N_k(r) \geq m_k(T)) \right]$$

$$\leq m_k(T) + \mathbb{E}\left[ \sum_{r=1}^{T-1} \mathbb{1}(k \in \mathcal{A}_{r+1}, \ell^r = 1, N_k(r) \geq m_k(T)) \right] ,$$

where we used that for any $n$, the event $\{k \in \mathcal{A}_{r+1}, N_k(r) = n\}$ can happen at most once. The next step is to further split the second term by defining a "good event" of large probability under which $(k \in \mathcal{A}_{r+1})$ has a low probability. As we aim at keeping some level of generality in this section, we simply define this event as

$$\mathcal{G}_k^r = \left\{ \mathcal{X}_{N_k(r)}^k \in \mathcal{B}_{k, N_k(r)} \right\} \cap \left\{ \mu(\mathcal{X}_{N_1(r)}) \leq \mu_1 - \varepsilon_1 \right\} ,$$

where $\varepsilon_1 > 0$, and use the same notations as in Appendix B for the other terms.

We further define the event

$$\mathcal{W}_k^r = \{k \in \mathcal{A}_{r+1}, \ell^r = 1, N_k(r) \geq m_k(T)\} .$$

Then, it holds that

$$n_k(T) \leq m_k(T) + \mathbb{E}\left[\sum_{r=1}^{T-1} \mathbb{1}(\mathcal{W}_k^r, \mathcal{G}_k^r) + \sum_{r=1}^{T-1} \mathbb{1}(\mathcal{W}_k^r, \bar{\mathcal{G}}_k^r)\right] .$$

We use the first assumption in the lemma to upper bound the left-hand term as

$$
\begin{aligned}
\mathbb{E}\left[\sum_{r=1}^{T-1} \mathbb{1}(\mathcal{W}_k^r, \mathcal{G}_k^r)\right] &= \mathbb{E}\left[\sum_{r=1}^{T-1} \sum_{n=m_k(T)}^{T-1} \mathbb{1}\left(k \in \mathcal{A}_{r+1}, \ell^r = 1, N_k(r) = n, \mathcal{G}_k^r\right)\right] \\
&\leq \mathbb{E}\left[\sum_{r=1}^{T-1} \sum_{n=m_k(T)}^{T-1} \mathbb{1}\left(\widetilde{\mu}\left(\mathcal{X}_{N_k(r)}^k, \mathcal{X}_{N_1(r)}\right) \geq \mu(\mathcal{X}_{N_1(r)}), N_k(r) = n, \mathcal{W}_k^r, \mathcal{G}_k^r\right)\right] \\
&\leq \mathbb{E}\left[\sum_{r=1}^{T-1} \sum_{n=m_k(T)}^{T-1} \mathbb{P}\left(\widetilde{\mu}\left(\mathcal{X}_{N_k(r)}^k, \mathcal{X}_{N_1(r)}\right) \geq \mu(\mathcal{X}_{N_1(r)})\Big| \mathcal{X}_{N_k(r)}^k, \mathcal{X}_{N_1(r)}\right) \mathbb{1}(N_k(r) = n, \mathcal{W}_k^r, \mathcal{G}_k^r)\right] \\
&\leq \mathbb{E}\left[\sum_{r=1}^{T-1} \sum_{n=m_k(T)}^{T-1} \exp\left(-f\left(n, \mathcal{B}_{k,n}, \mu(\mathcal{X}_{N_1(r)})\right)\right) \mathbb{1}(k \in \mathcal{A}_{r+1}, \ell^r = 1, N_k(r) = n, \mathcal{G}_k^r)\right] \\
&\leq \mathbb{E}\left[\sum_{r=1}^{T-1} \sum_{n=m_k(T)}^{T-1} \exp\left(-f\left(n, \mathcal{B}_{k,n}, \mu_1 - \varepsilon_1\right)\right) \mathbb{1}(k \in \mathcal{A}_{r+1}, N_k(r) = n)\right] ,
\end{aligned}
$$

where the last two lines come directly from Assumption 1 in the lemma ans using the second term of $\mathcal{G}_k^r$ involving arm 1. We complete this step of the proof by further using the monotonicity of $f$ in $n$,

$$
\begin{aligned}
\mathbb{E}\left[\sum_{r=1}^{T-1} \mathbb{1}(\mathcal{W}_k^r, \mathcal{G}_k^r)\right] &\leq e^{-f(m_k(T), \mathcal{B}_{k,m_k(T)}, \mu_1 - \varepsilon_1)} \mathbb{E}\left[\sum_{r=1}^{T-1} \sum_{n=m_k(T)}^{T-1} \mathbb{1}(k \in \mathcal{A}_{r+1}, N_k(r) = n)\right] \\
&\leq e^{-f(m_k(T), \mathcal{B}_{k,m_k(T)}, \mu_1 - \varepsilon_1)} \mathbb{E}\left[\sum_{n=m_k(T)}^{T-1} 1\right] \\
&\leq T \exp\left(-f(m_k(T), \mathcal{B}_{k,m_k(T)}, \mu_1 - \varepsilon_1)\right) .
\end{aligned}
$$

We handle the right-hand term before discussing this result, and directly write the union bound $\bar{\mathcal{G}}_k^r \subset \left\{\mathcal{X}_{N_k(r)}^k \notin \mathcal{B}_{k,N_k(r)}\right\} \cup \left\{\mu(\mathcal{X}_{N_1(r)}) > \mu_1 - \varepsilon_1\right\}$, that leads to

$$\mathbb{E}\left[\sum_{r=1}^{T-1} \mathbb{1}(\mathcal{W}_k^r, \bar{\mathcal{G}}_k^r)\right] = \mathbb{E}\left[\sum_{r=1}^{T-1} \mathbb{1}(k \in \mathcal{A}_{r+1}, \ell^r = 1, N_k(r) \geq m_k(T), \bar{\mathcal{G}}_k^r)\right]$$

$$\leq \underbrace{\mathbb{E}\left[\sum_{r=1}^{T-1} \mathbb{1}(\ell^r = 1, \mu(\mathcal{X}_{N_1(r)}) > \mu_1 - \varepsilon_1)\right]}_{A_1}$$

$$+ \underbrace{\mathbb{E}\left[\sum_{r=1}^{T-1} \mathbb{1}(k \in \mathcal{A}_{r+1}, N_k(r) \geq m_k(T), \mathcal{X}_{N_k(r)}^k \notin \mathcal{B}_{k,N_k(r)})\right]}_{A_2},$$

where the two terms $A_1$ and $A_2$ depend respectively only of arm $k$ and arm 1. The first term can be handled thanks to condition (C1) on arm 1, and using that the leader has necessarily a linear number of samples,

$$A_1 \leq \sum_{r=1}^{T-1} \mathbb{E}\left[\mathbb{1}(N_1(r) \geq \lceil r/K \rceil, \bar{\mu}_1^r \geq \mu_1 + \varepsilon_1)\right]$$

$$\leq \sum_{r=1}^{T-1} \sum_{n=\lceil r/K \rceil}^{r} \mathbb{P}(\mu(\mathcal{X}_{N_1(r)}) \geq \mu_1 - \varepsilon_1)$$

$$\leq \sum_{r=1}^{T-1} \sum_{n=\lceil r/K \rceil}^{r} e^{-nI_1(\mu_1 - \varepsilon_1)}$$

$$\leq \sum_{r=1}^{T-1} r e^{-\lceil r/K \rceil I_1(\mu_1 - \varepsilon_1)}$$

$$= \mathcal{O}(1).$$

We now upper bound $A_2$, using again that $\sum_{r=1}^{T-1} \mathbb{1}(k \in \mathcal{A}_{r+1}, N_k(r) = n) \leq 1$ for any $n \in \mathbb{N}$,

$$A_2 \leq \sum_{r=1}^{T-1} \sum_{n=m_k(T)}^{T-1} \mathbb{E}\left[\mathbb{1}(k \in \mathcal{A}_{r+1}, N_k(r) = n, \mathcal{X}_{N_k(r)}^k \notin \mathcal{B}_{k,N_k(r)})\right]$$

$$\leq \sum_{n=m_k(T)}^{T-1} \mathbb{P}\left(\mathcal{X}_n^k \notin \mathcal{B}_{k,n}\right).$$

Combining these results, we obtain a bound on $n_k(T)$ for arm $k$ as

$$n_k(T) \leq m_k(T) + T e^{-f(m_k(T), \mathcal{B}_{k,m_k(T)}, \mu_1 - \varepsilon_1)} + \sum_{n=m_k(T)}^{T-1} \mathbb{P}\left(\mathcal{X}_n^k \notin \mathcal{B}_k\right) + \mathcal{O}(1).$$

We see that if $\mathcal{B}_{k,n}$ is designed to make the series convergent, and $m_k(T)$ is chosen as the sequence satisfying $f(m_k(T), \mathcal{B}_{k,m_k(T)}, \mu_1 - \varepsilon_1) = \log T$, then we finally obtain

$$n_k(T) \leq m_k(T) + \mathcal{O}(1).$$

$\square$

Thanks to this result, when we will adapt the proof for each algorithm we will be able to combine Lemma D.1 with Lemma C.1 to directly look for a proper choice of the set $\mathcal{B}_{k,n}$.

**Condition (C2)**

In this section we take the result of Corollary C.5.1 and use it to derive a ratio between the likelihood of an empirical distribution and its BCP in the case of multinomial distributions. This result will be useful in the proofs of Theorem 3.4 and 3.5 where we use an adaptive discretization in order to work with multinomial distributions.

**Lemma D.2** (Balance between the likelihood and the BCP for multinomial distribution). *Consider observations $\mathcal{X} = (X_1, \ldots, X_n, B)$ and denote $\bar{\mathcal{X}} = \max \mathcal{X} \geq \mu \in \mathbb{R}$. Now consider a multinomial distribution $\nu_n$ supported on $(X_1, \ldots, X_n)$ and of probability $p_n \in \mathcal{P}^n$.*

*We fix some $N \in \mathbb{N}$ and denote by $\beta_N \in \mathbb{N}^N$ a random vector denoting the counts of each item $X_1, \ldots, X_n$ when drawing $N$ observations from $\nu_n$. Then for any vector $\beta = (\beta_1, \ldots, \beta_n)$ satisfying $\sum_{i=1}^{n} \beta_i = N$ and $\beta^t X \leq \mu$ for some $\mu \in \mathbb{R}$, it holds that*

$$\frac{\mathbb{P}(\beta_N = \beta)}{\mathbb{P}_{w \sim Dir((\beta^t, 1))}(\sum_{i=1}^{n} w_i \beta_i + w_{n+1} B \geq \mu)} \leq \exp\left(-n\left[\mathrm{KL}\left(\frac{\beta}{N}, p_n\right) - \mathcal{K}_{\inf}^{\bar{\mathcal{X}}}\left(\frac{\beta}{N}, \mu\right)\right]\right) .$$

*Proof.* Lemma 2.1.6 in (Dembo and Zeitouni, 2010) provides that for a multinomial distribution it holds that

$$\mathbb{P}(\beta_N = \beta) = \frac{N!}{\prod_{i=1}^{n} \beta_i!} \prod_{i=1}^{n} p_{n,i}^{\beta_i} = \frac{N!}{\prod_{i=1}^{n} \beta_i!} \exp\left(-N(\mathrm{KL}(\beta/N, p_n) + H(\beta/N))\right) ,$$

where $H$ denotes the entropy. Then, Corollary C.5.1 directly provides the result as all the constant terms are equal, and the entropy term can be simplified. $\square$

## D.2 Proof of Theorem 3.4: regret bound for BDS

We recall that two hypothesis are considered for BDS: (B1) distributions are bounded and the upper bound of the support is known, and (B2) the upper bound is not known but for each distribution $\nu_k$ it holds that $\mathbb{P}_{\nu_k}([B - \gamma, B]) \geq p$ for some known $\gamma, p$. We know restate the Theorem.

**Theorem 3.4** (Optimality of BDS). *If $\forall k \in \{2, \ldots, K\}$, $\nu_k \sim \mathcal{F}_{\mathcal{B}}^{\gamma, \rho}$, choosing the exploration bonus of Equation 6 with $\rho \geq -1/\log(1 - p)$ ensures that*

$$\mathbb{E}[N_k(T)] \leq \frac{\log(T)}{\mathcal{K}_{\inf}^{B_{\rho, \gamma}}(\nu_k, \mu_1)} + O(1) ,$$

*where $B_{\rho, \gamma} = \max\left(B + \gamma, \mu_1 + \rho \mathbb{E}_{\nu_k}[(\mu_1 - \mu_k)_+]\right)$.*

*Proof.* We denote $\bar{\mathcal{X}} = \max_{X \in \mathcal{X}} X$ and for $\mathcal{X} = (X_1, \ldots, X_n)$,

$$B_{\mathrm{BDS}}(\mathcal{X}, \mu) = \max\left\{\bar{\mathcal{X}} + \gamma, \mu + \rho \frac{1}{n} \sum_{i=1}^{n} (\mu - X_i)^+\right\} ,$$

where parameter $\gamma$ directly comes from the hypothesis on the distributions, and we justify below the tuning of parameter $\rho$ as a function of $p$. First of all, the bounded support hypothesis ensures condition (C1) thanks to Hoeffding inequality, with a rate function $I_k(x) = \frac{2(x - \mu_k)^2}{B^2}$. We can now focus on the expression of $n_k(T)$ and on proving condition (C2).

**First-order term**

We use Lemma D.1. To define the large-probability event $\mathcal{B}_{k,n}$, we consider the Levy distance

$$d(\nu_F, \nu_G) = \inf\{\varepsilon > 0 : G(x - \varepsilon) - \varepsilon \leq F(x) \leq G(x + \varepsilon) + \varepsilon\} ,$$

where $\nu_F$ and $\nu_G$ are two distributions of respective cdf $F$ and $G$.

In this section, we fix some $\varepsilon > 0$ (different than the one from Th. 3.1 but we avoid an index for simplicity) and consider $\mathcal{B}_{k,n}$ as a Levy ball of size $\varepsilon$ around the true distribution:

$$\mathcal{B}_{k,n} \left\{ \mathcal{X} \in \mathbb{R}^n : d(\nu_\mathcal{X}, \nu_k) \le \varepsilon \right\} ,$$

The objective is to use the continuity of the $\mathcal{K}_{\inf}^B$ function in its first argument with respect to the Levy distance (see e.g [Honda and Takemura (2010)](#)). From now on we denote (in this section) distributions by their cdf: let $F_{k,n}$ be the empirical distribution associated with the set $\mathcal{X}_n^k$ and $\widetilde{F}_{k,n}$ be the *biased* empirical distribution to which the bonus of the BDS algorithm has been added. We first prove that $F_{k,n}$ belongs to the Levy ball with high probability, using the relation between the Levy distance and the supremum norm

$$d(F_{k,n}, F_k) \le ||F_{k,n} - F_k||_\infty .$$

We use the Dvoretzky-Kiefer-Wolfowitz (DKW) inequality (see e.g [Massart (1990)](#)), that states that

$$\mathbb{P}\left(||F_{k,n} - F_k||_\infty \ge \varepsilon\right) \le 2e^{-2n\,\varepsilon^2} .$$

Hence, the convergence of $\sum_{n=1}^{+\infty} \mathbb{P}\left(d(F_{k,n}, F_k) \ge \varepsilon\right)$ is direct. Now considering the event $||F_{k,n} - F_k||_\infty \ge \varepsilon$, we prove that the biased distribution $\widetilde{F}_{k,n}$ (adding the bonus in the set of observations) is also close to $F_k$ in the sense of the supremum norm. First, the triangular inequality provides $||\widetilde{F}_{k,n} - F_k||_\infty \le ||F_{k,n} - F_k||_\infty + ||\widetilde{F}_{k,n} - F_{k,n}||_\infty$. We upper bound the second term with

$$
\begin{aligned}
|\widetilde{F}_{k,n}(x) - F_{k,n}(x)| &\le \max \left\{ \left| \frac{k}{n+1} - \frac{k}{n} \right|, \left| \frac{k}{n+1} - \frac{k}{n} \right| \right\} \\
&\le \max \left\{ \left| \frac{k}{n(n+1)} \right|, \left| \frac{n-k}{n(n+1)} \right| \right\} \\
&\le \frac{1}{n+1} ,
\end{aligned}
$$

so if $||F_{k,n} - F_k||_\infty \le \varepsilon$, then $||\widetilde{F}_{k,n} - F_k||_\infty \le \varepsilon + (n+1)^{-1}$, and finally

$$||F_{k,n} - F_k||_\infty \le \varepsilon \Rightarrow d(\widetilde{F}_{k,n}, F_k) \le \varepsilon + \frac{1}{n+1} .$$

Hence, if we combine these results we obtain that for $n$ large enough $\widetilde{F}_{k,n}$ is also in a Levy ball around $F_k$, of size $\varepsilon'$ slightly larger than $\varepsilon$, with large probability.

Now that the event $\mathcal{B}_{k,n}$ is defined and we derived its properties, we can find the function $f$ in Lemma D.1 in the case of BDS. We denote $X_{n+1}$ the bonus of BDS and use Lemma C.1 to obtain

$$\mathbb{P}_{w \sim \mathcal{D}_{n+1}} \left( \sum_{i=1}^n w_i X_i + w_{n+1} B_{\text{BDS}}(\mathcal{X}, \mu) \ge \mu \right) \le e^{-(n+1)\mathcal{K}_{\inf}^{B_{\text{BDS}}(\mathcal{X},\mu)}(\widetilde{F}_{k,n}, \mu)} ,$$

Furthermore, under the event $\mathcal{B}_{k,n}$ and the fact that the mean of the leader is concentrated around its true mean the bonus of the BDS index is upper bounded by $B_{\rho,\gamma} + \varepsilon'$, for some $\varepsilon' > 0$. We use the continuity of $\mathcal{K}_{\inf}^B$ with respect to 1) the first argument in terms of the Levy distance, 2) the second argument (e.g w.r.t the euclidian norm), 3) the upper bound: for any $\varepsilon_0 > 0$, we can calibrate the $\varepsilon$ in the Levy ball to obtain

$$\mathbb{P}_{w \sim \mathcal{D}_n} \left( \sum_{i=1}^{n+1} w_i X_i \ge \mu \right) \le e^{-(n+1)(\mathcal{K}_{\inf}^{B_{\rho,\gamma}}(\nu_k, \mu_1) - \varepsilon_0)} ,$$

hence we conclude this part by setting exactly $m_k(T) = \dfrac{\log(T)}{\mathcal{K}_{\inf}^{B_{\rho,\gamma}}(\nu_k, \mu_1) - \varepsilon_0}$.

**Condition (C2)**

We now study the quantity

$$E_n = \mathbb{E}_{\mathcal{X}_n \sim \nu_1^n} \left[ \frac{\mathbb{1}(\mu(\mathcal{X}_n)) \leq \mu)}{\mathbb{P}\left(\widetilde{\mu}(\mathcal{X}_n, \mu) \geq \mu\right)} \right]$$

when $\mu < \mu_1$. We first use Lemma 3.3 to obtain the lower bound on the BCP

$$\mathbb{P}_{w \sim \mathcal{D}_{n+1}} \left(\widetilde{\mu}(\mathcal{X}_n, \mu) \geq \mu\right) \geq e^{-\frac{n}{\rho}} \ ,$$

with $\rho$ the parameter chosen in the component of the bonus that follows Equation 5

Using this results and (C1), we obtain a first bound

$$E_n \leq e^{-n(I(\mu) - 1/\rho)} \ ,$$

which is sufficient to obtain (C3) if $I_1(\mu) \geq 1/\rho$. We then consider the case when it is not sufficient, and now use the hypothesis $\mathbb{P}([B - \gamma, B]) \geq p$ and the second component of the bonus, $\bar{\mathcal{X}}_n + \gamma := \max X_i + \gamma$ to obtain

$$
\begin{aligned}
E_n \leq & \mathbb{E}_{\mathcal{X}_n} \left[ \frac{\mathbb{1}(\mu(\mathcal{X}_n) \leq \mu)(\mathbb{1}(\bar{\mathcal{X}}_n \leq B - \gamma) + \mathbb{1}(\bar{\mathcal{X}}_n \geq B - \gamma))}{\mathbb{P}\left(\widetilde{\mu}(\mathcal{X}_n, \mu) \geq \mu\right)} \right] \\
\leq & \underbrace{(1 - p)^n e^{\frac{n}{\rho}}}_{E_{n,1}} + \underbrace{\mathbb{E}_{\mathcal{X}_n} \left[ \frac{\mathbb{1}(\mu(\mathcal{X}_n) \leq \mu)\mathbb{1}(\bar{\mathcal{X}}_n + \gamma \geq B)}{\mathbb{P}\left(\widetilde{\mu}(\mathcal{X}_n, \mu) \geq \mu\right)} \right]}_{E_{n,2}} \ .
\end{aligned}
$$

The two terms correspond to the two possible expressions for the bonus. The term $E_{n,1}$ gives the sufficient condition for the tuning of $\rho$ in Theorem 3.4 with

$$\rho > \frac{-1}{\log(1 - p)} \Rightarrow \sum_{n=1}^{+\infty} E_{n,1} = O(1) \ .$$

In the second term, the exploration bonus is larger than $B$, so we can use the same proof scheme as in Riou and Honda (2020), which is also the case (B1) we consider here. First, we discretize the interval $[0, B]$ in equally sized bins of size $\eta$, and consider the truncated variables $\widetilde{X}_i = \eta\lfloor X_i/\eta \rfloor$. $\eta$ is chosen small enough to ensure that $\mu_1 - \eta > \mu$, i.e the truncated distribution still has a mean larger than $\mu$. An upper bound of $E_{n,2}$ is obtained by replacing the variables $X_i$ by $\widetilde{X}_i$. We associate a set of observations $(\widetilde{X}_1, \ldots, \widetilde{X}_n)$ with the vector $\beta_n$ of size $S = \lceil B/\eta \rceil$ which counts the number of observations falling in each bin. The number of possible values for $\beta_n$ is upper bounded by $n^S$, and we use Lemma D.2 to obtain

$$
\begin{aligned}
\frac{\mathbb{P}(\beta)}{\mathbb{P}_{w \sim \text{Dir}(\widetilde{\beta})}\left(w^t \widetilde{\beta} \geq \mu\right)} \leq & \exp\left(-n \left[\text{KL}(\beta/n, \widetilde{p}_1) - \mathcal{K}_{\text{inf}}^{\bar{\mathcal{X}}+\gamma}(\beta/n, \mu)\right]\right) \\
\leq & \exp\left(-n \left[\text{KL}(\beta/n, \widetilde{p}_1) - \mathcal{K}_{\text{inf}}^B(\beta/n, \mu)\right]\right) \\
\leq & \exp\left(-n \left[\mathcal{K}_{\text{inf}}^B(\beta/n, \mu_1 - \eta) - \mathcal{K}_{\text{inf}}^B(\beta/n, \mu)\right]\right) \ ,
\end{aligned}
$$

As $\mu < \mu_1 - \eta$, there exists some $\delta > 0$ satisfying $\forall \beta : \beta^t \widetilde{X} < \mu$, $\mathcal{K}_{\text{inf}}^B(\beta/n, \mu_1 - \eta) - \mathcal{K}_{\text{inf}}^B(\beta/n, \mu) > \delta$. So finally, denoting $\mathcal{C}$ the set of the possible count vectors $\beta$, it holds that

$$E_{n,2} \leq \sum_{\beta \in \mathcal{C}} e^{-n\left[\mathcal{K}_{\text{inf}}^B(\beta/n, \mu_1 - \eta) - \mathcal{K}_{\text{inf}}^B(\beta/n, \mu)\right]} \leq n^S e^{-n\delta} \ ,$$

The two components of the bonus ensures that condition (C2) is satisfied for the distribution that satisfies hypothesis of Theorem 3.4. This completes the proof of the theorem. □

### D.3 Proof of Theorem 3.5: logarithmic regret of QDS for semi-bounded supports

**Theorem 3.5** (Logarithmic Regret of QDS). *Consider a bandit model $\nu = (\nu_1, \ldots, \nu_K)$ satisfying $\forall k$, $\nu_k \in \mathcal{F}^\alpha_{[b,+\infty)}$ for some $b > -\infty$ (lower-bounded support) and a known $\alpha > 0$. Then, for any $\varepsilon_0 > 0$ small enough QDS with any parameters $\alpha' < \alpha$ and $\rho \geq (1 + \alpha')/\alpha'^2$ satisfies*

$$\mathbb{E}[N_k(T)] \leq \frac{\log T}{\mathcal{K}^{\mathfrak{M}^C_k}_{\inf}\left(\mathcal{T}_\alpha(\nu_k), \mu_1\right) - \varepsilon_0} + \mathcal{O}(1) ,$$

*with $\mathfrak{M}^C_k = \max\{C_\alpha(\nu_k), \mu_1 + \rho \mathbb{E}_\nu[(\mu_1 - X)^+]$, and $\mathcal{T}_\alpha$ is the truncation operator we defined.*

*Proof.* We start by simply stating that conditions (C1) hold, for the same reason as for RDS because we consider again light-tailed distributions. The rest of the proof is similar to the proof of Theorem 3.4 for BDS.

**Upper bounding $n_k(T)$**

We again want to use Lemma D.1, and formulate a high-probability event on the observations. First, we can build a Levy ball around the true distribution to control the value of the quantile thanks to DKW inequality. Secondly, we use as in RDS that the variable $(\mu - X)^+$ for $X \sim \nu_k$ is light-tailed and hence admits a good rate function $I^+_k$ thanks to Cramér's theorem.

$$\mathcal{B}_{k,n} = \{\mathcal{X} \in \mathbb{R}^n : d_L(\nu_\mathcal{X}, \nu_k) \leq \varepsilon, |B(\mathcal{X}, \rho, \mu) - B_{k,\rho,\mu}| \leq \varepsilon_1, C_{\alpha'}(\nu_\mathcal{X}) \leq C_{\alpha'}(\nu_k) + \varepsilon_2\} ,$$

for some $\varepsilon > 0, \varepsilon_1, \varepsilon_2 > 0$, denoting $\nu_\mathcal{X}$ the empirical distribution associated with a set $\mathcal{X}$, $B_{k,\rho,\mu} = \mu + \rho \times \mathbb{E}_{\nu_k}[(\mu - X)_+]$, and defining the application $C_{\alpha'}$ as the *Conditional Value at risk* for a level $\alpha'$. If $\nu_k$ is continuous, it simply holds that $C_{\alpha'}(\nu_k) = \mathbb{E}_{\nu_k}[X | X \geq q_{1-\alpha'}(\nu_k)]$.

We consider $\sum_{n=1}^{+\infty} \mathbb{P}_{\mathcal{X}_n}(\mathcal{X}_n \notin \mathcal{B}_{k,n})$ and first refer the reader to the proofs of Theorem 3.4 and 3.7, respectively for the terms corresponding to the Levy distance and the concentration of the bonus (relying as we recall on the DKW inequality and a good rate function for the data-dependent bonus), for empirical distribution we simply take the mean of all data larger than the empirical quantile $q_{1-\alpha'}(\nu_\mathcal{X})$. However, as in the proof of Theorem 3.7 we will be able to handle this thanks to the concentration of Wasserstein metrics for light-tailed distribution, using that (Lemma 2 from Bhat and L.A. (2019)) for two distributions $\nu$ and $\nu'$ it holds that

$$|C_\alpha(\nu) - C_\alpha(\nu')| \leq \frac{1}{1 - \alpha'} W_1(\nu, \nu') .$$

Then we can again use Theorem 2 from Fournier and Guillin (2015) to obtain a concentration inequality on this term. With all these results, it holds that

$$\sum_{n=1}^{+\infty} \mathbb{P}_{\mathcal{X}_n}(\mathcal{X}_n \notin \mathcal{B}_{k,n}) < +\infty ,$$

so the observations are in $\mathcal{B}_{k,n}$ with large probability, hence we can now consider the BCP under $\mathcal{X}_n \in \mathcal{B}_{k,n}$. The difference compared with previous section is that this time the BCP is considered for the truncated distribution $\mathcal{T}(\nu_{\mathcal{X}_n})$. However, this is not a problem as the upper bound of lemma C.1 still holds. Thanks to the aggregation properties of the Dirichlet distribution (see Appendix C.1 for more details), the BCP with parameter $(1, \ldots, 1, n_\alpha)$ (of size $n - n_{\alpha'}$) is the same as the BCP with parameters $(1, \ldots, 1)$ (of size $n$) with $n_{\alpha'}$ copies of the last term. Hence, the QDS index satisfies

$$\mathbb{P}\left(\widetilde{\mu}(\mathcal{X}_n, \mu) \geq \mu\right) \leq \exp\left(-(n+1)\mathcal{K}^{M_{\mathcal{X}_n}}_{\inf}\left(\mathcal{T}(\nu_{\mathcal{X}_n}), \mu\right)\right) .$$

If $\mathcal{X}_n \in \mathcal{B}_{k,n}$, then $M_{\mathcal{X}_n}$ is upper bounded by

$$M_{\mathcal{X}_n} \leq \max\left(C_{\alpha'}(\nu_k), B_{k,\rho,\mu}\right) + \max(\varepsilon_1, \varepsilon_2) ,$$

We now define $\mathfrak{M}^C_k = \max\left(C_{\alpha'}(\nu_k), B_{k,\rho,\mu}\right)$, that is independent of the run of the bandit algorithm.

Finally, the definition of the Levy distance ensures that $d\left(\nu_{\mathcal{X}_n}, \nu_k\right) \leq \varepsilon \Rightarrow \leq d\left(\mathcal{T}(\nu_{\mathcal{X}_n}), \mathcal{T}(\nu_k)\right) \leq \varepsilon$. Hence, we can use the continuity of $\mathcal{K}_{\inf}^{M_k}$ in all arguments (including $M_{\mathcal{X}_n}$, see e.g Honda and Takemura (2015)) and obtain that for any $\varepsilon_0$ we can calibrate $\varepsilon, \varepsilon_1, \varepsilon_2$ in order to obtain

$$\mathbb{P}\left(\widetilde{\mu}(\mathcal{X}_n, \mu) \geq \mu\right) \leq \exp\left(-(n+1)\left(\mathcal{K}_{\inf}^{\mathfrak{M}_k^C}\left(\mathcal{T}(\nu), \mu\right) - \varepsilon_0\right)\right) ,$$

which gives the first order term of Theorem 3.5 choosing $m_k(T) = \frac{\log T}{\mathcal{K}_{\inf}^{M_k}(\mathcal{T}(\nu_k), \mu) - \varepsilon_0}$ in Lemma D.1.

**Condition (C2)**

In this section we use the assumption that rewards are semi-bounded with a range $[b, +\infty]$. Then, we can find a value $y$ and a discretization step $\eta$ such that truncating the values $X_i$ to $\min(X_i, y)$, and truncating each $X_i < y$ to $\widetilde{X}_i = \eta\left\lfloor\frac{X_i}{\eta}\right\rfloor$ preserves the order of $\mu < \widetilde{\mu}_1$. Note that this value $y$ does not have to be known by the algorithm and is purely an artifact for the proof. This discretization is similar to the proof of Theorem 3.4 in Appendix D.2. We denote $S$ the number of items created by the discretization, and $\beta \in \mathbb{N}^S$ some vector of counts.

However, contrarily to the proof of BDS we directly try to use Lemma D.2 and consider for any $\beta \in \mathbb{N}^S : ||\beta||_1 = n$ the quantity

$$K_\beta = \mathrm{KL}(\beta/n, \widetilde{\nu}_1) - \mathcal{K}_{\inf}^{m_{\widetilde{\beta}}}(\beta/n, \mu_k) ,$$

where $\widetilde{\nu}_1$ denote the discretized/truncated version of $\nu_1$ and $m_\beta$ denotes the largest item with a non-zero coefficient in $\widetilde{\beta}$, which is itself $\beta$ with an additional value associated with the bonus. We recall that QDS summarizes the information larger than the empirical $(1 - \alpha)$-quantile by their mean (i.e the $\mathrm{CVaR}_\alpha$ of the empirical distribution). The truncation in $y$ does not change that, and will simply makes this quantity smaller which will itself makes the BCP smaller (although not so much with well chosen $\eta, y$). We use the result from Honda and Takemura (2010) (proof of Theorem 7) stating that for any $\beta$

$$\mathcal{K}_{\inf}^{m_{\widetilde{\beta}}}(\beta/n, \mu_k) \leq \frac{\bar{\Delta}_n}{M_\beta - \mu} ,$$

As we know that $M_\beta$ is at least larger than the exploration bonus, we furthermore have

$$\mathcal{K}_{\inf}^{m_{\widetilde{\beta}}}(\beta/n, \mu_k) \leq \frac{\bar{\Delta}_n}{\rho\bar{\Delta}_n^+} \leq 1/\rho .$$

This means that for any $\xi > 0$ it holds that $K_B \geq \xi$ on all the sub-space of empirical distributions satisfying $\mathrm{KL}(\beta/n, \widetilde{\nu}_1) \geq (1 + \xi)/\rho$.

We now use Pinsker inequality to link the KL divergence with the total variation $\delta$, in the sub-space where $\mathrm{KL}(\beta/n, \widetilde{\nu}_1) \leq (1 + \xi)/\rho$,

$$\delta(\beta/n, \widetilde{\nu}_1) \leq \sqrt{\frac{1 + \xi}{2\rho}} .$$

If this quantity is small, we can control the probability of each measurable event. In particular, we want the quantile *used by the algorithm* to be strictly larger than the $(1-\alpha)$-quantile *of the assumption* of Theorem 3.5. If the parameter of the condition of the theorem is $\alpha$, and we run the algorithm with a parameter $\alpha' < \alpha$, then we know that if we properly tune $\rho$ we will have $F_{k,n}(q_{1-\alpha}(F_k)) < 1 - \alpha$. This means that the *true quantile $q_{1-\alpha}(\nu_k)$ is present in the set $\mathcal{X}_n$ and is not truncated by the algorithm*. In particular, if $\rho \geq \frac{1+\alpha'}{\alpha'^2}$ this is satisfied, and finally

$$\mathrm{KL}(\beta/n, \widetilde{\nu}_1) - \mathcal{K}_{\mathrm{inf}}^{m_\beta}(\beta/n, \mu_k) \geq \mathcal{K}_{\mathrm{inf}}^{\mathcal{F}}(\beta/n, \mu_1 - \eta) - \mathcal{K}_{\mathrm{inf}}^{q_{1-\alpha'}}(\beta/n, \mu_k)$$
$$\geq \mathcal{K}_{\mathrm{inf}}^{q_{1-\alpha}}(\beta/n, \mu_1 - \eta) - \mathcal{K}_{\mathrm{inf}}^{q_{1-\alpha'}}(\beta/n, \mu_k)$$
$$\geq \mathcal{K}_{\mathrm{inf}}^{q_{1-\alpha'}}(\beta/n, \mu_1 - \eta) - \mathcal{K}_{\mathrm{inf}}^{q_{1-\alpha'}}(\beta/n, \mu_k)$$
$$\geq \kappa \,,$$

for some $\kappa > 0$ and thanks to the definition of the family $\mathcal{F}_{[b,+\infty]}^\alpha$. This result concludes the proof as it ensures that condition (C2) is satisfied by the QDS algorithm on $\mathcal{F}_{[b,+\infty]}^\alpha$. $\qquad\square$

## D.4   Proof of Theorem 3.7: robust regret of RDS

**Theorem 3.7** (Robust regret bound for RDS). *Let $\nu = (\nu_1, \dots, \nu_K)$ a bandit model satisfying $\nu_k \in \mathcal{F}_\ell$ for all $k$. Consider **any** increasing sequence $(\rho_n)_{n \in \mathbb{N}}$ with $\rho_n \to +\infty$, $\rho_n = o(n)$. Then, for $T$ large enough the expected number of pull of any sub-optimal arm $k$ in RDS is upper bounded by*

$$\mathbb{E}[N_k(T)] \leq n_k^{\eta, \varepsilon_0}(T) + \mathcal{O}(1) \,,$$

*where for any $\eta \in (0, 1], \varepsilon_0 > 0$, $n_k^{\eta, \varepsilon_0}(T)$ is the sequence satisfying*

$$n_k^{\eta, \varepsilon_0}(T) = \frac{\log T}{\eta(\Delta_k - \varepsilon_0)}(M_{k, n_k^{\eta, \varepsilon_0}(T)} - \mu) \,, \quad \text{with } M_{k,n} = \max\left\{ F_k^{-1}\left(\exp\left(-\frac{1}{n^2(\log n)^2}\right)\right), \rho_n\right\}.$$

*In particular, if $\rho_n = \mathcal{O}(\log n)$ then $\mathbb{E}[N_k(T)] = \mathcal{O}(\log(T) \log \log(T))$ for any light-tailed distribution $\nu_k \in \mathcal{F}_\ell$.*

*Proof.* We recall that the bonus function of RDS is $B(\mathcal{X}, \rho_n, \mu) = \mu + \frac{\rho_n}{n} \sum_{i=1}^n (\mu - X_i)^+$, as defined in equation 5, for a sequence $(\rho_n)_{n \in \mathbb{N}}$ satisfying $\rho_n \to +\infty$ and $\rho_n = o(n)$. We show that with this simple bonus conditions (C2) hold for all light-tailed distributions.

**Preliminary: concentration of the means**   We recall the definition of the family of light-tailed distributions,

$$\mathcal{F}_\ell = \{\nu \in \mathcal{F}_{(-\infty, +\infty)} : \exists \lambda_\nu > 0, \forall \lambda \in [-\lambda_\nu, \lambda_\nu], \mathbb{E}_\nu[\exp(\lambda X) < +\infty]\} \,.$$

Then, Cramér's theorem (see e.g Theorem 2.2.3 in (Dembo and Zeitouni, 2010)) ensures the condition (C1) for this family, with a good rate function that is defined with the Fenchel-Legendre transform of each distribution, itself finite thanks to the existence of the MGF of the distributions in a neighborhood of 0.

### Concentration of the DS index

We again try to find a proper set $\mathcal{B}_{k,n}$ for observations $\mathcal{X}_n = (X_1, \dots, X_n)$ that would allow to use Lemma D.1. In this setting, we show that we only need to control the sample $\mathcal{X}_n$ through its mean, the "positive gap" used in the bonus, and a range on its maximum value. Hence, we fix some $\varepsilon > 0$ and consider

$$\mathcal{B}_{k,n} = \left\{\mathcal{X} \in \mathbb{R}^n : \mu(\mathcal{X}) \leq \mu_k + \varepsilon, \mu(\mathcal{X}^+) \leq \Delta_k^+ + \varepsilon, \sigma(\mathcal{X}, \mu) \leq \sigma_{k,\mu} + \varepsilon, \bar{\mathcal{X}} \in [m_n, M_n]\right\} \,,$$

where $\mathcal{X}^+$ is the set $((\mu - X_1)_+, \dots, (\mu - X_n)_+)$, $\sigma(\mathcal{X}, \mu) = \frac{1}{n} \sum_{i=1}^n (X_i - \mu)^2$, $\bar{\mathcal{X}}$ denotes as in other sections the maximum of the set $\mathcal{X}$ and $(m_n)_{n \in \mathbb{N}}, (M_n)_{n \in \mathbb{N}}$ are two fixed sequences.

We start with the two conditions $\{\mu(\mathcal{X}) \leq \mu_k + \varepsilon\}$ and $\{\mu(\mathcal{X}^+) \leq \Delta_k^+ + \varepsilon\}$ (sharing the same $\varepsilon$ for convenience). We already proved that condition (C1) holds as $\nu_k$ is light-tailed, thanks to Cramér's theorem, but this is true also for the distribution of a random variable $(\mu - X)_+$ for $X \sim \nu_k$, as the transformed distribution is still light-tailed. Hence, thanks to Cramér's theorem there exists also a rate function $I_k^+$ satisfying $\mathbb{P}\left(\mu(\mathcal{X}_n^+) \geq \Delta_k^+ + \varepsilon\right) \leq \exp\left(-nI_k^+(\Delta_k^+ + \varepsilon)\right)$, and then

$$\sum_{n=1}^{+\infty} \left( \mathbb{P}\left( \mu(\mathcal{X}_n) \geq \mu_k + \varepsilon \right) + \mathbb{P}\left( \mu(\mathcal{X}_n^+) \geq \Delta_k^+ + \varepsilon \right) \right)$$

$$\leq \sum_{n=1}^{+\infty} \left( \exp(-nI_k(\mu_k + \varepsilon)) + \exp(-nI_k^+(\Delta_k^+ + \varepsilon)) \right)$$

$$\leq \frac{1}{1 - e^{-I_k(\mu_k + \varepsilon)}} + \frac{1}{1 - e^{-I_k^+(\Delta_k^+ + \varepsilon)}} \ .$$

We now consider the event with the quadratic sum. To handle this, we consider the Wasserstein metric $W_2$ between the empirical distribution of $\mathcal{X}$ and the true distribution $\nu_k$. First we recall the definition of this metric considering two distributions $\nu$ and $\nu'$ of real random variables

$$\mathcal{L}_p(\nu, \nu') = \inf \left\{ \int_{\mathbb{R} \times \mathbb{R}} |x - y|^p \xi(dx, dy) : \xi \in \mathcal{H}(\nu, \nu') \right\} \ ,$$

where $\mathcal{H}(\nu, \nu')$ is the set of all probability measures on $\mathbb{R} \times \mathbb{R}$ with marginals $\nu$ and $\nu'$. Then, the Wasserstein metric $W_p(\nu, \nu')$ is defined as $W_p(\nu, \nu') = \mathcal{L}_p(\nu, \nu')^{1/p}$ for $p > 1$. Two reasons motivate the use of this metric in our case: 1) concentration inequalities exist for $\mathcal{L}_p$ for light-tailed distribution, and 2) the moments of order $p$ are continuous with respect to the Wasserstein metric $W_p$ (see Theorem 6.9 in Villani (2008)). These two properties make $W_p$ a good substitute for the Levy metric we used for bounded distributions. Here we choose $W_2$ as we want to control moments of order 2, and obtain with the parameters of our problem the following concentration inequality from Fournier and Guillin (2015) (Theorem 2). Denoting $\bar{\nu}_{k,n}$ the empirical distribution of $\mathcal{X}_n$, there exist some constants $c, C$ satisfying for any $x \leq 1$

$$\mathbb{P}\left( \mathcal{L}_2(\nu_{k,n}, \nu_k) \geq x \right) \leq C \left[ \exp(-cnx^2) + \exp(-c(nx)^{\frac{1}{3}}) \right] \ . \tag{10}$$

The coefficient $1/3$ comes from choosing $\varepsilon$ as $(1 - \varepsilon)/2 = 1/3$ in the statement of the Theorem (which is different from the $\varepsilon$ in this proof). We see that this inequality is dominated by the second term. Hence, starting from our target, for any $\varepsilon > 0$, there exists $\varepsilon_1 > 0$ satisfying $W_2(\nu_{k,n}, \nu_k) \leq \varepsilon_1 \Rightarrow \sigma(\mathcal{X}, \mu) \leq \sigma_{k,\mu} + \varepsilon$. Furthermore, the series of term $\mathbb{P}(W_2(\nu_{k,n}, \nu_k) \geq \varepsilon_1)$ converges thanks to Equation 10, which concludes the part of the proof corresponding to this term.

Now that the events about sample means are handled, we investigate possible values for the sequence $m_n$ and $M_n$ that would allow $\mathcal{B}_{k,n}$ to happen with high probability. The maximum $\bar{\mathcal{X}}_n$ of a set of $n$ i.i.d random variables $\mathcal{X}_n = (X_1, \ldots, X_n)$ has an explicit distribution, which is (in terms of the cdf $F_k$ of $\nu_k$) for any $x \in \mathbb{R}$,

$$\mathbb{P}_{\mathcal{X}_n \sim \nu_k^n}(\bar{\mathcal{X}}_n \leq x) = F_k(x)^n \ .$$

We first look at the term $M_n$, we calibrate it to ensure that $\mathbb{P}(\bar{\mathcal{X}}_n \leq M_n) \geq 1 - \frac{1}{n \log(n)^2}$, so that

$$M_n = F_k^{-1} \left( \left( 1 - \frac{1}{n(\log n)^2} \right)^{\frac{1}{n}} \right) \leq F_k^{-1} \left( \exp\left( -\frac{1}{n^2 (\log n)^2} \right) \right) \ .$$

This way, $\sum \mathbb{P}(\bar{\mathcal{X}}_n \leq M_n) \geq 1 - \frac{1}{n \log(n)^2}$ converges. Then we consider $m_n$, and this time we want $\mathbb{P}(\bar{\mathcal{X}}_n \leq m_n) \leq \frac{1}{n \log(n)^2}$ to ensure the same convergence guarantees. We obtain

$$m_n = F_k^{-1} \left( \frac{1}{n(\log n)^2}^{\frac{1}{n}} \right) = F_k^{-1} \left( \exp\left( -\frac{\log n + 2 \log \log n}{n} \right) \right) \ .$$

Combining all these results, we obtain

$$\sum_{n=1}^{T-1} \mathbb{P}_{\mathcal{X}_n \sim \nu_k^n}(\mathcal{X}_n \notin \mathcal{B}_{k,n}) = \mathcal{O}(1) \ .$$

We now use the first part of Lemma C.1 and the fact that for any $\eta \in [0,1)$ and $x \in (-\infty, \eta]$, $-\log(1-x) \leq x + \frac{1}{1-\eta}\frac{x^2}{2}$. Denoting $M_{\mathcal{X}_n} = \max\left(\bar{\mathcal{X}}_n, B(\mathcal{X}_n, \rho_n, \mu)\right)$, $X_{n+1} = B(\mathcal{X}_n, \rho_n, \mu)$ and using the representation of Dirichlet samples as normalized exponential variables, Chernoff inequality provide

$$
\begin{aligned}
\mathbb{P}_{w \sim \mathcal{D}_{n+1}} &\left( \sum_{i=1}^{n} w_i X_i + w_{n+1} B(\mathcal{X}_n, \rho_n, \mu) \geq \mu \right) \\
&= \mathbb{P}_{R_1,\ldots,R_{n+1} \sim \mathcal{E}(1)} \left( \sum_{i=1}^{n+1} R_i(X_i - \mu) \geq 0 \right) \\
&\leq \inf_{\lambda \in [0, \frac{\eta}{M_{\mathcal{X}_n} - \mu})} \prod_{i=1}^{n+1} \mathbb{E}_{R_i \sim \mathcal{E}(1)} \left[ e^{\lambda R_i (X_i - \mu)} \right] \\
&\leq \exp\left( -\sum_{i=1}^{n+1} \log\left( 1 - \eta \frac{X_i - \mu}{M_{\mathcal{X}_n} - \mu} \right) \right) \\
&\leq \exp\left( -\sum_{i=1}^{n} \log\left( 1 - \eta \frac{X_i - \mu}{M_{\mathcal{X}_n} - \mu} \right) - \log(1-\eta) \right) \\
&\leq \frac{1}{1-\eta} \exp\left( -\sum_{i=1}^{n} \log\left( 1 - \eta \frac{X_i - \mu}{M_{\mathcal{X}_n} - \mu} \right) \right) \\
&\leq \frac{1}{1-\eta} \exp\left( \sum_{i=1}^{n} \left( \eta \frac{X_i - \mu}{M_{\mathcal{X}_n} - \mu} + \frac{\eta^2}{2(1-\eta)} \left( \frac{X_i - \mu}{M_{\mathcal{X}_n} - \mu} \right)^2 \right) \right) \\
&= \frac{1}{1-\eta} \exp\left( -n\eta \frac{\bar{\Delta}_n}{M_{\mathcal{X}_n} - \mu} + n \frac{\eta^2}{2(1-\eta)} \frac{\bar{\sigma}_n(\mu)^2}{(M_{\mathcal{X}_n} - \mu)^2} \right),
\end{aligned}
$$

where $\bar{\Delta}_n = \frac{1}{n}\sum_{i=1}^{n} \mu - X_i$, $\bar{\sigma}_n^2(\mu) = \frac{1}{n}\sum_{i=1}^{n}(X_i - \mu)^2$.

We recall that we consider this upper bound under the event $\mathcal{X}_n \in \mathcal{B}_{k,n}$, which ensures that 1) $\bar{\mathcal{X}}_n \in [m_n, M_n]$ with the sequences we defined, 2) $\bar{\Delta}_n \geq \mu - \mu_k + \varepsilon$, 3) the bonus is upper bounded by $\mu + \rho_n \times (\Delta_k^+ + \varepsilon)$, and 4) the quadratic deviation satisfies $\bar{\sigma}_n(\mu) \leq \sigma_{k,\mu} + \varepsilon$. For any $\varepsilon_0 > 0$, if we further assume that $M_n = o(m_n^2)$, for any $n$ large enough these results finally provide

$$
\begin{aligned}
\mathbb{P}\left( \widetilde{\mu}(\mathcal{X}_n, \mu) \geq \mu \right) &\leq \frac{1}{1-\eta} \exp\left( -n\eta \frac{\Delta_k - \varepsilon}{\max(M_n, B_n) - \mu} + n \frac{\eta^2}{2(1-\eta)} \frac{(\sigma_{k,\mu} + \varepsilon)^2}{(m_n - \mu)^2} \right) \\
&\leq \frac{1}{1-\eta} \exp\left( -n\eta \frac{\Delta - \varepsilon_0}{\max(M_n, B_n) - \mu} \right),
\end{aligned}
$$

where $B_n = \mu + \rho_n \left( \mathbb{E}_{\nu_k}[(\mu - X)_+] + \varepsilon \right)$. The condition $M_n = o(m_n^2)$ is satisfied for light-tailed distributions, as they generally have at most a poly-logarithmic growth of the maximum (e.g $\log(n)$ for exponential tails, $\sqrt{\log n}$ for gaussian tails, ... ) and so $M_n$ and $m_n$ are actually of the same order of magnitude. We then recover all the terms of Theorem 3.7 by matching the exponent of the upper bound with $-\log T$.

To conclude this part, the light-tailed hypothesis allows to provide an asymptotic upper bound on the expected number of pulls of each sub-optimal arm for the RDS index. Then, the choice of $m_k(T)$ in Lemma D.1 can be $m_k(T) = O(\log(T)M_{\log(T)})$ if $M_n$ is a power of log. The algorithm then achieves asymptotically a robust sub-linear instance dependent regret.

**Remark D.3.** *The concentration bound we use on the Dirichlet weighted average requires the control of the second empirical decentred moment $\bar{\sigma}_n^2(\mu)$ since we use $-\log(1-x) \leq x + \frac{1}{1-\eta}\frac{x^2}{2}$. This control*

*follows from the existence of exponential moments i.e $\nu_k \in \mathcal{F}_\ell$. A tighter analysis is possible, indeed for any $q > 0$ and $\eta \in (0,1)$ there exists $C_{q,\eta} > 0$ such that $\forall x \le \eta, -\log(1-x) \le x + C_{q,\eta}|x|^{1+q}$ ($C_{1,\eta} = \frac{1}{1-\eta}$), relaxing the requirement to a mere control of the moment of order $1+q$ in the topology of $W_{1+q}$ (for which concentration results are similar to the one we provide, for the families we consider). Furthermore, it would relax the condition relating $m_n$ and $M_n$ to $M_n = o(m_n^{1+q})$.*

### Condition (C2)

We use the left-hand term of Lemma 3.3 and obtain a lower bound of the BCP in $e^{-\frac{n}{\rho_n}}$. Combining this result with condition (C1) we obtain

$$E_n \le e^{-n(I_1(\mu) - 1/\rho_n)} \ ,$$

and for $n$ large enough $\rho_n > 1/I_1(\mu)$, which is sufficient to obtain the convergence of $\sum_{n=1}^{+\infty} E_n$. If we choose a sequence $(\rho_n)$ that is strictly increasing and of first term $\rho_1$, we see that if $I_1(\mu) > \rho_1$ then the term $E_n$ is exponentially decreasing from the start.

$\square$

## E   Examples of distributions fitting the family of QDS

We first show a given distribution $\nu$ can always be fitted in $\mathcal{F}_{[b,+\infty)}^\alpha$ at the cost of a higher exploration bonus $\rho$, thus satisfying the quantile condition of Theorem 3.5.

**Lemma E.1.** *Let $\mathcal{F} \subset \mathcal{F}_{[b,+\infty)}$ a base family of distributions with continuous cdf and $\alpha \in (0,1)$. For all $\nu \in \mathcal{F}$ and $\mu > \mathbb{E}_\nu[X]$, there exists $\rho > 0$, $\mathfrak{M} = \mathfrak{M}(\rho)$ such that $\mathcal{K}_{\inf}^{\mathfrak{M}}(\mathcal{T}_\alpha(\nu),\mu) \le \mathcal{K}_{\inf}^{\mathcal{F}}(\nu,\mu)$.*

*Proof.* Let $\mathfrak{M} = \max\left\{\mathrm{CVaR}_\alpha(\nu), \mu + \rho\mathbb{E}_\nu\left[(\mu - X)_+\right]\right\}$. By construction, the support of $\mathcal{T}_\alpha(\nu)$ is upper bounded by $\mathfrak{M}$ and $\mu < \mathfrak{M}$, therefore it follows from Theorem 8 in Honda and Takemura (2010) that $\mathcal{K}_{\inf}^{\mathfrak{M}}(\mathcal{T}_\alpha(\nu),\mu) = \max_{\lambda \in [0, \frac{1}{\mathfrak{M}-\mu}]} \mathbb{E}_{\mathcal{T}_\alpha(\nu)}\left[\log(1 - \lambda(X - \mu))\right]$. It follows from the concavity of $\log$ that, for $\lambda \in [0, \frac{1}{\mathfrak{M}-\mu}]$,

$$\begin{aligned}
\mathbb{E}_{\mathcal{T}_\alpha(\nu)}\left[\log(1 - \lambda(X-\mu))\right] &\le -\mathbb{E}_{\mathcal{T}_\alpha(\nu)}\left[\lambda(X-\mu)\right] \\
&= \lambda\left(\mu - \mathbb{E}_{\mathcal{T}_\alpha(\nu)}[X]\right) \\
&= \lambda\left(\mu - \mathbb{E}_\nu\left[X\mathbb{1}_{X \le q_{1-\alpha}(\nu)}\right] - \mathbb{E}_{\mathcal{T}_\alpha(\nu)}\left[X\mathbb{1}_{X > q_{1-\alpha}(\nu)}\right]\right) \\
&= \lambda\left(\mu - \mathbb{E}_\nu\left[X\mathbb{1}_{X \le q_{1-\alpha}(\nu)}\right] - \alpha C_\alpha(\nu)\right) \\
&= \lambda\left(\mu - \mathbb{E}_\nu\left[X\mathbb{1}_{X \le q_{1-\alpha}(\nu)}\right] - \alpha\mathbb{E}_\nu\left[X | X > q_{1-\alpha}(\nu)\right]\right) \\
&= \lambda\left(\mu - \mathbb{E}_\nu\left[X\mathbb{1}_{X \le q_{1-\alpha}(\nu)}\right] - \mathbb{E}_\nu\left[X\mathbb{1}_{X > q_{1-\alpha}(\nu)}\right]\right) \\
&= \lambda\left(\mu - \mathbb{E}_\nu[X]\right) \ ,
\end{aligned}$$

by definition of $C_{1-\alpha}(\nu)$ and the conditional expectation. Since $\mu > \mathbb{E}_\nu[X]$, the maximum of the RHS in $\lambda$ is attained at the rightmost point, which yields

$$\mathcal{K}_{\inf}^{\mathfrak{M}}(\mathcal{T}_\alpha(\nu),\mu) \le \frac{\mu - \mathbb{E}_\nu[X]}{\mathfrak{M} - \mu}.$$

For $\rho > 0$ large enough, we have $\mathfrak{M} = \mu + \rho\mathbb{E}_\nu\left[(\mu - X)^+\right]$ which further simplifies as

$$\mathcal{K}_{\inf}^{\mathfrak{M}}(\mathcal{T}_\alpha(\nu),\mu) \le \frac{\mu - \mathbb{E}_\nu[X]}{\rho\mathbb{E}_\nu\left[(\mu - X)_+\right]}.$$

Therefore for $\rho$ large enough, in particular $\rho \ge \frac{\mu - \mathbb{E}_\nu[X]}{\mathbb{E}_\nu[(\mu-X)_+]\mathcal{K}_{\inf}^{\mathcal{F}}(\nu,\mu)}$, we have

$$\mathcal{K}_{\inf}^{\mathfrak{M}}(\mathcal{T}_\alpha(\nu),\mu) \le \mathcal{K}_{\inf}^{\mathcal{F}}(\nu,\mu).$$

$\square$

The bound on $\rho$ given in the above lemma can be rather loose because of the crude concave inequality we use. It also comes at the price of increasing $\rho$, which may hurt the performances of QDS due to overexploration. We now show that this quantile condition can be calculated almost in closed-form and is naturally satisfied by some classical families of distributions.

### E.1 Exponential

Let $\mathcal{F} = (\nu_\theta)_{\theta \in \mathbb{R}_+^*}$ with density $p_\theta(x) = \theta e^{-\theta x} \mathbb{1}_{x \geq 0}$. We summarize in the below lemma a number of explicit formulas for the $\mathcal{K}_{\inf}$ operators and quantiles of $\nu_\theta$.

**Lemma E.2** (Some statistics of exponential distributions). *Let* $0 < \phi < \theta$ *and* $\alpha \in (0, 1)$.

(i) $\mathbb{E}_\theta[X] = \frac{1}{\theta}$.

(ii) $q_{1-\alpha}(\nu_\theta) = -\frac{\log \alpha}{\theta}$.

(iii) $\mathbb{E}_\theta[X | X \geq q_{1-\alpha}(\nu_\theta)] = \frac{1}{\theta} + q_{1-\alpha}(\nu_\theta)$.

(iv) $\mathbb{E}_\theta \left[ \left( \frac{1}{\phi} - X \right)_+ \right] = \frac{1}{\phi} - \frac{1}{\theta} \left( 1 - e^{-\theta/\phi} \right)$.

(v) $\mathcal{K}_{\inf}^{\mathcal{F}}(\nu_\theta, \frac{1}{\phi}) = \frac{\phi}{\theta} - \log \frac{\phi}{\theta} - 1$.

*Proof.* (i)-(iv) result from straightforward integral calculations. (v) is a direct consequence of $\mathcal{F}$ being a SPEF, which implies $\mathcal{K}_{\inf}^{\mathcal{F}}(\nu_\theta, \frac{1}{\phi}) = KL(\nu_\theta, \nu_\phi)$, which has the stated closed-form for exponential distributions. $\square$

Using these formulas, we numerically compute $\mathcal{K}_{\inf}^{\mathfrak{M}(\rho)}$ as a function of $\rho$ by solving the convex dual problem (see Honda and Takemura (2010)) and compare it to $\mathcal{K}_{\inf}^{\mathcal{F}}$. Conversely, for a fixed exploration bonus $\rho$, we compute the $\mathcal{K}_{\inf}$ of the truncated distribution $\mathcal{T}_\alpha(\nu_\theta)$ for a range of $\alpha$. As per intuition, smaller values of $\alpha$, corresponding to smaller truncations of the support, help satisfy the $\mathcal{K}_{\inf}$ condition. Results are reported in Figure 3.

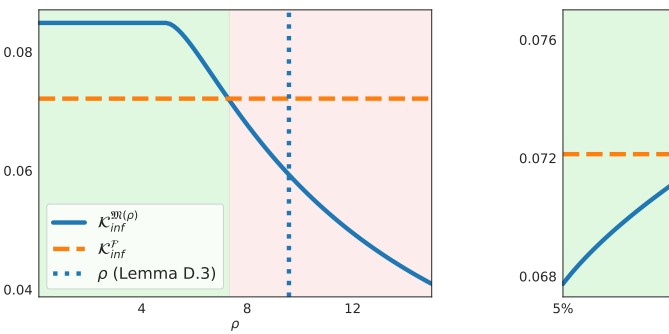 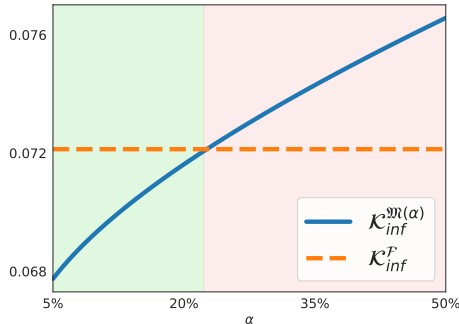

Figure 3: Comparison of $\mathcal{K}_{\inf}^{\mathfrak{M}(\rho,\alpha)}(\mathcal{T}_\alpha(\nu_\theta), \frac{1}{\phi})$ and $\mathcal{K}_{\inf}^{\mathcal{F}}(\nu_\theta, \frac{1}{\phi})$ for the exponential distribution $\mathcal{E}(\theta)$ with $\theta = \frac{1}{2}$, $\phi = \frac{1}{3}$. Left: $\alpha = 5\%$. Right: $\rho = 8$. Admissible regions for $\rho$ and $\alpha$ are shaded in green.

### E.2 Gaussian

Let $\sigma > 0$ and $\mathcal{F}_\sigma = (\nu_\theta)_{\theta \in \mathbb{R}}$ with density $p_\theta(x) = \frac{1}{\sqrt{2\pi}\sigma} e^{-\frac{(x-\theta)^2}{2\sigma^2}}$. We recall some useful statistics of the SPEF of fixed variance Gaussian distributions.

**Lemma E.3** (Some statistics of fixed variance Gaussian distributions). *Let $\theta < \phi$ and $\alpha \in (0,1)$. We denote $\Phi(x) = \frac{1}{\sqrt{2\pi}} \int_{-\infty}^{x} e^{-\frac{y^2}{2}} dy$ the standard Gaussian cdf.*

  *(i)* $\mathbb{E}_\theta[X] = \theta$.

  *(ii)* $q_{1-\alpha}(\nu_\theta) = \theta + \sigma \Phi^{-1}(1-\alpha)$.

  *(iii)* $\mathbb{E}_\theta[X|X \geq q_{1-\alpha}(\nu_\theta)] = \theta + \frac{\sigma}{\alpha\sqrt{2\pi}} e^{-\frac{\Phi^{-1}(1-\alpha)}{2}}$.

  *(iv)* $\mathbb{E}_\theta[(\phi - X)_+] = (\phi - \theta) \Phi\left(\frac{\phi-\theta}{\sigma}\right) + \frac{\sigma}{\sqrt{2\pi}} e^{-\frac{(\phi-\theta)^2}{2\sigma^2}}$.

  *(v)* $\mathcal{K}_{\inf}^{\mathcal{F}_\sigma}(\nu_\theta, \phi) = \frac{(\phi-\theta)^2}{2\sigma^2}$.

The proof is similar to that of the previous lemma; in particular *(v)* uses the fact that $\mathcal{F}_\sigma$ forms a SPEF. Results are reported in Figure 4. The lighter right tail of Gaussian distributions, compared to that of exponential distributions, results in much less stringent conditions on $\alpha$ and $\rho$; in other words, Gaussian distributions are "easier" to summarize with the truncation and conditional Value-at-Risk operator.

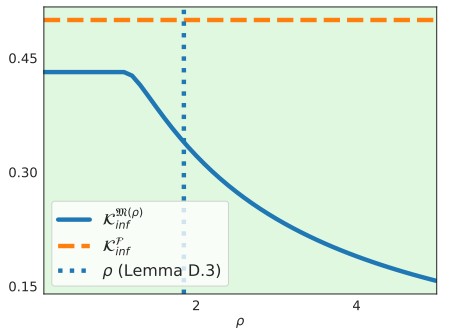
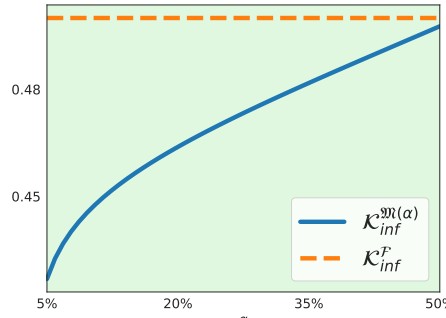

Figure 4: Comparison of $\mathcal{K}_{\inf}^{\mathfrak{M}(\rho,\alpha)}(\mathcal{T}_\alpha(\nu_\theta), \phi)$ and $\mathcal{K}_{\inf}^{\mathcal{F}}(\nu_\theta, \phi)$ for the Gaussian distribution $\mathcal{N}(\theta, \sigma^2)$ with $\theta = 0$, $\sigma = 1$, $\phi = 1$. Left: $\alpha = 5\%$. Right: $\rho = 1$. Admissible regions for $\rho$ and $\alpha$ are shaded in green.

# F    Additional Experiments

We present in this section various experimental results for the DS algorithms. First, we expand on the empirical analysis of the DSSAT bandit problem introduced in Section 4 and compare various competitor algorithms. In particular, we show that each of the three settings we introduced, namely bounded with unknown but detectable upper bound (BDS), unbounded with a quantile condition (QDS) and robust (RDS), are eligible assumptions for simulated grain yields, providing new modeling tools for the practitioner.

Going further, we detail another setting for which QDS and RDS apply but that is outside the scope of SPEF algorithms: the Gaussian mixtures distributions. This may be of practical relevance as Gaussian mixtures can be used to estimate arbitrary densities.

We also test the sensitivity of the DS algorithms to their exploration bonus. For BDS, we consider a toy bandit problem with uniform arms and provide heuristics to tune the hyperparameter $\rho$ that complete the theoretical recommendations. For RDS, we follow the experimental setting of the robust UCB for light-tailed distributions presented in Ashutosh et al. (2021) and show the superiority of the Dirichlet Sampling approach for different bonus functions $\rho_n$.

Finally, we note that a by-product of our analysis of Boundary Crossing Probabilities provides an asymptotic result on the empirical $\mathcal{K}_{\inf}$ operator, at the root of the regret analysis of Dirichlet Sampling. We illustrate it on a few classical families of distributions.

### F.1 Summary of competitor algorithms

We first present in Table 2 details on the algorithms we study in this section, with the hypothesis they make on the distributions of the arms and the knowledge they require. This is a non-exhaustive list, and for a more detailed (but still non-exhaustive) list we refer the reader to Section 1. Except for UCB1 and Binarized TS (which are classical benchmarks), we choose the other competitors because they target asymptotic optimality in the families of distributions they consider.

Table 2: Comparison of competitor bandit algorithms matching the Burnetas & Katehakis bound for various assumptions on the arm distribution $\nu$. Elements listed as parameters are considered prior knowledge and are used within the algorithm.

| Algorithm | Scope for optimality | Algorithm parameters |
|---|---|---|
| UCB1 Auer et al. (2002) | $\sigma$-sub-Gaussian (not optimal) | $\sigma$ |
| Binarized TS Agrawal and Goyal (2012) | $\mathrm{Supp}(\nu) \subset [b, B]$ (not optimal) | $b$, B |
| kl-UCB Cappé et al. (2013) | SPEF $(\nu_\theta)_{\theta \in \Theta}$ | $\mathrm{KL}(\nu_\theta, \nu_{\theta'})$ |
| Empirical KL-UCB Cappé et al. (2013) | $\mathrm{Supp}(\nu) \subset (b, B]$ | $B$ |
| IMED Honda and Takemura (2015) | SPEF $(\nu_\theta)_{\theta \in \Theta}$ $\mathbb{E}_{X \sim \nu}[e^{\lambda X}] < +\infty$ | $\mathrm{KL}(\nu_\theta, \nu_{\theta'})$ |
| Empirical IMED Honda and Takemura (2015) | $\mathrm{Supp}(\nu) \subset (-\infty, B]$ $\mathbb{E}_{X \sim \nu}[e^{\lambda X}] < +\infty$ | $B$ |
| RB-SDA Baudry et al. (2020) | SPEF $(\nu_\theta)_{\theta \in \Theta}$ | Non-parametric |
| TS Thompson (1933) Korda et al. (2013) | SPEF $(\nu_\theta)_{\theta \in \Theta}$ | Suitable SPEF prior/posterior |
| NPTS Riou and Honda (2020) | $\mathrm{Supp}(\nu) \subset [b, B]$ | $B$ |

### F.2 DSSAT bandit

In this section, we present additional competitors (see Table 2) on the DSSAT bandit problem, a 7-armed stochastic bandit where each arm corresponds to a simulated dry grain yield for a given planting date (see Figure 5).

Assuming yield distributions are bounded, one can use classical algorithms such as UCB1 (Auer et al., 2002) or Thompson Sampling with Beta prior using the binarization trick introduced in (Agrawal and Goyal, 2012). These algorithms enjoy logarithmic regret without the optimal rate of (Burnetas and Katehakis, 1996).

Other bounded algorithms include empirical IMED (Honda and Takemura, 2015) and NPTS (Riou and Honda, 2020). The former is based on the calculation of $\mathcal{K}_{\inf}$ indices inspired by the Burnetas-Katehakis lower bound. We distinguish IMED, which relies on a SPEF assumption to explicitly compute the $\mathcal{K}_{\inf}$ and therefore falls short of the scope of DSSAT, from empirical IMED, which solves the convex optimization problems defined by the $\mathcal{K}_{\inf}$ of the empirical distribution of each arm and requires boundedness.

Such bounded algorithms require the explicit knowledge of an upper bound on the support of the arms distributions. To represent the fact that a tight bound is sometimes unknown to the practitioner (uncertain environment, possibility of yet unobserved black swan events...) we run two variants of the above algorithms, one with the exact maximum yield across all simulated data, which we believe is a strong prior information, and one with the same bound inflated by $50\%$, which we deem a conservative estimate.

Finally, RB-SDA (Baudry et al., 2020) is a recent sub-sampling algorithm based on a similar round-based structure as our DS algorithms. Its optimality is only established under tail conditions satisfied by some SPEF; in particular, despite its appealing regret growth on this specific instance of DSSAT,

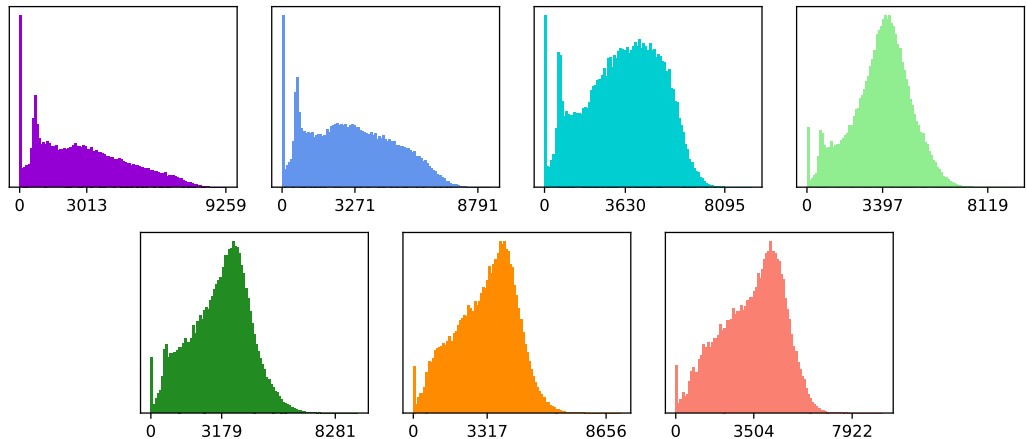

Figure 5: Distribution of simulated dry grain yield (kg/ha) for seven different planting dates over $10^6$ samples. Reported on the x-axis are the distribution minimum, mean and maximum values. The optimal arm is the third one (mean 3630 kg/ha).

it enjoys none of the theoretical guarantees of the previous algorithms and is shown for empirical comparison only. Note that although it has been analyzed under strong parametric assumptions, the algorithm itself is non-parametric, and in particular is agnostic to the choice of the upper bound.

We run the three instances of DS algorithms, which we believe capture three aspects of the DSSAT problem that are of practical interest: the boundedness without the need to know the bound a priori (BDS), the robustness in face of unknown distributions (RDS) or the assumption that the conditional Value-at-Risk at some defined level $\alpha$ is a meaningful summary of tail statistics (QDS). All three compares similarly with the other optimal algorithms using the exact upper bound, RDS being the overall winner, and significantly outperform their conservative counterparts.

Results are reported in Figure 6 and Table 3. As expected, UCB1 and binarized TS perform poorly on the DSSAT problem, hinting that this particular bandit instance is not easy and requires more sophisticated methods. RB-SDA achieves good performances but exhibits larger dispersion than other methods (95% quantile is $0.99 \times 10^6$, standard deviation is $0.26 \times 10^6$), meaning that some runs in the Monte Carlo simulations suffer high regret; we interpret this as evidence that RB-SDA operates outside its theoretical scope here and is therefore not backed by strong regret guarantees. The DS algorithms achieve similar or even slightly better regret than empirical IMED and NPTS using the prior knowledge of the true upper bound. However the latter two suffer from using the conservative upper bound in place of the true bound. Note that contrary to RB-SDA, empirical IMED and NPTS remain theoretically sound in both the prior knowledge and conservative case, but the larger bound drives the exploration-exploitation balance towards more exploration than is optimal. Finally, the tuning of the $\rho$ bonus in the DS algorithms is done using plausible heuristics (see Appendix F.4) and have not been optimized to suit this particular problem.

### F.3 Gaussian Mixture

Many real-world situations (loss profile of a portfolio of financial assets, crop yields, statistics of heterogeneous populations...) exhibit multimodal distributions. The Gaussian mixture model is perhaps the simplest example of such distributions and is ubiquitous in many areas of machine learning and engineering (speech recognition, clustering...), in particular as a nonparametric model for kernel density estimation. Still, to the best of our knowledge, it escapes the scope of current optimal bandit methods as it is neither bounded nor SPEF. Thanks to the different sets of assumptions in which they operate, both RDS and QDS are eligible algorithms to tackle the problem of sequential decision-making in a Gaussian mixture environment, at the cost of slightly larger-than-logarithmic regret and slightly lower $\mathcal{K}_{\inf}$ rate respectively.

We consider two arms distributed as a 50%-50% independent mixture of $\mathcal{N}(-0.3, 0.5^2)$ and $\mathcal{N}(1.3, 0.5^2)$ and a 10%-80%-10% independent mixture of $\mathcal{N}(-1.5, 0.5^2)$, $\mathcal{N}(0.6, 0.5^2)$ and

Table 3: Regret on DSSAT bandit at $T = 10^4$, over $N = 5000$ independent simulations. Scale $= 10^6$.

| Algorithm | 5% quantile | Mean ($\pm$ standard deviation) | 95% quantile |
|---|---|---|---|
| UCB1 | 1.56 | $1.74 \pm 0.11$ | 1.92 |
| UCB1 (conservative) | 2.00 | $2.13 \pm 0.08$ | 2.26 |
| TS Binarization | 0.72 | $1.36 \pm 0.46$ | 2.20 |
| TS Binarization (conservative) | 0.94 | $1.70 \pm 0.50$ | 2.57 |
| Empirical IMED | 0.23 | $\mathbf{0.38} \pm 0.13$ | **0.58** |
| Empirical IMED (conservative) | 0.29 | $0.42 \pm 0.10$ | 0.60 |
| NPTS | 0.30 | $0.46 \pm 0.14$ | 0.69 |
| NPTS (conservative) | 0.39 | $0.55 \pm 0.13$ | 0.77 |
| RB-SDA | **0.20** | $0.42 \pm 0.26$ | 0.99 |
| BDS | 0.31 | $0.47 \pm 0.13$ | 0.68 |
| RDS | 0.22 | $\mathbf{0.38} \pm 0.16$ | 0.63 |
| QDS | 0.25 | $0.41 \pm 0.14$ | 0.64 |

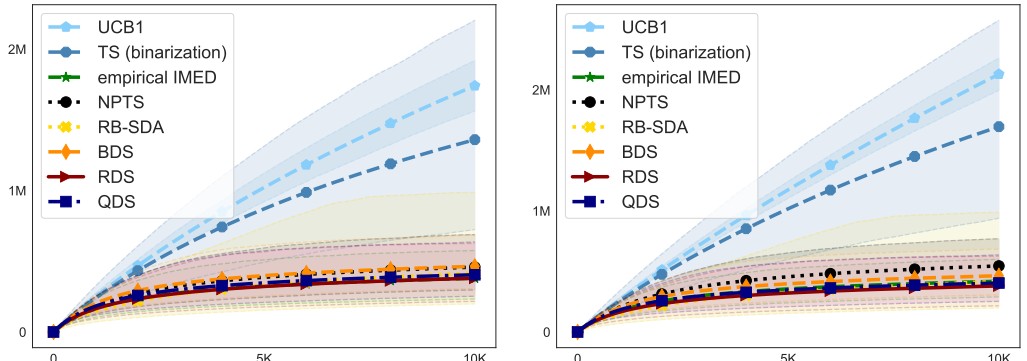

Figure 6: Average regret on 5000 simulations and horizon $T = 10^4$. Dashed lines correspond to 5%-95% regret quantiles. UCB1, Binarized Thompson Sampling, Empirical IMED and NPTS are run with exact upper bounds around $1.5 \times 10^4$ kg/ha (left) and the conservative upper bound $1.5 \times 10^4$ kg/ha (right). BDS: $\rho = 4$. RDS: $\rho_n = \sqrt{\log(1+n)}$. QDS: $\rho = 4, \alpha = 5\%$.

$\mathcal{N}(2.5, 0.5^2)$. Note that both mixtures have total variance equal to $0.5^2$. Due to the lack of theoretically grounded benchmark, we run three SPEF algorithms (kl-UCB, IMED and Thompson Sampling) assuming the arms belong to the SPEF of Gaussian distributions with fixed variance $0.5^2$. This is an example of *model misspecification*.

We run RDS with $\rho_n = \sqrt{\log(1+n)}$, which matches the asymptotic growth rate of the maximum of i.i.d Gaussian samples, and QDS with $\alpha = 5\%, \rho = 4$; we recall that Appendix E shows empirical evidence that the quantile condition required by QDS holds for a large variety of $\alpha$ and $\rho$ in the case of Gaussian tails. Note that the use of QDS in this context is technically out of scope of Theorem 3.5 since Gaussian mixtures are not lower bounded; we believe however that this is an artifact of our proof technique that could lifted with a finer analysis.

Results are reported in Figure 7. Both RDS and QDS outperform other existing methods; in particular, among the misspecified SPEF algorithms, only IMED exhibit comparable regret growth. As this bandit problem is complicated (small optimality gap, non-SPEF distributions), all algorithms have a relatively large variance.

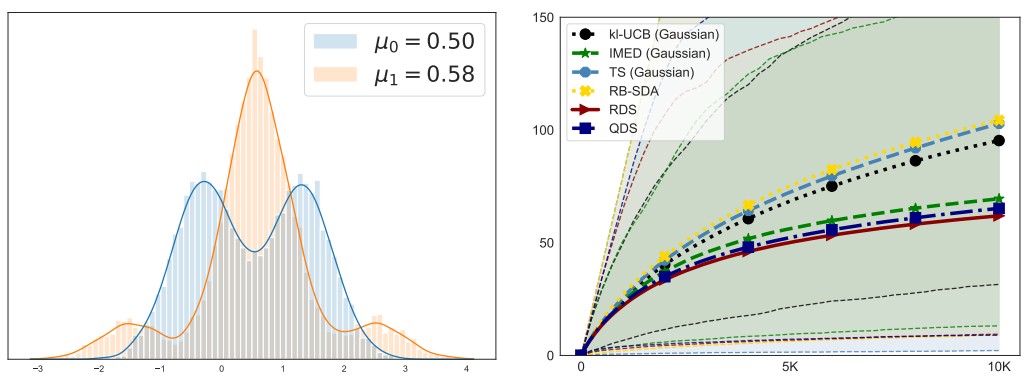

Figure 7: Left: Gaussian mixture arms ($10^4$ samples each). Right: average regret on 5000 simulations and horizon $T = 10^4$. kl-UCB, IMED and Thompson Sampling are run assuming Gaussian arms with same variance as the mixtures. RDS: $\rho_n = \sqrt{\log(1+n)}$. QDS: $\rho = 4, \alpha = 5\%$.

### F.4 BDS parameters sensitivity

We study the sensitivity of BDS to its parameter $\rho$. Theorem 3.4 suggests to scale the exploration bonus $B_{\rho,\gamma}$ as $\rho = -1/\log(1-p)$, which is a proxy of an upper bound of $1/(1-F(\mu_1))$ in Lemma 3.2. We believe this bonus to be rather conservative when $p$ is small and the distributions considered exhibit little skewness; as an example, if a distribution is such that at most $25\%$ of its mass is located to the right of the optimal mean reward $\mu^*$, $\rho \approx 4$ should be a suitable tuning.

To investigate this, we consider a toy bandit instance with two arms following uniform distributions on $[0, 1]$ and $[0.2, 0.9]$ respectively (note that the upper bound is different for each arm yet the distribution of mass near their respective bounds is the same, thus fitting the setting of BDS). These distributions are shown in Figure 8, and in particular their means are $0.5$ and $0.55$ respectively. For $\gamma = 0.1$, we compute the expected regret of BDS obtained with the theoretical tuning $\rho = -1/\log(1-p) \simeq 9.5$, and compare it with other choices of $\rho$. Figure 8 shows that only the most extreme tuning $\rho = 50$ exhibits significant, albeit still sublinear, regret. Small deviations from the theoretical tuning yields similar regret, the heuristic $\rho = 4$ discussed above being slightly better, which tends to confirm our belief that the analysis of Theorem 3.4 can be sharpened. Note that the exploration incentive given by $\rho$ is necessary since smaller values (e.g $\rho = 0.1$) tends to accumulate more regret.

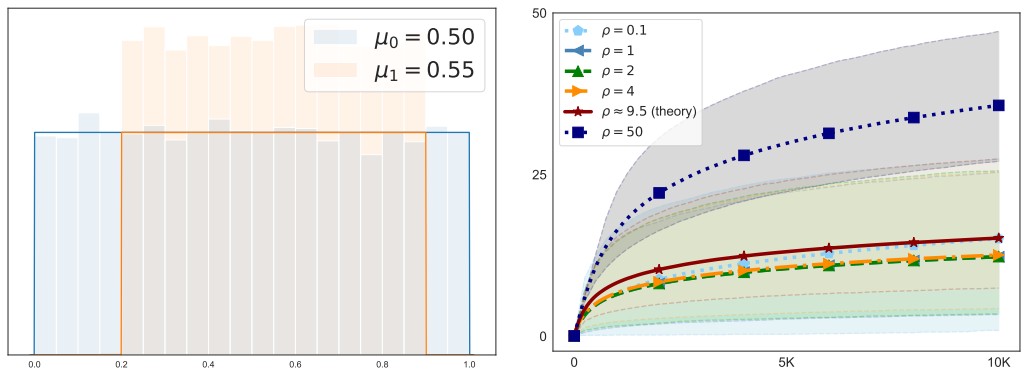

Figure 8: Left: bandit with two uniform arms $\mathcal{U}(0, 1)$ and $\mathcal{U}(0.2, 0.9)$ ($10^4$ samples each). Right: average regret on 5000 simulations and horizon $T = 10^4$ of BDS for various values of $\rho$.

### F.5 Robustness for light-tailed bandits: comparison with R-UCB-LT

The study of statistically robust bandit algorithms is fairly recent, and as such is yet to have well-established benchmarks. Ashutosh et al. (2021) introduce R-UCB-LT, an adaptation of the standard sub-Gaussian UCB to enforce robustness w.r.t light-tailed distributions (as defined in Appendix A.1).

We reproduce the setting of their experiment, namely two Gaussian arms $\mathcal{N}(1,1)$ and $\mathcal{N}(2,3)$, and compare several variants of both R-UCB-LT and RDS against a misspecified UCB1 (the misspecification takes the form of an overly optimistic 1-sub-Gaussian assumption, while the second arm is only $\sqrt{3}$-sub-Gaussian). Both R-UCB-LT and RDS rely on a slowly growing exploration bonus, denoted respectively by $f$ and $\rho$; we run both algorithms with $f$ and $\rho$ equal to $\log^2$, $\log$ and $\sqrt{\log}$.

Results are reported in Figure 9. As expected, the misspecified UCB1 exhibits much faster regret growth than the robust algorithms. However, RDS seems to outperform R-UCB-LT, the best average regret being achieved by RDS with $\rho_n = \sqrt{\log(1+n)}$ and $\rho_n = \log(1+n)$. Furthermore, the regret to RDS appears to be somewhat monotonic (slightly increasing) with respect to the hyperparameter $\rho$, and the best results are achieved by the one matching the asymptotic growth rate of the maximum of a i.i.d Gaussian samples, as recommended by Theorem 3.7. On the other hand, the best version of R-UCB-LT is obtained with $f \approx \log$ (for which we do not find a theoretical intuition) and the performance gap is significant when other bonuses are considered. We also tested R-UCB-LT with powers of $\log \log$ with similar results; we do not report these curves for the readability of the figures. In light of these results, RDS seems less sensitive to its parameter choice than R-UCB-LT, which is another sort of robustness guarantee.

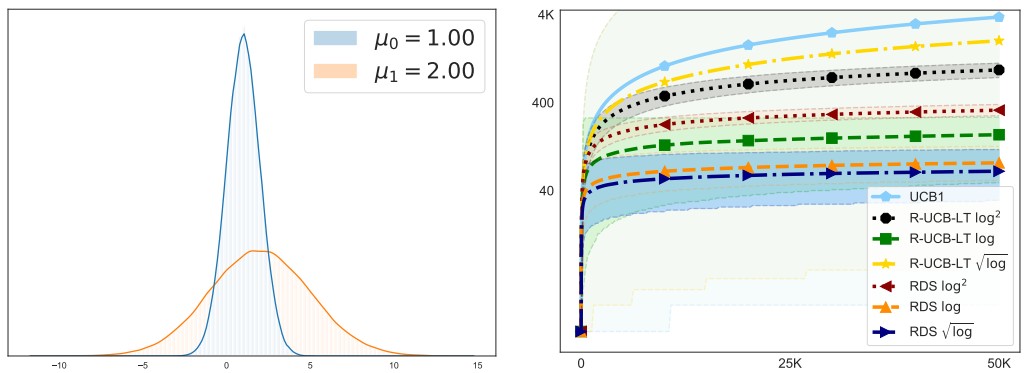

Figure 9: Left: Gaussian arms $\mathcal{N}(1,1)$ and $\mathcal{N}(2,3)$ ($5 \times 10^4$ samples each). Right: average regret (in log scale) on 5000 simulations and horizon $T = 5 \times 10^4$. UCB1 is run assuming a 1-sub-Gaussian instance.

## F.6    Asymptotic behavior of empirical $\mathcal{K}_{\text{inf}}$

A critical part of the regret analysis of DS algorithms relies on the control of the BCP by the empirical $\mathcal{K}_{\text{inf}}^{\bar{\mathcal{X}}_n}$ operator, which we recall is calculated over distributions bounded by $\bar{\mathcal{X}}_n = \max_{i \in \{1,\dots,n\}} X_i$ for a given set of observations $\mathcal{X} = (X_1,\dots,X_n)$. However, the regret lower bound Burnetas and Katehakis (1996) involves $\mathcal{K}_{\text{inf}}^{\mathcal{F}}$, calculated over distributions belonging to a base family $\mathcal{F}$. The analysis of NPTS (Riou and Honda, 2020) considers only distributions with bounded support, for which both operators can be made arbitrarily close in the relevant topology. We show that it is essentially the only favorable case and provide empirical evidence on the limit behavior of $\mathcal{K}_{\text{inf}}^{\bar{\mathcal{X}}_n}$ for standard unbounded distributions.

Combining intuitions from Lemma 3.3 and Lemma C.1 we obtain the following control of the empirical $\mathcal{K}_{\text{inf}}^{\bar{\mathcal{X}}_n}$.

**Lemma F.1** (Asymptotic behaviour of $\mathcal{K}_{\text{inf}}^{\bar{\mathcal{X}}}$). *Consider a set $\mathcal{X}_n = (X_1,\dots,X_n) \in \mathbb{R}^n$ and a target value $\mu \in \mathbb{R}$, and denote $\bar{\mathcal{X}}_n = \max_{i \in \{1,\dots,n\}} X_i$ and $\nu_{\mathcal{X}_n}$ the empirical distribution associated to $\mathcal{X}_n$. Assume that $\bar{\mathcal{X}}_n \geq g(n)$. Then,*

$$\mathcal{K}_{\text{inf}}^{\bar{\mathcal{X}}_n}(\nu_{\mathcal{X}_n}, \mu) = \mathcal{O}\left(\frac{1}{g(n)}\right) .$$

In particular, if the distribution of $X_i$ is unbounded, $g(n) \xrightarrow[n \to +\infty]{} +\infty$ and $\mathcal{K}_{\text{inf}}^{\bar{\mathcal{X}}_n}(\nu_{\mathcal{X}_n}, \mu) \xrightarrow[n \to +\infty]{} 0$. This shows that the $\mathcal{K}_{\text{inf}}$ operator is not continuous w.r.t the family over which it is defined in the sense that the following assertions are mutually exclusive:

(i) $\mathcal{F}$ contains an unbounded distribution $\nu$ and $\mathcal{K}_{\text{inf}}^{\mathcal{F}}(\nu, \mu) > 0$,

(ii) $\mathcal{K}_{\text{inf}}^{\bar{\mathcal{X}}_n}(\widehat{\nu}_n, \mu) \xrightarrow[n \to +\infty]{} \mathcal{K}_{\text{inf}}^{\mathcal{F}}(\nu, \mu)$.

A direct consequence of this is that it makes impossible a direct generalization of NPTS to unbounded distributions while preserving logarithmic regret, forcing us to either let go of logarithmic guarantees (RDS) or assume a quantile condition and use a truncation operator to recover the continuity between the empirical $\mathcal{K}_{\text{inf}}^{\bar{\mathcal{X}}_n}$ and $\mathcal{K}_{\text{inf}}^{\mathcal{F}}$ (QDS).

Lemma F.1 only provides a control of $\mathcal{K}_{\text{inf}}^{\bar{\mathcal{X}}_n}$ by $\frac{1}{g(n)}$, not an symptotic equivalent; we believe however this control to be quite tight. To sharpen our intuition, we compute $\mathcal{K}_{\text{inf}}^{\bar{\mathcal{X}}_n}$ for various sample sizes $n$ on classical SPEF (exponential and Gaussian with fixed variance, Bernoulli), using the dual formulation of Honda and Takemura (2010). For the unbounded SPEF (Figure 10), we see the empirical $\mathcal{K}_{\text{inf}}^{\bar{\mathcal{X}}_n}$ decreases away from $\mathcal{K}_{\text{inf}}^{\mathcal{F}}$ with $n$; by contrast, the Bernoulli distribution (Figure 11) shows no significant deviation from $\mathcal{K}_{\text{inf}}^{\mathcal{F}}$.

Furthermore, we empirically validate the relation between $\mathcal{K}_{\text{inf}}^{\bar{\mathcal{X}}_n}$ and the growth rate $g$ of the maximum of $n$ i.i.d samples. Indeed, we have $g(n) \approx \log n$ for exponential distributions and $g(n) \approx \sqrt{\log n}$ for Gaussian distributions, we therefore expect $\log \mathcal{K}_{\text{inf}}^{\bar{\mathcal{X}}_n}(\nu_{\mathcal{X}_n}, \mu) \approx -\log \log n$ and $\log \mathcal{K}_{\text{inf}}^{\bar{\mathcal{X}}_n}(\widehat{\nu}_n, \mu) \approx -\frac{1}{2} \log \log n$ respectively. Figure 12 shows the outcome of the least squares regression of $\log \mathcal{K}_{\text{inf}}^{\bar{\mathcal{X}}_n}(\nu_{\mathcal{X}_n}, \mu)$ on $\log \log n$, which recovers approximately the expected slopes. Again, the case of the Bernoulli SPEF shows no significant dependency on $n$ as $g(n) \approx 1$.

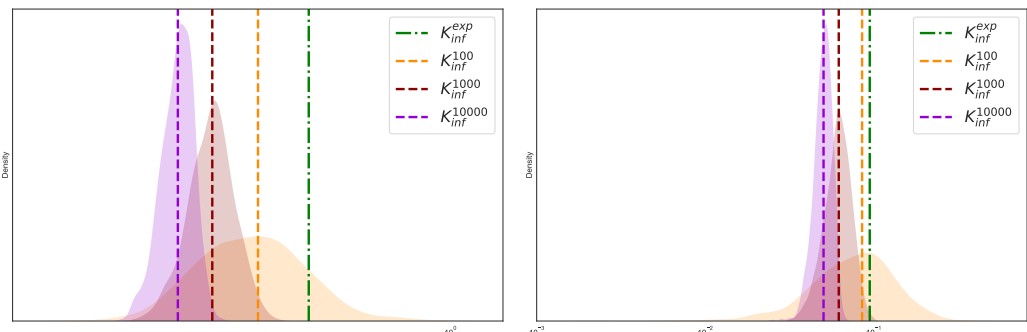

Figure 10: $\mathcal{K}_{\text{inf}}^{\mathcal{F}}(\nu, \mu^*)$ w.r.t the corresponding SPEF and empirical $\mathcal{K}_{\text{inf}}^{\bar{\mathcal{X}}_n}(\widehat{\nu}_n, \mu)$ for sample size $n = 10^2, 10^3, 10^4$, and $\mu^* = 3$, averaged over 1000 simulations (fitted density are shown in the background). Left: Gaussian $\nu = \mathcal{N}(2, 1)$. Right: exponential $\nu = \mathcal{E}(\frac{1}{2})$.

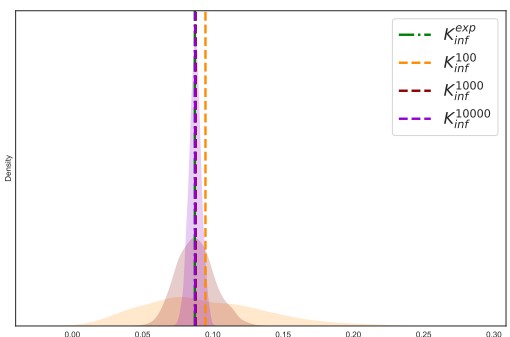

Figure 11: $\mathcal{K}^{\mathcal{F}}_{\inf}(\nu, \mu^*)$ w.r.t the Bernoulli SPEF and empirical $\mathcal{K}^{\bar{\mathcal{X}}_n}_{\inf}(\widehat{\nu}_n, \mu)$ for sample size $n = 10^2, 10^3, 10^4$, averaged over 1000 simulations (fitted density are shown in the background).

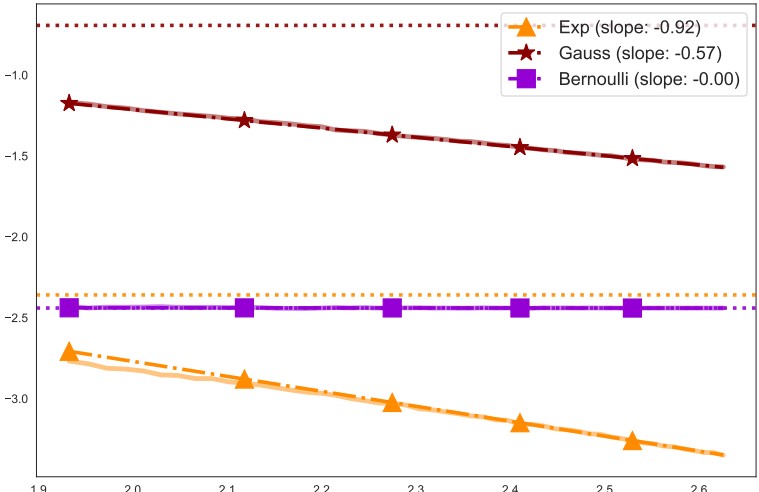

Figure 12: Linear fit (dashed) of $\log \mathcal{K}^{\bar{\mathcal{X}}_n}_{\inf}(\widehat{\nu}_n, \mu)$ (solid) on $\log \log n$ and resulting slopes. Dotted lines correspond to $\log \mathcal{K}^{\mathcal{F}}_{\inf}(\nu, \mu)$ for the corresponding SPEF $\mathcal{F}$. X-axis: $\log \log n$, Y-axis: $\log \mathcal{K}_{\inf}$.