# OpenReview forum: "From Optimality to Robustness: Adaptive Re-Sampling Strategies in Stochastic Bandits"
_NeurIPS.cc/2021/Conference — NeurIPS 2021 Poster_

### Official Review · Reviewer_8C9i · 2021-07-14

**Rating:** 7
**Confidence:** 2

**Summary:**

This paper considers robustness of MAB algorithm to model misspecification when knowledge on the distribution (e.g., tails) is not directly accessible. They study a generic Dirichlet Sampling algorithm and provide a generic theorem for regret decomposition. They applied the algorithm in three examples: 1) bounded distributions where optimal regret can be achieved, 2) light-tailed unbounded distributions where we can achieve slightly-worse-than-log regret through simple tuning strategies, and 3) unbounded distributions with some quantile condition where log regret is achievable.

**Limitations And Societal Impact:**

The authors discussed the limitations in the perspectives section. They look fair to me (as a non-expert reviewer in this particular line of work), and I look forward to their response to questions / comments in "Main Review".

**Main Review:**

*Originality*

I'm not too familiar with this specific line of literature so can't comment with full confidence on its originality. From my understanding, it fills in a gap in existing works: it achieves logarithmic regret without specifying the tail with precise parameters, and also avoids pathological distributions that are not likely to be true in practice by imposing a quantile condition.

*Quality*

- This paper tells a complete story with all theorems proved and their implications well explained. The Dirichlet Sampling bandit algorithm looks intuitive to me on a high-level.

- Some clarification questions:
1. For theorem 3.5, can you elaborate on how we get $g_k = O(\sqrt{\log n})$ for sub-gaussian and $O(\log n)$ for sub-exponential? Also, I'm a bit confused at its "robustness"; from my understanding of Theorem 3.5, $\rho_n$ can't be arbitrary and needs to be of order lower than $g_k$?
2. I don't fully understand why $\mathcal{T}_{\alpha}(\mu)(\{x\})$ should be defined as $\alpha$ at CVaR. Can you explain?
3. In Theorem 3.6, how does $b$ come into this regret? Why the current proof does not support $b=-\infty$? What's making $b=-\infty$ difficult?

*Clarity*

This paper is overall well organized. However, I do think it's a bit notation-heavy and sometimes makes the reading experience not so smooth (though authors do summarize notations in Appendix A). I am occasionally confused, for example, what's $M_{\nu, \mu, \rho}$ in line 280? Besides, I think it would be good if the authors could provide more intuitive explanations. For example, in "Logarithmic regret for unbounded distributions", it can be helpful to also verbally discuss what these quantities represent and why they are important, which can save some processing time for readers.

*Significance*

I view this submission as a solid contribution to a somewhat relevant problem. In particular, many distributions I see in practice do look like Figure 1, so it can be of some values to practitioners.


__________

After discussion: I think my questions were well-addressed by the authors, so I'm willing to raise my score.

**Time Spent Reviewing:**

3.5 hours

---

> ### Author Response · Authors · 2021-08-09
> **Author Response, reviewer 3**
>
> Thank you for your helpful feedback. In particular, we are pleased that you enjoyed both the theoretical aspect of the paper and our intent to provide algorithms aimed towards practitioners. Please find the answer to your technical questions below.
>
> * **Originality**: Your intuition is right and summarises well the purpose of this paper.
>
> * **On Theorem 3.5**: We consider *robustness* in the same sense as [Ashutosh et al. (2021)](http://proceedings.mlr.press/v130/ashutosh21a.html), i.e achieving consistent regret guarantees with respect to the assumptions on the distribution. RDS is then robust, because the regret bound of Theorem 3.5 holds *for any choice of $\rho_n=o(n)$* if the distributions are *light-tailed*.
> The regret is slightly larger than logarithmic, with a term $g_k(T)$ that depends on the growth rate of the maximum of a sequence of i.i.d observations drawn from arm $\nu_k$ *and* the exploration bonus depending on $\rho_n$ (see Appendix D.3 for the detailed proof).
> Regarding said growth rate, an upper bound on the expected maximum of $n$ light-tailed variables can be found for example in Theorem 2.5 of [Boucheron, Lugosi and Massart (2013)](https://hal.archives-ouvertes.fr/hal-00942704/). This bound turns out to be $\mathcal{O}\left(\sqrt{\log n}\right)$ and $\mathcal{O}\left(\log n\right)$ for sub-Gaussian and sub-exponential distributions respectively.
> We then discuss that if $\rho_n$ happens to grow slower than the largest observations, it has no influence on the asymptotic regret (which is noteworthy in our opinion). In appendix E.5 we discuss the choice of $\rho_n$ in an experiment with Gaussian arms, and in this example the choice of $\rho_n=\sqrt{\log n}$ or $\rho_n=\log n$ has little impact on the practical performance of RDS.
> We will rephrase the paragraph below Theorem 3.5 to improve its clarity.
>
> * **Truncation (QDS)**:  Consider the quantile $q_{1-\alpha}(\nu)$ of a distribution $\nu$, then by definition we know that the distribution has a mass $1-\alpha$ below the quantile, and a mass $\alpha$ after (assuming it is continuous). On $[b, q_{1-\alpha})$ the truncation operator do not change the distribution. Its role is just to *summarise the upper tail by its mean* (i.e CVaR$_{\alpha}(\nu)$ ). In order to maintain a probability distribution it simply *transfers the total mass $\alpha$* of this upper tail on this single observation, explaining the notation.
> The main interest of using the CVaR is that the truncated distributions have now the same expectations as the original ones, keeping the order of the arms.
>
> * **Semi-bounded support (QDS)** Actually, $b$ has no influence on the final theoretical result, because for any distribution $\nu$ supported on $[b, B]$ and any $\mu \in \mathbb{R}$, its $\mathcal{K}_{\text{inf}}^{\mathcal{F}}(\nu, \mu)$ does not change if $\mathcal{F}$ is the set of distributions supported on $[b, B]$ or the set of distributions supported on $[a, B]$ for any $a<b$ (including $a=-\infty)$.
>
> This result is proved for instance in Theorem 2 of [Honda \& Takemura (2015)](https://jmlr.org/papers/v16/honda15a.html), and is actually intuitive: the K-inf is obtained by finding an alternative distribution close to $\nu$ (in terms of KL divergence) but with a better mean. Hence, transferring some probability mass in the lower tail (on $[a, b)$) is clearly a bad move. Please note that this is the reason why we use the shorthand notation $\mathcal{K}_{\text{inf}}^{B} $ when distributions are semi-bounded for any (possibly unknown) lower bound.
> Now, allowing $b=-\infty$ for QDS would require additional technical challenges in the proof. To be precise, in Appendix D we use a technique based on a discretization of the support of the truncated distribution, requiring a finite size for this discretized support (see Lemma D.2 and l.1082-1109 for instance). Finding technical tools to remove this condition is an interesting future research direction.
>
> * **Notations/Intuitions**: Thank you for carefully listing some notations/points that need to be clarified, we will take care of them in our revision (e.g the "Logarithmic regret for unbounded distributions"). Note that we will include the changes listed in the "Erratum" p.13 in the revision (this will remove the unexplained $M_{\nu, \mu, \rho}$ you spotted l.280).

---

> ### Author Response · Authors · 2021-09-01
> **Follow-up**
>
> Dear reviewer,
>
> Thank you again for your interesting questions and positive feedback. Please let us know if our answers and our discussion with the other reviewers addressed your concerns. Do not hesitate to ask any additional question that would help you in your evaluation of the paper.
>
> Best,
> the authors

---

> > ### Comment · Reviewer_8C9i · 2021-09-02
> > **Thank you for addressing the questions**
> >
> > Dear authors,
> >
> > Sorry for the late reply and thank you for addressing my questions. I think the paper looks technically sound, so I'm willing to increase my score :)
> >
> > Yours,
> > Reviewer 3

---

### Official Review · Reviewer_MKzx · 2021-07-16

**Rating:** 7
**Confidence:** 3

**Summary:**

This work investigates a Dirichlet Sampling algorithm in stochastic multi-arm bandit problem. From the theoretical perspective, this work provides regret bounds of the DS algorithms for three different instances such as bounded, general light-tailed, and light-tailed distribution with additional quantile assumption.

**Limitations And Societal Impact:**

The last paragraph is denoted as "Perspective", but I suggest to have "Conclusion" section instead to make body better organized.

**Main Review:**

Originality:
It is a novel approach to combine non-parametric Thompson sampling with pairwise comparison between arms. Also, this duel-based structure makes it possible to have a general regret decomposition for index, which gives a theoretical foundation for three bandit setups.

Quality:
Theoretical results sound well-grounded and numerical experiments seem reasonably fair and supportive to this theoretical outcome, which makes a complete piece.

Clarity:
This submission starts with clear categorization for algorithm families, introduces sub-sampling based algorithm family, and develop new algorithms by combining existing algorithm along with novel duel structure. Theoretical arguments and following experimental results are overall well organized and easy to read.

Significance:
There are three algorithm categories for stochastic multi-arm bandit problems: UCB-type, TS-type algorithms, and sub-sampling based methods. This work focuses on third one, and extensively generalizes asymptotic theoretical guarantee to outside of single-para exponential family. Also, the DS algorithm this work suggested is useful in practice in that it does not require prior knowledge on the tail distributions.

**Time Spent Reviewing:**

4

---

> ### Author Response · Authors · 2021-08-09
> **Author Response, reviewer 2**
>
> Thank you for your positive feedback and your appreciation of this work. We are glad that you found the presentation clear and well-structured, and will further improve it following your suggestion to replace the *Perspectives* section by a (a bit more detailed) *Conclusion* in the final version. Furthermore, we appreciate your comments on the usefulness of our settings and theoretical guarantees for practitioners as the DS algorithms elude strong tail assumptions. This is indeed the key take-home message we wanted to convey.

---

### Official Review · Reviewer_37Ux · 2021-07-16

**Rating:** 7
**Confidence:** 3

**Summary:**

The goal of this paper is to construct a sequential decision-making algorithm under non-standard distributional assumptions. The authors propose a Dirichlet Sampling algorithm, employing pairwise comparisons of indices computed with the arms’ observations and a data-dependent exploration bonus.  They suggest three specific variants of this strategy under different distributional assumptions having distinct tail behaviors.  When it is applied to the bounded distributions, the algorithm is shown to be optimal. Numerical studies display some advantages of the proposed algorithms.

**Limitations And Societal Impact:**

In the numerical studies, don’t we need to include the UCB in the comparison list?

**Main Review:**

The authors study a class of algorithms that spins off nonparametric Thompson Sampling (NPTS) along the line of Subsample-mean comparisons (SSMC).  Their motivation is to handle so-called light-tailed distributions which cover more relaxed distributions than bounded or subgaussian distributions but avoids difficult situations of `mass leakage’ to infinity, which is the main cause of technical difficulties. The authors draw strengths of the distribution-free nature of NPTS and SSMC to compose a generic class of algorithms and give three specific cases with varying degrees of tail behavior. Regret analysis for each case and numerical studies are provided.

The paper is dense and has a lot of information, yet is well written. The authors blend existing ideas of NPTS and SSMC which do not rely on any particular distributional assumptions.    Bringing pairwise comparison through SSMC seems to be a smart idea to estimate the critical value through subsampling without resorting to the distributional properties.

It is nice to see finding a class of distributions satisfying Condition 2.  Some intuitive explanations on Condition 2 would be helpful.

In the numerical studies, don’t we need to include the UCB in the comparison list?

**Time Spent Reviewing:**

4 hours

---

> ### Author Response · Authors · 2021-08-09
> **Author Response, reviewer 1**
>
> Thank you for your helpful comments and enthusiasm, especially regarding our main idea of combining pairwise comparisons (SSMC) with distribution-free guarantees (NPTS). We address your two remarks below.
>
> * **UCB in experiments** To avoid cluttering, we decided to report only the best-performing competitor algorithms in the numerical experiments contained in the main paper. However, we included UCB in our results in Appendix E.2 along with other algorithms for a broader picture. We also compare RDS with the robust variant of UCB introduced in [Ashutosh et al. (2021)](http://proceedings.mlr.press/v130/ashutosh21a.html) in Appendix E.5, as they share similar robustness properties.
>
> * **Condition (C2)** Thanks for this question, we acknowledge that condition (C2) is indeed a bit technical, and will introduce it with more high level intuitions in the paper's revision.
> High-level idea: the LHS represents the expected cost in terms of regret of *under-estimating the optimal arm*: if arm $1$ collected bad samples how long will we wait before pulling it again? This is a classic decomposition in bandit analysis, and a counterpart of (C2) holds for most index policies with provable regret guarantees, for example we refer the reader to the following proofs:
>     * regret of GIRO with bounded distributions ($a_i$ in Theorem 1), [Kveton et al. (2019b.)](https://arxiv.org/abs/1811.05154).
>     * regret of ReBoot with sub-gaussian distributions ($a_k$ in Eq. (22)), [Wang et al. (2020)](https://arxiv.org/abs/2002.08436).
>     * Lemma 4 of the analysis of Bernoulli TS of [Agrawal \& Goyal (2012)](http://proceedings.mlr.press/v23/agrawal12/agrawal12.pdf).
>
> Still on (C2), we find interesting that after some work in the proof (Appendix B) we can exhibit a quantity that only depends on (1) the distribution of the best arm, and (2) the index computed by the algorithm if the best arm is a challenger, and is then independent of the run of the bandit algorithm. We think it makes the presentation of the proofs of Theorem 3.4, 3.5 and 3.6 easier. We will add these intuitions in the revision.

---

### Author Response · Authors · 2021-08-09
**Author Response: general comment**

We thank the reviewers for their careful reading and their useful feedback. We also thank them for their shared enthusiasm for the main contributions of the paper: (1) the design of algorithms that are suitable for the practitioners, by reconsidering the assumptions on the arms' distributions, (2) the idea of pairwise comparisons (duels) that allowed to derive a generic regret bound with data-dependent exploration bonuses, and (3) the three instances of Dirichlet Sampling we propose and analyse in Section 3, aimed at tackling different objectives. Furthermore, we are glad to see that the paper was clear despite the technicality of some notations and results. Following your suggestions, our revision will be mostly aimed at further improving its clarity (high level explanations (R1, R3), conclusion (R2), detailing notations and technical points (R3)).

We also thank the reviewers for their interesting questions, that we answer below in individual threads.

---

### Author Response · Authors · 2021-09-01
**Follow-up**

Dear Reviewers,

We hope that our answers comforted your opinion on the paper. Even if the discussion period ends soon, we would be glad to answer any additional question.

Best,
the authors.

---

### Decision · Program_Chairs · 2021-09-27

**Decision:**

Accept (Poster)

**Comment:**

I believe there is a strong consensus around this paper's novelty and its contributions, and on that note it is also a good chance to thank the reviewers and authors for the informative dialogue. From my own reading I can add that while the problem is well motivated, the exposition in various parts and the notational conventions make the paper less accessible than it can be and this is something that can be improved with the final version (I hope). It may also be worth commenting further on the nature of the histograms and the relevance to ones seen in various modeling instances, e.g., mixture models,  to ensure these are not perceived as just esoteric one-off examples.